

# Knots and entanglement

**Jin-Long Huang, John McGreevy and Bowen Shi**

Department of Physics, University of California at San Diego, La Jolla, CA 92093, USA

## Abstract

We extend the entanglement bootstrap program to (3+1)-dimensions. We study knotted excitations of (3+1)-dimensional liquid topological orders and exotic fusion processes of loops. As in previous work in (2+1)-dimensions [1, 2], we define a variety of superselection sectors and fusion spaces from two axioms on the ground state entanglement entropy. In particular, we identify fusion spaces associated with knots. We generalize the information convex set to a new class of regions called immersed regions, promoting various theorems to this new context. Examples from solvable models are provided; for instance, a concrete calculation of *knot multiplicity* shows that the knot complement of a trefoil knot can store quantum information. We define *spiral maps* that allow us to understand consistency relations for torus knots as well as spiral fusions of fluxes.

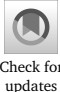

# 1   Introduction

The entanglement bootstrap [1,2] is a program to understand liquid topological orders from a couple of well-motivated axioms concerning the structure of entanglement in a single ground state wave function. By a topological order [3,4], we mean a gapped phase of matter with a ground state subspace of locally indistinguishable ground states whose dimension depends on the topology of space. By 'liquid', we mean that this ground state degeneracy does not depend on the geometry; this excludes fractons.

   The program thus far has been focused on bosonic topological order in two spatial dimensions. It has been successful in explaining much of the structure of the mathematical theory of anyons, from a very simple starting point; this includes the explanation of the fusion rules of anyons [1] and the nondegenerate mutual braiding statistics after that [5]. The axioms follow from the area law of the entanglement entropy [6,7] but are weaker. These axioms, which can be stated in the approximate case, can be useful for the study of 2d chiral systems with a bulk energy gap and gapless edge modes, whose thermal Hall conductance is quantized [8,9] according to its chiral central charge [10]. In fact, with related techniques, a formula for the chiral central charge, in terms of a single bulk ground state wave function, has been argued and tested [11,12]. No rigorous derivation of this formula is known, at the moment, which calls for new innovation of concepts and techniques of the framework.

   In addition to the goal of deriving the emergent physical laws of 2d gapped phases, one hopes that this program will lead to new insights in broader physical contexts. An example where this has already happened is in the theory of gapped domain walls [2,13].

   A central notion of the entanglement bootstrap is the *information convex set*. Given a reference state $|\psi\rangle$ on some manifold, this machine associates with any subsystem $X$ (with boundary) a convex set of density matrices, $\Sigma(X, |\psi\rangle)$. Roughly, these are all the density matrices on the subsystem that are indistinguishable from the reference state on balls. The basic observation is that $\Sigma(X, |\psi\rangle)$ depends only on the deformation class of both $|\psi\rangle$ and $X$. We will often omit the $|\psi\rangle$ label on $\Sigma(X)$. For various choices of the topology of $X$, $\Sigma(X)$ encodes the information about the topological excitations (anyons, for the case of 2d), as well as their fusion spaces.

   In this work, we ask about the extension of this program to three spatial dimensions (3d). For simplicity, we continue to restrict attention to systems made of bosons. The main new ingredient in three dimensions is that there are many more ways to choose a subsystem. Each choice of (deformation class of) subsystem, then, has a role to play in the theory of 3d topological order. As in two dimensions, these include identifying excitation types and their possible fusion spaces. As we will explain, one new ingredient is the fusion space associated with knotted loop excitations. As an example, we show that a single trefoil excitation on a 3-sphere can be associated with a nontrivial *knot multiplicity*, and thus a topological qubit may be stored in the knot complement of a trefoil knot.

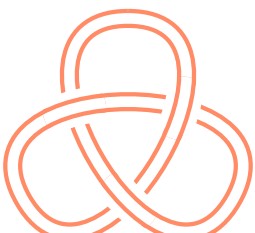

We begin in §1.1 with a brief overview of entanglement bootstrap, starting from the axioms. After that, we present an overview of the whole paper, focusing on physical intuition (§1.2). We postpone the comparison of our findings and other approaches of 3d topological orders to the discussion section.

## 1.1 Brief overview of 3d entanglement bootstrap

Without further ado, here are the axioms for the entanglement bootstrap in three dimensions. We assume given a *reference state $\sigma$* supported on a large ball (much larger than any other length scales such as correlation lengths or lattice spacing). About this state, we assume that the following two axioms hold on all balls contained in the large ball:

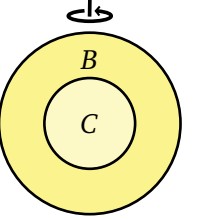   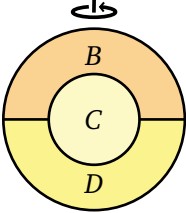

$$\mathbf{A0}: \ (S_{BC} + S_C - S_B)_\sigma = 0 \qquad \mathbf{A1}: \ (S_{BC} + S_{CD} - S_B - S_D)_\sigma = 0\,, \tag{1.1}$$

where $(S_X)_\sigma = -\mathrm{Tr}(\sigma_X \ln \sigma_X)$ is the von Neumann entropy of the state $\sigma$ reduced to the region $X$, and each region is the volume of revolution of the indicated 2d region.[1] We will find it convenient to denote the entropy combinations appearing in the axioms as

$$\Delta(B,C) \equiv S_{BC} + S_C - S_B\,, \qquad \Delta(B,C,D) \equiv S_{BC} + S_{CD} - S_B - S_D\,. \tag{1.2}$$

We note that if we erase the rotation label, these axioms have the same form as the axioms in the 2d case [1].

As in the 2d case [1], assuming the axioms only on bounded-radius balls can be used to prove the same conditions on larger balls by application of strong subadditivity (SSA) [14]. Similarly, the axioms can be shown to hold on any collection of regions with the indicated topology. The axioms (1.1) follow from a strict area law for the von Neumann entropy but are strictly weaker (for example, in two dimensions, regions with corners may violate the strict area law in chiral states while still satisfying the axioms up to an error exponentially decaying with the system size).

The central actor in this work is the *information convex set* [15–17], which, given a reference state $\sigma$ satisfying the axioms **A0** and **A1**, associates to each region of space $X$ a convex

---

[1]In the figure, we used a rotation label to indicate the revolution. Note, however, we do not assume rotation symmetry; the axioms apply to any partition of balls topologically equivalent to these.

set of density matrices, $\Sigma(X)$. (The dependence on the state $\sigma$ is implied.) Informally, $\Sigma(X)$ is the set of density matrices on $X$ that agree with the reference state on any ball in $X$. A more precise definition enlarges the region $X$ slightly to take care of balls that overlap the boundary of $X$.

The information convex set is a topological invariant in two senses. First, it is a property of the phase of matter represented by the reference state $|\psi\rangle$ – it is invariant under adiabatic deformations of $|\psi\rangle$. Secondly, two regions related by a regular homotopy have isomorphic information convex sets; this is the content of the Isomorphism Theorem (Theorem 2.5). The isomorphism in question maps extreme points to extreme points and preserves entropy differences and distances between states.

The structure of a (compact) convex set is determined by its extreme points. If the extreme point density matrices of $\Sigma(X)$ are all mutually orthogonal, then $\Sigma(X)$ is a simplex. In this case, the states of $\Sigma(X)$ encode only classical information, which may be copied. In contrast, the information convex sets of some regions contain *fusion spaces*, which can store quantum information. The relation between the geometry of a region and these properties of the information convex set is understood [1], and in §2.2, we review the dichotomy between sectorizable regions [2], whose information convex set is a simplex, and non-sectorizable regions, whose states can encode quantum information. Sectorizable is a simple geometric condition: a region is sectorizable if it contains two disjoint pieces such that each can be deformed back to the whole via a sequence of extensions.

A crucial tool for understanding the structure of the information convex sets is *merging* [1, 18]: given elements $\rho, \tau$ of the information convex sets of two intersecting regions $ABC$ and $BCD$, which agree on the overlap $BC$, a unique element of the information convex set of their union $\rho \bowtie \tau \in \Sigma(ABCD)$ can be obtained, if a quantum Markov chain condition is satisfied. This merging is possible because quantum Markov chain conditions on large regions can be deduced from SSA and the axioms on local regions [1, 19]. The merged state $\rho \bowtie \tau$ is the maximum-entropy state consistent with its marginals. This process has a lot in common with the topological notion of *surgery*. Meanwhile, merging is flexible enough to glue either part of a boundary component or whole boundary components.

## 1.2 Heuristic overview of the paper

The main text of this paper attempts to take care of both physical ideas and the mathematical rigor needed for building a theoretical framework. Despite the many illustrations, this unavoidably makes multiple places lengthy and more technical than needed in grasping the physical idea. For this reason, we include the following overview of its contents from a physics perspective.

Section 2 develops the entanglement bootstrap technology, focusing on novelties in 3d, as compared to 2d. The first important innovation (§2.1) is the use of what we call *immersed regions*. The idea, first used in [5], is to make regions with non-trivial topology by realizing them as regions *locally* (but not globally) embedded in a ball. It is well-established that non-trivial spatial topology is a useful probe of topological order; the method of immersed regions allows us to exploit non-trivial spatial topology even when given only a single wave function on a region with the topology of a ball.

A key step in §2.1 is to show that the Isomorphism Theorem generalizes to include immersed regions, and moreover to include regular homotopies between regions that allow immersed regions in the intermediate steps of deformation (even if the starting and ending regions are embedded). This is the content of the Generalized Isomorphism Theorem, 2.5. An immediate consequence of the Generalized Isomorphism Theorem is that any two thickened knots have the same information convex set. (The information convex set of a *knot complement*, however, will be much more interesting.)

It is not a priori clear that the information convex set of an immersed region has a vacuum sector. One important role the vacuum sector plays is in the entanglement bootstrap definition of the quantum dimension of an extreme point: $d_a = e^{(S(\rho^a)-S(\rho^1))/2}$. This motivates an alternative, more general, definition of quantum dimension, described in Appendix D, in terms of a linear combination of entropies of subregions. The existence of a vacuum sector is proved, however, for any (embedded) subregion of a ball by a partial trace of the reference state in Lemma 3.2.

In §2.3, we describe what we call the Associativity Theorem (Theorem 2.22), which describes how to build the information convex set of a region by decomposing the region into parts. Lemma 2.20 shows that when merging two extreme points along a whole component of their boundary, we always produce an extreme point of the merged region. The proof of this lemma uses a nice necessary and sufficient criterion for a state to be an extreme point, given in Lemma 2.13. The Associativity Theorem 2.22 then says that *all* extreme points of the merged region $\Omega = \Omega_L \Omega_R$ arise in this way, so that their fusion multiplicities satisfy

$$N(\Omega) = \sum_I N_I(\Omega_L) N^I(\Omega_R), \tag{1.3}$$

where $I$ labels an extreme point of the thickened boundary component along which $\Omega_L$ and $\Omega_R$ were merged.

The purpose of §3 is to study the information convex set for various 3d regions and to make connections to the fusion data and fusion processes of 3d topological order. We said above that the method of immersed regions allows us to build states on non-trivial topological spaces, starting with just a state on a ball. But so far, these spaces must be immersed in a ball, and this is not always possible. For example, a torus $T^2$ cannot be immersed in a disk, but a torus with a hole ($T^2 \setminus$ disk) can be. In 3.1, we prove the Sphere Completion Lemma 3.1, which shows that a state on a manifold with a hole can be used to produce a state on the closed manifold obtained by filling in the hole. One way in which we will repeatedly use the Sphere Completion Lemma is to turn regions inside-out by deforming them to the point at infinity on the sphere. An immediate application of this technique is to study the complement of knots on a 3-sphere and to gain an additional intuition on various anti-sectors.

§3.2 explains the structure of the information convex set for basic sectorizable regions in 3d:

- the sphere shell, whose extreme points label particle excitations

- the solid torus, whose extreme points label a class of loop excitations called *pure fluxes*, and

- the torus shell, whose extreme points label more general Hopf link excitations.

Another class of 3d sectorizable regions is the handlebodies of genus $g > 1$; their extreme points are labeled by graph excitations with $g$ edges. We defer their discussion to [20]. In 2d, there is a unique notion of total quantum dimension, defined by $\mathcal{D} = \sqrt{\sum_a d_a^2}$. In 3d, one could a priori define such a notion for each type of excitation listed above. Proposition 3.4 shows that they are all equal.

In §3.3 we translate known results about fusion spaces of particles and loops into statements about information convex sets. For example, the fusion of two particles to a third particle is encoded in the structure of the information convex set of a ball minus two balls. In Appendix E we show that a procedure of dimensional reduction precisely relates all of the above information to an entanglement bootstrap problem in 2d with a special kind of gapped boundary [2]. The idea is that each of the above regions enjoys an action of revolution; the loci where the circle action is not free produces a gapped boundary.

It is not true, however, that *all* information about 3d topological orders can be obtained by dimensional reduction. In §3.4, we initiate the study of information convex sets of knot complements. These regions are not sectorizable, and the information convex set with a specific sector of its torus-shell boundary is the state space of some finite-dimensional Hilbert space. Its dimension, which we call a *knot multiplicity*, is an isotopy invariant of the knot. In §3.5, we identify several exotic fusion processes of flux loops. One example is that a flux loop deforms into a spiral and then makes a new flux loop. We analyze this with the spiral maps we introduce in §5. Another example is a fusion process of two loops related to the Borromean rings complement.

The main goal of the rest of the paper is to understand the structure of these knot multiplicities and the spiral fusion of fluxes.

In §4 we use a technique called *minimal diagram* (explained in §2.4) to compute basic information convex sets for the 3d quantum double model. This includes the complement of the trefoil knot. In particular, for a general 3d quantum double model, we demonstrate a close relationship between the structure of the information convex set of a region and its fundamental group. Specifically, the structure of the information convex set of a knot complement is determined by (the Wirtinger presentation of) its fundamental group, *the knot group* (a nice description can be found in [21]). When we wish to be completely explicit, we specialize to the case of 3d quantum double with gauge group $S_3$, the smallest non-abelian group. Further examples, including the Borromean rings complement, are explained in Appendix C.

In §5 we introduce a family of maps on information convex sets that we call *spiral maps*. The simplest spiral map takes $\Sigma(T)$, the information convex set of the solid torus, to itself. It is defined by tracing out the complement of a sub-solid-torus that winds around the hole multiple times, followed by a deformation that unwinds it, to map back to a state on the original region. This map takes extreme points to extreme points, and thus it takes fluxes to fluxes. As a logical consequence, the spiral fusion of any flux provides a unique flux as the outcome; moreover, the quantum dimension of flux cannot increase under such a process. The spiral maps can be composed. In the special case of the 3d quantum double model, it is a realization of the group multiplication law, which is information that goes beyond the character table. The spiral maps will play an important role in relating knot multiplicities to each other.

In §6, we illustrate consistency conditions between the fusion spaces and quantum dimensions associated with various regions, including torus knots. There are two kinds of consistency conditions. The simplest kind involves chopping up a region along some hypersurface internal to the region. Then the Associativity Theorem in the form (1.3) determines the fusion multiplicities of the whole region in terms of a convolution of those of its parts. If, instead, we chop up a region along a cut that intersects a region's boundary so that we are not merging along a whole boundary component, we obtain a relation that involves the quantum dimensions of the labels on the cut boundary. In particular, we derive a consistency relation for torus knots which involves knot multiplicities, quantum dimensions, and spiral maps. This provides an upper bound on torus knot multiplicities in terms of the total quantum dimension.

## 2 General theorems and calculation tools

In this section, we describe basic concepts and general theorems of the entanglement bootstrap approach in 3d. Most of them can be stated in a general context and originate from recent literature [1, 2]. In addition to reviewing these known results, we highlight two innovations that will have a broad application in 3d. The notable new concept we introduce is *immersed*

*region*[2] (see *e.g.* Fig. 2), which extends the scope of previous results to this broader type of region and also makes the information-preserving deformations of a region more flexible, by allowing the region to "pass through itself"; see Theorem 2.5. The second innovation is the associativity theorem presented in §2.3.

We further provide a discussion on a calculation tool for solvable models; see §2.4. While this is not part of the entanglement bootstrap, the approach works for solvable models whose ground state(s) satisfy axioms **A0** and **A1**. Therefore, it is capable of providing concrete examples, demonstrating the various objects and consistency relations predicted by entanglement bootstrap.

## 2.1 Immersed regions and generalized isomorphism theorem

### 2.1.1 Immersed regions

When studying a quantum many-body system on a spatial manifold $M$, it is natural to consider subsystems, which are embedded manifolds (of the same dimension as $M$) with boundary. The concept of subsystem has played an important role in entanglement bootstrap. This is because information convex sets, the isomorphism theorem, and structure theorems are associated with subsystems.

Immersed regions are natural generalizations of subsystems. They are locally embedded in the physical system but may not be globally embedded. For our purpose, we shall be interested in immersions of regions that are of the same dimension as $M$; see Fig. 1 for an illustration. As we shall discuss, most of the nice properties associated with subsystems generalize to immersed regions.

We think of the physical system on $M$ as a coarse-grained lattice that possesses a finite-dimensional Hilbert space on each site; we always work with a large enough length scale so that the manifold can be treated as smooth. A quantum state $\sigma$, which we shall call the reference state, is defined on the total Hilbert space of the physical system. In this work, we shall focus on bosonic systems by assuming that the total Hilbert space is the tensor product of the onsite Hilbert spaces. Let $\Gamma(M)$ be the set of bounded-radius balls on $M$, for which the axioms are imposed.[3]

**Definition 2.1** (Immersed region)**.** An immersed region is specified by either $(\widehat{\Omega}, i)$ or $\Omega$, as we will discuss. When useful, we write $\Omega$ as $(\Omega, \mathfrak{p})$. (See Fig. 1 for an illustration.)

*The first definition,* $(\widehat{\Omega}, i)$: Let $\widehat{\Omega}$ be a topological manifold (possibly with boundary), which has the same dimension as $M$. Consider a continuous map

$$i : \widehat{\Omega} \to M . \tag{2.1}$$

We call this map an *immersion map* if the preimage of any ball $b \in \Gamma(M)$, for which $b \subset i(\widehat{\Omega})$, is the union of disjoint balls of $\widehat{\Omega}$, and $i$ is a homeomorphism (between a region and its image under $i$) when restricted to any of these disjoint balls of $\widehat{\Omega}$. We say $(\widehat{\Omega}, i)$ defines an immersed region when $i$ is an immersion map.

*The second definition,* $\Omega$ *or* $(\Omega, \mathfrak{p})$: Let $\Omega$ be a topological manifold of the same dimension as $M$, with coordinates: $(x, q) \in \Omega$ with $x \in M$ and $q$ is a discrete, finite variable that specifies

---

[2]This includes non-subsystem regions whose quantum states can nevertheless be constructed from the reference state, an observation that first appeared in [5].

[3]The set covers $M$ and adjacent balls overlap with each other.

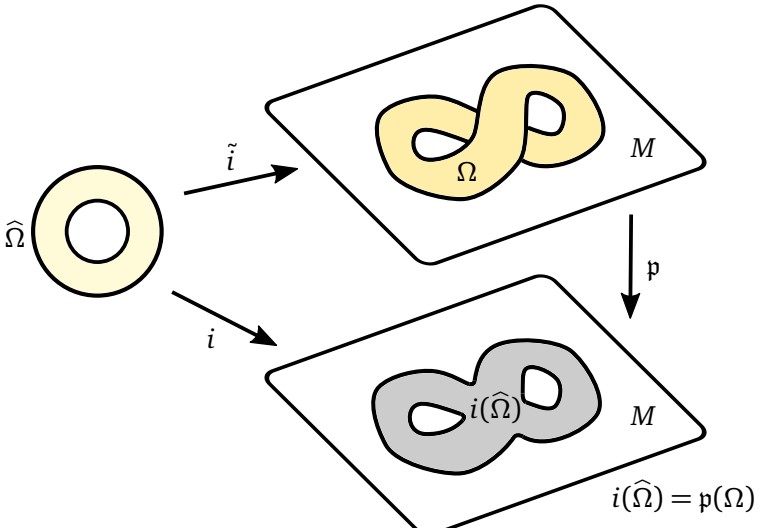

Figure 1: An illustration of the concept of immersed region (Definition 2.1). $M$ is the spatial manifold of the physical system. $\widehat{\Omega}$ is an abstract topological manifold with the same dimension as $M$. $i$ is the immersion map. $\tilde{i}$ is an homeomorphism such that $i = \mathfrak{p} \circ \tilde{i}$. Note that the image $i(\widehat{\Omega})$ (or $\mathfrak{p}(\Omega)$) usually has a different topology than $\widehat{\Omega}$. $\Omega$ is an immersed region whose coordinate determines $\mathfrak{p}$.

the information of the layers and branch cuts;[4] the topology of $\Omega$ is consistent with the map

$$\begin{aligned} \mathfrak{p}: \quad \Omega \quad &\to M\,, \\ (x,q) &\mapsto x\,, \end{aligned} \tag{2.2}$$

such that $(x, q) \in \Omega$ is mapped to $x \in M$ by $\mathfrak{p}$. $\Omega$ defines an immersed region if the preimage of any ball $b \in \Gamma(M)$ for which $b \subset \mathfrak{p}(\Omega)$ is the union of disjoint balls of $\Omega$, and $\mathfrak{p}$ is a homeomorphism (between a region and its image under $\mathfrak{p}$) when restricted to any of these disjoint balls of $\Omega$. We write $\Omega$ as $(\Omega, \mathfrak{p})$ when we wish to specify $\mathfrak{p}$. We emphasize that $\mathfrak{p}$ is part of the information of $\Omega$.

To relate these two definitions, we let $\tilde{i}$ be a homeomorphism from $\widehat{\Omega}$ to $\Omega$; see Fig. 1. Such a homeomorphism must exist with an appropriate choice of $\widehat{\Omega}$ because $\Omega$ has a manifold structure. The two alternative definitions of immersed region, $\Omega$ and $(\widehat{\Omega}, i)$ are related by

$$\Omega = \tilde{i}(\widehat{\Omega})\,, \quad \mathfrak{p} = i \circ \tilde{i}^{-1}\,. \tag{2.3}$$

The immersion in the definition above is precisely local embedding.[5] In general topology, an embedding is a homeomorphism onto its image. A local embedding (or local homeomorphism) is a map $\mathfrak{p} : \Omega \to M$ with the property that for each point $x \in \Omega$ there is a neighborhood $U$ containing $x$ such that $\mathfrak{p}(U)$ is an open subset of $M$ and $\mathfrak{p}|_U$ is a homeomorphism. Embedding is a special type of immersion. In that case, $\mathfrak{p}$ can be taken to be the identity map. For

---

[4]The assignment of layers and branch cuts determines the topology of $\Omega$. Note that the assignment is not without possible redundancy, e.g., branch cuts can freely deform without changing the topology. We are only interested in the assignment up to this redundancy.

[5]The terminology 'immersion' is motivated by its use in differential topology. In that context, local embedding is achieved by a condition on tangent space. (Roughly speaking, this condition is for having no sharp corners.) We are only interested in immersion between manifolds with the same dimension. In this case, immersion is also called submersion. We do not consider the tangent space because we are interested in systems made of qudits, which are discrete on small scales.

our purpose, an immersed region $\Omega$ will always have the same dimension as $M$. An embedded region is a subsystem.

It is tempting to use the term 'covering space' for what we call here 'immersed region.' However, a covering space must have the same number of preimages of each point in the image; in contrast, as can be seen in Fig. 1, in an immersed region, the number of sheets (range of the discrete coordinate $q$) is allowed to vary over the image. The crucial difference in the definition is that we only demand that $i$ is a homeomorphism from each component of its preimage *to its image* and not to a fixed set in $M$. The former is weaker.

The Hilbert space of an immersed region is defined as the tensor product of the local Hilbert spaces of its embedded local patches.

**Definition 2.2** (Hilbert space of immersed region)**.** If $\Omega$ is an immersed region, we define the Hilbert space of $\Omega$ (denoted as $\mathcal{H}_\Omega$) as the tensor product

$$\mathcal{H}_\Omega = \otimes_i \mathcal{H}_{\mathfrak{p}(A_i)}, \tag{2.4}$$

where $\Omega = \prod_i A_i$ is a partition of $\Omega$ into local patches $\{A_i\}$, such that each patch $A_i$ is embedded into $M$ by $\mathfrak{p}$. $\mathcal{H}_{\mathfrak{p}(A_i)}$ is the Hilbert space of subsystem $\mathfrak{p}(A_i)$.

**Remark.** Below are a few remarks on Definition 2.1:

1. Either $\Omega$ or the pair $(\widehat{\Omega}, i)$ is sufficient for defining the immersed region $\Omega$. We kept both of them because each provides a useful perspective. $\Omega$ is the most convenient for visualization of the topological region and the Hilbert space associated with it; the pair $(\widehat{\Omega}, i)$ is insightful in relating smooth deformations to the concept of regular homotopy.

2. How is the immersed region $\Omega$ consistent with the previous idea of subsystems? If $\Omega$ is a subsystem, the assignment of the layer is trivial. Every point on $\Omega$ is in the same layer. We can omit the data $q$ in the coordinate $(x, q)$, and then $\mathfrak{p}$ becomes the identity operator.

3. Consider $i' = i \circ u$, where $u$ is a self-homeomorphism of $\widehat{\Omega}$. We consider $i$ and $i'$ as equivalent, and we shall only be interested in immersion maps up to this equivalence relation.

4. The *second* definition of the immersed region can be alternatively formulated without using "layers and branch cuts" as follows. (We emphasize that this idea is not necessary, but it may help some readers.) Think of $\Omega$ as an embedded region in a space with tiny extra dimensions: $M \times E$, where $E$ is an $\epsilon$-ball with high enough dimensions. $\Omega$ is embedded in $M \times E$ in such a way that points in $\Omega$ has coordinate $(x, e)$ where $x \in M$ and $e \in E$. $\mathfrak{p}$ maps $(x, e)$ to $x$. (Again, we do not care about the detailed assignment of coordinates $e$ as long as it works.) In this way, $\Omega$ lies on top of $M$, and its coordinates contain the information of $\mathfrak{p}$.

5. In practice, the drawing of branch cuts can often be omitted, supplemented with other ways to specify the layering. This is done in Fig. 1. See Fig. 2 (b1) and (b2) for further explanations.

**Information convex sets for immersed regions:** We are ready to define information convex sets for immersed regions. The definition below is a direct generalization of the original definition.[6] It reduces to the original definition if the region is a subsystem. As in the original references, when we define an information convex set, we are interested in a region that can

---

[6]See Definition 3.1 of [1] as well as the alternative Definition C.1 in the same reference.

be thickened. Below, we always consider an immersed region $\Omega$ that can be identified as the interior of some $\Omega_+$, where $(\Omega_+, \mathfrak{p})$ is an immersed region.

Not every immersed region can be thickened. For instance, $\Omega$ in Eq. (2.5) can be thickened into $\Omega_+$. In contrast, $\Omega_+$ cannot be thickened into any immersed region.

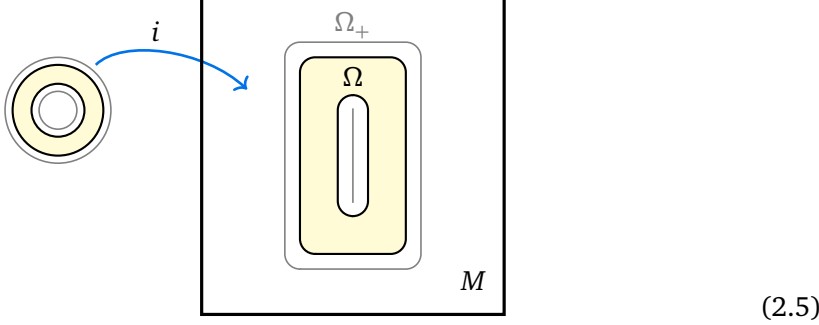

$$(2.5)$$

Recall that we have a reference state $\sigma$, defined on $M$. The total Hilbert space of the physical system is assumed to be a tensor product of finite-dimensional local Hilbert spaces associated with the lattice sites. The reference state $\sigma$ satisfies the two axioms **A0** and **A1** on bounded radius balls $b \in \Gamma(M)$. Because $(\Omega_+, \mathfrak{p})$ is an immersed region, we can obtain a set of balls in $\Omega_+$ by finding the connected components of the preimage of $b \subset \mathfrak{p}(\Omega_+)$. We call this set of balls of $\Omega_+$ as $\widetilde{\Gamma}(\Omega_+)$.

Balls in $\widetilde{\Gamma}(\Omega_+)$ are embedded into $M$ by $\mathfrak{p}$. Therefore, according to the definition of the Hilbert space $\mathcal{H}_{\Omega_+}$, there is a natural way to say a state $\rho_{\Omega_+}$ is locally indistinguishable from the reference state. Let $b \in \Gamma(M)$, where $b \subset \mathfrak{p}(\Omega_+)$. We say a state $\rho_{\Omega_+}$ is *consistent* with $\sigma_b$ if $\rho_{\Omega_+}$, reduced to any connected component of the preimage of $b$, is identical with $\sigma_b$ or its reduced density matrix (after mapping to the physical system according to $\mathfrak{p}$). We denote this consistent condition by

$$\rho_{\Omega_+} \stackrel{c, \mathfrak{p}}{=\!=\!=} \sigma_b. \tag{2.6}$$

**Definition 2.3** (Information convex set for immersed region)**.** For an immersed region $\Omega$, which can be thickened into immersed region $(\Omega_+, \mathfrak{p})$, the information convex set $\Sigma(\Omega)$, for a given reference state $\sigma$, is the set of density matrices

$$\Sigma(\Omega) \equiv \{\rho_\Omega | \text{conditions } 1, 2\}, \tag{2.7}$$

where the two conditions are

1. $\rho_\Omega = \mathrm{Tr}_{\Omega_+ \setminus \Omega} \rho_{\Omega_+}$, where $\rho_{\Omega_+}$ is a density matrix on $\Omega_+$.
2. $\rho_{\Omega_+} \stackrel{c, \mathfrak{p}}{=\!=\!=} \sigma_b$ for any $b \in \Gamma(M)$ such that $b \subset \mathfrak{p}(\Omega_+)$.

We shall always consider regions $\Omega$ containing a finite number of sites to avoid infinite-dimensional Hilbert spaces. An immediate consequence is that the information convex set $\Sigma(\Omega)$ must be a compact convex set. Therefore, the convex set is completely determined by the set of *extreme points*. We shall denote the set of extreme points as $\mathrm{ext}(\Sigma(\Omega))$.

**Remark.** For immersed regions that are not subsystems, the existence of reference state $\sigma$ (defined on $M$) does not immediately guarantee that $\Sigma(\Omega)$ is nonempty. This is because on the immersed region $\Omega_+$, which thickens $\Omega$, we have a set of local density matrices rather than a global reference state. We do not know if every immersed region has a *nonempty* information convex set. However, for all examples studied in this paper, we can verify the nonemptiness using the merging technique. A simple example is an immersed disk; see Example 2.4 below.

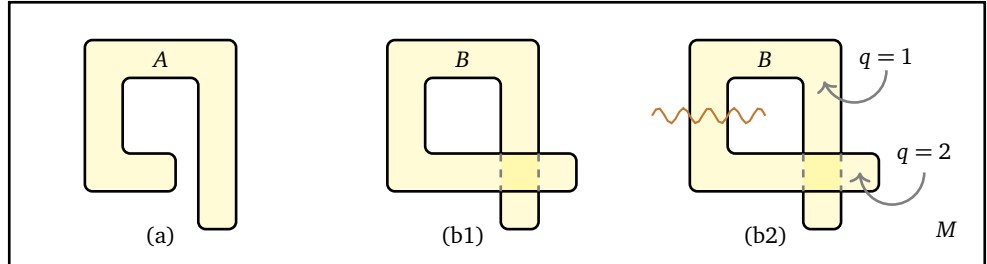

Figure 2: Examples of immersed disks in 2d. (a) $A$ is embedded. (b) $B$ is an immersed disk that is not embedded. (b1) or (b2) are without or with explicit labeling of the layers and branch cuts.

**Example 2.4** (Immersed disks). Here we consider the immersed disks illustrated in Fig. 2. Region $A$ in Fig. 2(a) is an embedded disk, which can be understood in ways described in the previous literature. The more nontrivial case, which we discuss in detail, is $B$ in Fig. 2(b). It is an immersed disk that is not a subsystem. Let us relabel $B$ as $(B, \mathfrak{p}_B)$ to specify the map $\mathfrak{p}_B$. $\mathfrak{p}_B(B)$ is the image of $B$ on $M$, and it is an annulus rather than a disk. Nonetheless, $\mathfrak{p}_B : B \to M$ is a local embedding, meaning it maps small balls of $B$ onto small balls of $M$.

Figure 2(b1) and (b2) are two ways to describe $B$. While (b2) gives more details, we shall prefer (b1) since it is more concise and contains the same necessary information (up to equivalence). We start by explaining (b2). Recall that $B \ni (x, q)$ according to the second definition of immersed region (in Definition 2.1). $q$ is a discrete label of the layers. Here, part of $B$ is labeled by $q = 1$, and the rest is labeled by $q = 2$. A branch cut separates the two layers. The way we draw $B$ on top of $M$ specifies the $x$ coordinate. (b1) omitted the branch cut and the layer labels $q = 1, 2$. However, (b1) contains the same information for the following reasons. First, the layering is implied by the drawing of the two layers with transparency (opaque in the case of Fig. 1). Second, the branch cut can deform freely and its precise location is unimportant; (b1) omitted this nonessential information.

The Hilbert space structure of $M$ induces a Hilbert space on $A$ and $B$ through Definition 2.2. Note that the Hilbert space associated with immersed disk $B$ is larger than the Hilbert space associated with the image $\mathfrak{p}_B(B)$. This is because the overlapping region appears twice in the tensor product form of $\mathcal{H}_B$ (by Eq. (2.4)). In other words, some qudits are "recycled".

Information convex sets $\Sigma(A)$ and $\Sigma(B)$ are well-defined as long as $A$ and $B$ can be thickened to immersed regions $A_+$ and $B_+$ respectively. In the rest of this paragraph, we assume such thickenings exist, which applies to the cases shown in Fig. 2 as one can check pretty easily. The information convex set $\Sigma(A)$ contains a unique element if $A_+$ is an embedded disk; this fact has an elementary proof (Proposition 3.5 of [1]). The same line of logic does not work for $\Sigma(B)$. The very fact that $\Sigma(B)$ is nonempty has to be justified. It is still possible to show that $\Sigma(B)$ contains a unique element; the proof needs the merging technique; see Corollary 2.6.1. The idea is that we can obtain a state on $B$ by merging (Lemma 2.6, Theorem 2.7) reference states on a pair of (smaller) embedded disks.

### 2.1.2 Generalized isomorphism theorem

There is a huge number of possible choices of immersed regions. Are the structure of their information convex sets different? A rough answer is that these structures only depend on the topology of the region we choose, and therefore smooth deformation of the regions cannot change the structure.

The generalized isomorphism theorem (Theorem 2.5) provides a precise version of this statement. It is a direct generalization of the isomorphism theorem proved in [1]. This gen-

eralization is important for some of our applications. We first explain what we mean by *path* in this context. Consider a finite sequence of immersed regions $\{\Omega^t\}$, where the parameter $t = i/N$, where $i \in \{0, 1, \cdots, N\}$ and $N$ is a positive integer. We call the set $\{\Omega^t\}$ as a path from $\Omega^0$ to $\Omega^1$ if an adjacent pair of elements in $\{\Omega^t\}$ are related by adding/removing a small ball in a topologically trivial manner at the boundary of the region. (We say two adjacent elements in $\{\Omega^t\}$ are related by an elementary step, where the elementary step can either be an extension or restriction.)

**Theorem 2.5** (Generalized isomorphism theorem). *Let $\Omega^0$ and $\Omega^1$ be two immersed regions that are connected by a path. Their information convex sets are isomorphic,*

$$\Sigma(\Omega^0) \cong \Sigma(\Omega^1). \tag{2.8}$$

As in the original isomorphism theorem (Theorem 3.10 of Ref. [1]), the "$\cong$" in Eq. (2.15) is an isomorphism that preserves three things:

1. the structure as a convex set
2. the entropy difference of two elements
3. distance measures and the fidelity between any two elements

**Remark.** A few remarks are in order.

1. If we view the immersed region $\Omega^t$ through the pair $(\widehat{\Omega}, i_t)$ instead, the sequence of immersion maps $\{i_{i/N}\}_{i=1}^N$ then describes a discrete version of *regular homotopy* equivalence of $i_0$ and $i_1$. Intuitively, the generalized isomorphism theorem is a statement about regular homotopy equivalence, whereas the original isomorphism theorem (for subsystems) is about isotopy equivalence.

2. We omit the proof because it is similar to the proof of the isomorphism theorem. We emphasize that the key ingredient is the ability to construct an unknown density matrix by merging two quantum Markov states,[7] by applying the merging lemma:

   **Lemma 2.6** (Merging Lemma [18]). *If there is a pair of quantum states $\rho_{ABC}$ and $\lambda_{BCD}$ satisfy $\rho_{BC} = \lambda_{BC}$ and $I(A:C|B)_\rho = I(B:D|C)_\lambda = 0$, there exists a unique quantum state ("merged state") $\tau_{ABCD}$ such that*

$$\mathrm{Tr}_D \tau_{ABCD} = \rho_{ABC}\,,$$
$$\mathrm{Tr}_A \tau_{ABCD} = \lambda_{BCD}\,,$$
$$I(A:CD|B)_\tau = I(AB:D|C)_\tau = 0\,.$$

   Here $I(A:C|B)_\rho \equiv (S_{AB} + S_{BC} - S_B - S_{ABC})_\rho$ is the *conditional mutual information*.

   By the merging lemma, for each elementary step of extension, one can obtain a density matrix on the new region by merging the original density matrix and the reference state on a ball. Finally, it is important that the merging lemma can be promoted to the merging theorem, which guarantees that the resulting state is in an information convex set. For readers' convenience, we write down the full statement of the merging theorem at the end of this section; see Theorem 2.7.

**Corollary 2.6.1.** *Let $\omega$ be an immersed ball (in either 2d or 3d), then $\Sigma(\omega)$ contains a unique element.*

---

[7]We say a state $\rho$ is a quantum Markov state with respect to partition $A, B, C$ if the conditional mutual information $I(A:C|B)_\rho \equiv (S_{AB} + S_{BC} - S_B - S_{ABC})_\rho = 0$.

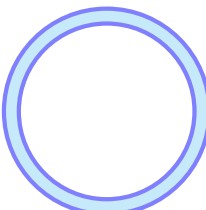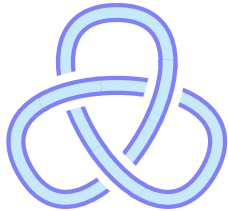

Figure 3: Two solid tori in 3d space, whose information convex sets are isomorphic. (Left) A solid unknot. (Right) A solid trefoil knot.

**Corollary 2.6.2.** *Let $T_1$ and $T_2$ be two solid tori embedded in a ball in 3d space. Their information convex sets are isomorphic, $\Sigma(T_1) \cong \Sigma(T_2)$.*

For example, $T_1$ can be an unknot and $T_2$ can be knotted. See Fig. 3.

The merging theorem below indicates that elements of the information convex sets are "closed" under the merging operation as long as a mild extra condition is satisfied.

**Theorem 2.7** (Merging Theorem [1]). [8] *Consider two density matrices $\rho_{ABC} \in \Sigma(ABC)$ and $\lambda_{BCD} \in \Sigma(BCD)$, such that $ABCD$ is an immersed region. $E$ is an immersed region that thickens $ABCD$. Consider three conditions:*

1. *$\rho_{BC} = \lambda_{BC}$ and $I(A:C|B)_\rho = I(B:D|C)_\lambda = 0$.*
2. *There exists a partition $B'C' = BC$, such that no bounded radius ball in $\widetilde{\Gamma}(E)$ overlaps with both $AB'$ and $CD$.*
3. *$I(A:C'|B')_\rho = I(B':D|C')_\lambda = 0$.*

*If these three conditions hold, the resulting density matrix generated by merging $\rho_{ABC}$ and $\lambda_{BCD}$ (by applying Lemma 2.6) belongs to $\Sigma(ABCD)$.*

**Remark.** In the context that the merging theorem applies, we shall often denote the merged state as $\tau = \rho \bowtie \lambda$. We emphasize that the extra conditions 2 and 3 are introduced to avoid pathological cases, which may happen when the regions are "too thin". For all the applications we consider in this paper, the regions are thick enough; thus, conditions 2 and 3 hold as long as condition 1 holds.

## 2.2 Structure theorems

Information convex sets are convex. What are the geometries of these sets? What are the entropy difference and distance measures between two elements of an information convex set? The (generalized) isomorphism theorem implies that the answer can only depend on the "topological class" of the immersed regions.

In this section, we review the structure theorems [1, 2], which provide a concrete answer to these questions. In particular, the information convex set for any sectorizable region [2] (see Definition 2.8 below) forms a simplex. For more general choices of immersed regions, the information convex set is the set of density matrices on a set of finite-dimensional Hilbert spaces, which we call fusion spaces. We describe a version of these theorems for immersed regions; these are immediate generalizations of the original version.

---

[8]Merging theorem is Proposition C.5 of [1], which is later printed as Theorem II.3 in [2], under the current name.

### 2.2.1 Simplex theorem and superselection sectors

Under what conditions is the information convex set of a region a simplex, with orthogonal extreme points? The simple and flexible notion of sectorizable regions [2] captures this simple condition. For sectorizable regions, an element of the information convex set carries only classical information, and the set of extreme points can be identified with a set of superselection sectors.

**Definition 2.8** (Sectorizable Region [2]). An immersed region $S$ is *sectorizable* if there is a region $\widehat{S}$ such that:

1. $\widehat{S}$ contains disjoint regions $S$ and $S''$.
2. Both $S$ and $S''$ can be deformed to $\widehat{S}$ by a path formed by extensions.

**Example 2.9.** Here are a few simple examples of sectorizable regions:

1. 2d regions: disk, annulus, the union of spatially-separated disks and annuli
2. 3d regions: ball, solid torus, sphere shell, torus shell (more in §3)

**Remark.** Every connected sectorizable region in the examples above has either one or two boundary components. This is a general fact; see Proposition 2.16. Furthermore, for every example above, $S = \mathcal{M} \times \mathbb{I}$, where $\mathbb{I}$ is an interval and $\mathcal{M}$ is a $(d-1)$-dimensional manifold, possibly with boundaries. We do not know if this feature is general; see Conjecture 2.18.

The simplex theorem for sectorizable region can be stated:

**Theorem 2.10** (Simplex theorem (Theorem 4.1 of Ref. [1])). *Let $S$ be a sectorizable region. Then $\Sigma(S)$ is a simplex, that is:*

$$\Sigma(S) = \left\{ \sum_I p_I \rho_S^I \,\middle|\, \sum_I p_I = 1, p_I \geq 0 \right\}, \tag{2.9}$$

*where $\{\rho_S^I\}$ is a set of mutually orthogonal density matrices.*

In the context of theorem 2.10, the set of labels $I$ forms a set $\mathcal{C}_S$, which we shall refer to as the set of superselection sectors.

In many contexts,[9] it is meaningful to talk about a special label, the vacuum sector, denoted as "1". If $S$ is a subsystem of a ball, we shall define the vacuum sector such that $\rho_S^1 = \sigma_S$ is the reduced density matrix of the reference state. (A nontrivial fact is that $\sigma_S$ is an extreme point no matter how complex the sectorizable subsystem is. See Lemma 3.2.)

**Definition 2.11** (Quantum dimension). Whenever the vacuum sector is well-defined, we define the quantum dimension of superselection sector $I \in \mathcal{C}_S$ as

$$d_I \equiv \exp\left( \frac{S\left(\rho_S^I\right) - S\left(\rho_S^1\right)}{2} \right). \tag{2.10}$$

For instance, a sphere shell in 3d is a sectorizable region. The extreme points correspond to the superselection sectors of the point excitations of the 3d topological order. We shall see a variety of 3d sectorizable regions in §3. For these examples, our definition is compatible with the idea that the quantum dimensions should be a positive eigenvector of the fusion multiplicities, as in 2d [1, 10].

---

[9]There are exceptions. For example, the simplex theorem can apply to an annulus surrounding a topological defect [22], even though none of the extreme points can be viewed as a vacuum.

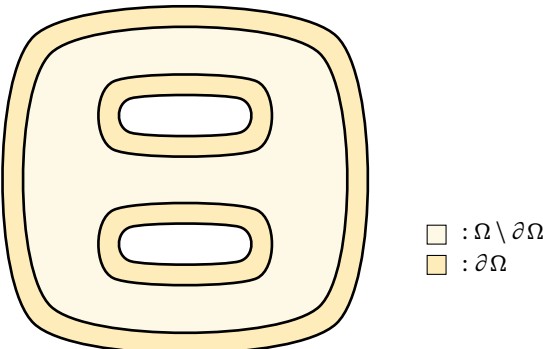

Figure 4: An illustration of $\Omega$ and $\partial\Omega$.

### 2.2.2 Hilbert space theorem and fusion spaces

For regions that are not sectorizable, to describe the structure of the information convex set, a set of finite dimensional Hilbert spaces is needed. These Hilbert spaces can be thought of as generalizations of the notion of fusion spaces in anyon theory. The Hilbert space theorem [1,2] is a concrete version of this statement.

Let $\Omega$ be an immersed region. Let $\Sigma(\Omega)$ be the information convex set associated with it. To state the Hilbert space theorem, we adopt the notion of *thickened boundary* of $\Omega$, denoted as $\partial\Omega$. $\partial\Omega$ is the subset of $\Omega$ that is obtained by thickening the boundary of $\Omega$ towards the interior of $\Omega$ by an enough distance;[10] see Fig. 4 for an illustration.

Below is a list of established facts about thickened boundary:

1. The thickened boundary $\partial\Omega$ is a sectorizable region because $\partial\Omega = \mathcal{M} \times \mathbb{I}$, where $\mathcal{M}$ is a $(d-1)$-dimensional closed manifold, and $\mathbb{I}$ is an interval.[11] Thus, $\Sigma(\partial\Omega)$ is a simplex. We can thus obtain a finite set of superselection sector labels $\mathcal{C}_{\partial\Omega}$.

2. If $\partial\Omega$ contains multiple connected components, then each component is sectorizable. Each label in $\mathcal{C}_{\partial\Omega}$ will be a collection of labels associated with the superselection sectors of each connected component of $\partial\Omega$. This is known as the "product rule" (Lemma IV.2 of [2]).

3. Every extreme point of $\Sigma(\Omega)$ reduces to an extreme point of $\Sigma(\partial\Omega)$ under partial trace. We review this fact in the proof of Lemma 2.13 below.

**Example 2.12.** For example, if $\partial\Omega$ has three connected components, as is shown in Fig. 4, then $I = \{a, b, c\}$ is an ordered triple. Here the three entries are the labels of superselection sector of each connected component.

These observations motivate the definition of the following convex subset of $\Sigma(\Omega)$. For any $I \in \mathcal{C}_{\partial\Omega}$, we define

$$\Sigma_I(\Omega) \equiv \left\{ \rho_\Omega \in \Sigma(\Omega) \,\middle|\, \mathrm{Tr}_{\Omega \backslash \partial\Omega} \rho_\Omega = \rho_{\partial\Omega}^I \right\}, \tag{2.11}$$

where $\rho_{\partial\Omega}^I$ is an extreme point of $\Sigma(\partial\Omega)$. Note that $\rho_\Omega \perp \lambda_\Omega$ if $\rho_\Omega \in \Sigma_I(\Omega)$ and $\lambda_\Omega \in \Sigma_J(\Omega)$, with $I \neq J$; this follows from the monotonicity of fidelity.

Thus $\Sigma(\Omega)$ is the convex hull of mutually orthogonal subsets $\{\Sigma_I(\Omega)\}_{I \in \mathcal{C}_{\partial\Omega}}$, namely

$$\Sigma(\Omega) = \left\{ \sum_{I \in \mathcal{C}_{\partial\Omega}} p_I \rho_\Omega^I \,\middle|\, \rho_\Omega^I \in \Sigma_I(\Omega), \sum_I p_I = 1, p_I \geq 0 \right\}. \tag{2.12}$$

---

[10]This is a few lattice spacings on the coarse-grained lattice.

[11]Whenever we write an immersed region as $\mathcal{M} \times \mathbb{I}$, we assume that the region can be further extended and restricted at both ends of the interval by a path.

There is a simple entropy condition that can unambiguously determine if an element is an extreme point of $\Sigma(\Omega)$:

**Lemma 2.13** (extreme point criterion)**.** *Let $\Omega$ be an immersed region. Let $\rho_\Omega \in \Sigma(\Omega)$. $\rho_\Omega$ is an extreme point if and only if*

$$(S_\Omega + S_{\Omega \setminus \partial\Omega} - S_{\partial\Omega})_\rho = 0\,. \tag{2.13}$$

*Here $\partial\Omega$ is the thickened boundary of $\Omega$.*

Note that the statement applies to the case that $\Omega$ is a closed manifold as well; in that case $\partial\Omega$ is empty.

*Proof.* First, if $\rho_\Omega$ is an extreme point of $\Sigma(\Omega)$ then Eq. (2.13) holds. This is known as the "factorization property" of the extreme points (Appendix C of Ref. [2]). Second, to see that Eq. (2.13) and $\rho_\Omega \in \Sigma(\Omega)$ implies that $\rho_\Omega$ is an extreme point, we consider proof of contradiction.

1. Let us observe that $\text{Tr}_{\Omega \setminus \partial\Omega} \rho_\Omega \in \Sigma(\partial\Omega)$ and it must be an extreme point. If not, it must be a mixture of extreme points. Divide $\partial\Omega$ into three layers (outer, middle, and inner) of increasing distance to the boundary of $\Omega$. Then there will be a nontrivial correlation between the inner layer and the outer layer on the superselection sectors. This is in contradiction with Eq. (2.13), which implies that the mutual information between the inner layer and outer layer vanishes. It follows that $\rho_\Omega \in \Sigma_I(\Omega)$, for some label $I \in \mathcal{C}_{\partial\Omega}$.

2. Any $\rho_\Omega \in \Sigma_I(\Omega)$ satisfying Eq. (2.13) must be an extreme point. This is because $(S_\Omega + S_{\Omega \setminus \partial\Omega} - S_{\partial\Omega})_\rho = 2(S_\Omega(\rho) - S_\Omega(\rho^{I,\langle e \rangle}))$ for any $\rho_\Omega \in \Sigma_I(\Omega)$ and $\rho_\Omega^{I,\langle e \rangle}$ is an extreme point of $\Sigma_I(\Omega)$. The right-hand side is positive for nonextreme points.

This completes the proof. $\qquad\square$

As a simple corollary of the extreme point criterion and Eq. (2.10), we have

**Corollary 2.13.1.** *When the definition of quantum dimension (Eq. (2.10)) is applicable, for each $I \in \mathcal{C}_{\partial\Omega}$ that corresponds to a nonempty $\Sigma_I(\Omega)$,*

$$\ln d_I = S\left(\rho_\Omega^{I,\langle e \rangle}\right) - S\left(\rho_\Omega^{1,\langle e \rangle}\right)\,, \tag{2.14}$$

*where $\rho_\Omega^{I,\langle e \rangle}$ is an extreme point of $\Sigma_I(\Omega)$ and 1 is the vacuum label.*

The proof is omitted because it is an analog of the proof of Lemma 4.8 of [1]. This result shows that we do not need to extend the definition of quantum dimension to non-sectorizable regions. Now we are in the position to state the Hilbert space theorem, which describes the structure of $\Sigma_I(\Omega)$.

**Theorem 2.14** (Hilbert space theorem [1,2]). [12] *For an immersed region $\Omega$,*

$$\Sigma_I(\Omega) \cong \mathcal{S}(\mathbb{V}_I)\,, \tag{2.15}$$

*where $\mathcal{S}(\mathbb{V}_I)$ is the state space[13] of a finite dimensional Hilbert space $\mathbb{V}_I$.*

Therefore, we can completely characterize the convex set $\Sigma_I(\Omega)$ by a non-negative integer $N_I = \dim \mathbb{V}_I$. We shall refer to this integer as a *(fusion) multiplicity*.

---

[12]The original proof (Theorem 4.5 of Ref. [1]) was stated for $\Omega$ being a 2-hole disk. Theorem D.5 of Ref. [2] was stated for a general embedded region $\Omega$. (The proof there works for immersed regions without change.)

[13]State space of Hilbert space $\mathcal{H}$ is the set of all density matrices on $\mathcal{H}$.

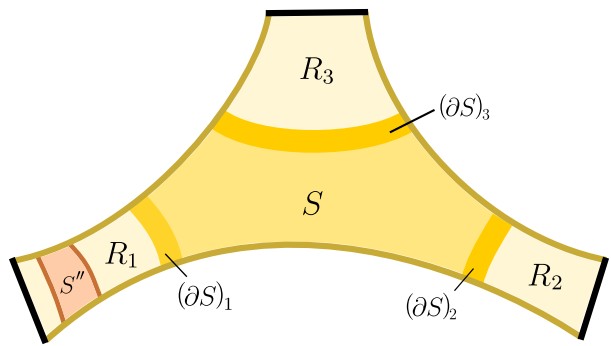

Figure 5: A schematic illustration of the contradiction, supposing that the thickened boundary of sectorizable region $S$ has three connected components. (The black lines are the boundary of $\hat{S}$.)

**Example 2.15.** When $N_I = 0$, $\Sigma_I(\Omega)$ is empty. When $N_I = 1$, $\Sigma_I(\Omega)$ contains a unique element; this element is an *isolated* extreme point of $\Sigma(\Omega)$ because no extreme point is close to it in terms of distance measures. When $N_I = 2$, $\Sigma_I(\Omega)$ is isomorphic to a Bloch ball. It contains (infinite number of) continuously parameterized extreme points.

The region $\Omega$ can store a piece of quantum information when $N_I > 1$. Quantum information cannot be copied in the sense that if one tears apart $\Omega$, the quantum information can be recovered from at most one party; the storage of this quantum information is nonlocal in the sense that one cannot decode this information from ball-shaped regions, nor from $\partial\Omega$. We shall come back to this point when discussing the storage of quantum information in knot complements §3.4.

### 2.2.3 General properties of sectorizable regions

We discuss a few general properties of sectorizable regions.

**Proposition 2.16.** *The thickened boundary of any connected sectorizable region has either one or two connected components.*

*Proof.* We shall denote the sectorizable region as $S$ and denote its thickened boundary as $\partial S$. First, we use the defining properties (Definition 2.8). Because $S$ and $S''$ can be deformed to $\hat{S}$ by a sequence of elementary steps and $S$ is connected, $S''$ and $\hat{S}$ must be connected as well.

Below, we proceed with a proof by contradiction. We shall assume that the thickened boundary of $S$, denoted as $\partial S$, has three connected components: $(\partial S)_1, (\partial S)_2$ and $(\partial S)_3$. (The proof generalizes straightforwardly to the case that the number of connected components of $\partial S$ is larger.)

The sequence of extensions that deforms $S$ to $\hat{S}$ is, in fact, a sequence of extensions of $(\partial S)_1, (\partial S)_2$ and $(\partial S)_3$ respectively. Denote the regions obtained after the extensions as $R_1, R_2$ and $R_3$, where $R_i \supset (\partial S)_i$. They must be spatially separated subsets of $\hat{S}$. Furthermore, $\partial\hat{S}$ must have precisely three connected components, denoted as $\{(\partial\hat{S})_i\}_{i=1}^3$, such that $(\partial\hat{S})_i \subset R_i$.

Since $S''$ is connected, it must be contained in one of $R_i$; without loss of generality, we write $S'' \subset R_1$. Hence $S''$ can be extended to include $(\partial\hat{S})_1$ without overlapping with $S$. However, to extend $S''$ to include $(\partial\hat{S})_2$ or $(\partial\hat{S})_3$, the extension must overlap with $S$. (In comparison, we saw that $S$ can be extended to include $(\partial\hat{S})_2$ and $(\partial\hat{S})_3$ without overlapping with $S''$.)

However, a parallel line of reasoning, switching the role of $S$ and $S''$, implies that it is possible to extend $S$ to only one of the three connected components of $(\partial\hat{S})$ without overlapping with $S''$. This is a contradiction, and this completes the proof. $\qquad\square$

**Conjecture 2.17.** *If the thickened boundary $\partial S$ of a connected sectorizable region $S$ has two connected components $(\partial S)_1$ and $(\partial S)_2$, then $\Sigma(S) \cong \Sigma((\partial S)_1) \cong \Sigma((\partial S)_2)$, where the isomorphism between the two boundary components is induced by partial trace.*

This conjecture is true for all sectorizable regions of which we are aware, but we do not have general proof. In fact, the following stronger conjecture holds for all sectorizable regions of which we are aware. (The stronger conjecture implies the previous one.)

**Conjecture 2.18.** *A connected sectorizable region $S$ can be written as $S = \mathcal{M} \times \mathbb{I}$, where $\mathcal{M}$ is a manifold and $\mathbb{I}$ is an interval. Furthermore, if $S$ has one boundary component, $\mathcal{M}$ has boundaries; if $S$ has two boundary components, $\mathcal{M}$ is closed.*

**Remark.** We emphasize that the conjecture may apply to a broad context. For example, we expect that a system with a gapped boundary is not a counterexample if a proper notion of topology is adopted for regions adjacent to a gapped boundary.

Consider a connected sectorizable region of the form $S = \mathcal{M} \times \mathbb{I}$. It is not difficult to see that if $\mathcal{M}$ has boundaries, then $S$ has one boundary component; if $\mathcal{M}$ is closed, then $S$ has two boundary components.

For this type of sectorizable region, starting from an extreme point $\rho_S^I \in \text{ext}(\Sigma(S))$, trace out the density matrix in the interior of $S$. The reduced density matrix on the thickened boundary $\partial S$ is an extreme point of $\Sigma(\partial S)$, whose label is completely determined by extreme point label $I \in \mathcal{C}_S$ (Proposition D.4 of Ref. [2]). Quantum dimension $d_I$ for $I \in \mathcal{C}_S$ is related to the quantum dimension of the extreme point on the thickened boundary $\partial S$ in the following way.

**Proposition 2.19.** *Consider a connected d-dimensional immersed region $S = \mathcal{M} \times \mathbb{I}$ where $\mathcal{M}$ is a $(d-1)$-dimensional manifold and $\mathbb{I}$ is an interval. Assume that $\Sigma(S)$ admits a special extreme point, i.e., the vacuum denoted as $\sigma_S$. Let $I \in \mathcal{C}_S$. If $\mathcal{M}$ has boundaries, $\partial S$ only has one connected component. We can denote $h(I)$ as the sector label for the reduced density matrix of $\rho_S^I$ on $\partial S$, and*

$$d_{h(I)} = d_I^2 . \tag{2.16}$$

*If $\mathcal{M}$ is closed, $\partial S$ has two connected components $(\partial S)_1, (\partial S)_2$. Similarly, $h_1(I), h_2(I)$ are denoted as sector labels for the reduced density matrix of $\rho_S^I$ on $(\partial S)_1, (\partial S)_2$ respectively, and*

$$d_{h_1(I)} = d_{h_2(I)} = d_I . \tag{2.17}$$

*Proof.* If $\mathcal{M}$ has boundaries, $\partial S$ only has one connected component. We get

$$\begin{aligned}
S(\rho_{\partial S}^{h(I)}) - S(\sigma_{\partial S}) &= \left[ S(\rho_S^I) + S(\rho_{S\setminus\partial S}^I) \right] - \left[ S(\sigma_S) + S(\sigma_{S\setminus\partial S}) \right] \\
&= 2 \left( S(\rho_S^I) - S(\sigma_S) \right) .
\end{aligned} \tag{2.18}$$

The first equality follows from the extreme point criterion (Lemma 2.13). The second equality follows from generalized isomorphism theorem. Then Eq. (2.16) follows from the definition of quantum dimension (Definition 2.11).

If $\mathcal{M}$ is closed, then $\partial S$ has two connected components $(\partial S)_1, (\partial S)_2$. From the tensor product structure for the density matrix supported on two disjoint regions $(\partial S)_1, (\partial S)_2$ ("product rule" shown in Lemma IV.2 of [2]), we have $(S_{(\partial S)_1} + S_{(\partial S)_2} - S_{\partial S})_{\rho_{\partial S}^{(h_1(I), h_2(I))}} = 0$. Then

$$\left[ S\left(\rho_{(\partial S)_1}^{h_1(I)}\right) - S\left(\sigma_{(\partial S)_1}\right) \right] + \left[ S\left(\rho_{(\partial S)_2}^{h_2(I)}\right) - S\left(\sigma_{(\partial S)_2}\right) \right] = S\left(\rho_{\partial S}^{(h_1(I), h_2(I))}\right) - S(\sigma_{\partial S}) . \tag{2.19}$$

Since $(\partial S)_1$ and $(\partial S)_2$ are connected by a path within $S$, from generalized isomorphism theorem, entropy difference is conserved, i.e. $S(\rho_{(\partial S)_1}^{h_1(I)}) - S(\sigma_{(\partial S)_1}) = S(\rho_{(\partial S)_2}^{h_2(I)}) - S(\sigma_{(\partial S)_2})$. Similar to the proof for Eq. (2.18), the righthand side of Eq. (2.19) equals to $2(S(\rho_S^I) - S(\sigma_S))$. Then Eq. (2.17) also follows from the definition of quantum dimension. $\square$

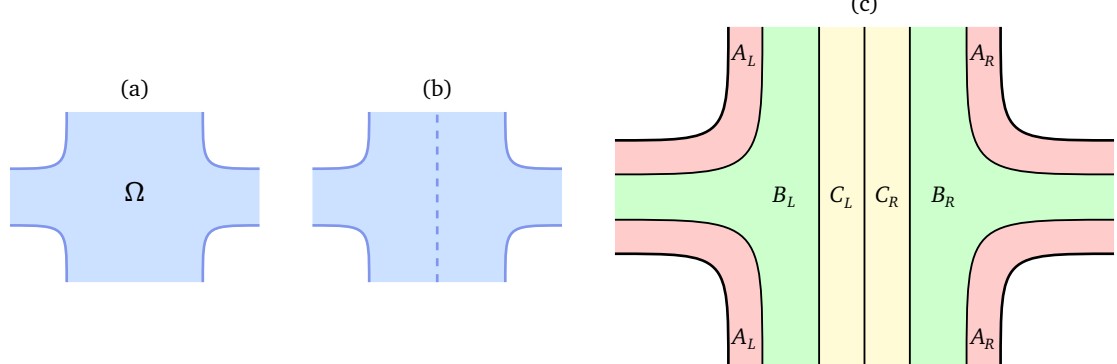

Figure 6: (a) A possibly immersed region $\Omega$, part of which is shown. (b) It is divided into halves by a hypersurface (dashed line). (c) Partition of $\Omega$ into $A_L B_L C_L C_R B_R A_R$. Here $C = C_L C_R$ is the thickening of the hypersurface, where $C_L$ and $C_R$ lie on opposite sides. $A_L A_R = \partial \Omega$. $\Omega_L = A_L B_L C$ and $\Omega_R = C B_R A_R$. Note that $\Omega = \Omega_L \cup \Omega_R$ and $\Omega_L \cap \Omega_R = C$.

## 2.3 Associativity theorem

The dimensions of the Hilbert spaces obey further consistency relations. One important such relation is known as associativity. In the anyon theory (Proposition 4.11 of Ref. [10]) the associativity relation relates the fusion multiplicities of the two-hole disk to those of the three-hole disk: $\sum_i N_{ab}^i N_{ic}^d = N_{abc}^d$. Similar relations also appear in broader physical contexts, e.g., in the presence of a gapped domain wall and in higher-dimensional systems. In particular, there are many associativity relations for 3d systems. Entanglement bootstrap is capable of deriving associativity relations. However, previous methods require a case-by-case analysis.

In this section, we present an associativity theorem (Theorem 2.22). The associativity relations for a large variety of cases can then be read off effortlessly as corollaries. The theorem applies to subsystems as well as immersed regions. It works in 2d and 3d as well as higher dimensions. The idea of the proof is to cut a region into pieces and analyze the ability to merge the pieces back.

Let us first state the general setup. Consider a region $\Omega$, divided into two parts by a hypersurface. (See Fig. 6 for an illustration. We emphasize that the consideration is general.) The hypersurface may have more than one connected component but must be disjoint from the boundary of $\Omega$. We shall consider a partition of $\Omega$ into $A_L B_L C_L C_R B_R A_R$. Here $C = C_L C_R$ is the thickening of the hypersurface, where $C_L$ and $C_R$ lie on the opposite sides. $A_L A_R = \partial \Omega$. $\Omega_L = A_L B_L C$ and $\Omega_R = C B_R A_R$.

Because $A_L, C, A_R$ are sectorizable regions, we can talk about superselection sectors on them. Let $\{a_L, \cdots\}, \{i, \cdots\}, \{a_R, \cdots\}$ be the labels of the extreme points of $\Sigma(A_L), \Sigma(C), \Sigma(A_R)$ respectively. Let $N_{a_L}^i(\Omega_L)$ and $N_i^{a_R}(\Omega_R)$ be the dimensions of fusion spaces associated with $\Sigma_{a_L}^i(\Omega_L)$ and $\Sigma_i^{a_R}(\Omega_R)$, and let $N_{a_L}^{a_R}(\Omega)$ be the dimension of the fusion space associated with $\Sigma_{a_L}^{a_R}(\Omega)$.

**Lemma 2.20.** *Suppose there is a pair of extreme points $\rho_{\Omega_L} \in \Sigma(\Omega_L)$ and $\lambda_{\Omega_R} \in \Sigma(\Omega_R)$, that are consistent on $C$. Then the following two statements hold:*

*1. $\rho_{\Omega_L}$ and $\lambda_{\Omega_R}$ can be merged.*

*2. The result of merging is an extreme point of $\Sigma(\Omega)$.*

**Remark.** Importantly, for generality, we allowed the hypersurface to be a union of connected components. Furthermore, we note that $A_L$ and $A_R$ can be empty sets.

The proof of this lemma is presented in Appendix B. The nontrivial part is to show that the merged state satisfies the extreme point criterion (Lemma 2.13).

**Lemma 2.21.** *If $\Sigma_{a_L}^{a_R}(\Omega)$ is nonempty, its maximum-entropy state satisfies*

$$I(A_L B_L : B_R A_R | C) = 0. \tag{2.20}$$

*Proof.* Suppose this were not the case. To see the contradiction, we reduce the given state to $\Omega_L$ and $\Omega_R$ then merge the marginals back. This merging is always possible and the newly obtained state is an element of $\Sigma_{a_L}^{a_R}(\Omega)$ that satisfies $I(A_L B_L : B_R A_R | C) = 0$. However, the newly obtained state has a larger entropy; this is because among states with identical marginals, the one with minimal conditional mutual information has the greatest entropy. This completes the proof. $\square$

**Theorem 2.22** (Associativity theorem). *In the general setup concerning immersed region $\Omega$, its partition and the labeling (see Fig. 6), the following associativity condition holds:*

$$N_{a_L}^{a_R}(\Omega) = \sum_{i \in \mathcal{C}_C} N_{a_L}^i(\Omega_L) N_i^{a_R}(\Omega_R). \tag{2.21}$$

The proof of is presented in Appendix B. Intuitively, the ability of deriving this theorem lies in the fact that subregions $\Omega_L$ and $\Omega_R$ know enough about both the extreme points (by Lemma 2.20) and the maximum-entropy state of $\Sigma_{a_L}^{a_R}(\Omega)$ (by Lemma 2.21).

**Proposition 2.23** (sectorizable region and restriction). *Let $S$ be a sectorizable region, and $\rho_S$ be an extreme point of $\Sigma(S)$. Let $\Omega$ be a region embedded in $S$. Then the following statements are true:*

1. *The reduced density matrix $\rho_\Omega \equiv \mathrm{Tr}_{S \backslash \Omega} \rho_S$ is an extreme point of $\Sigma(\Omega)$.*
2. *$\rho_\Omega$ is an isolated extreme point: Let $I \in \mathcal{C}_{\partial \Omega}$ be the label such that $\rho_\Omega \in \Sigma^I(\Omega)$, the associated fusion space dimension $N^I(\Omega) = 1$.*

*Proof.* We first show that the two statements hold if $\Omega$ is embedded in $S$ in such a way that $\Omega \cap \partial S = \emptyset$. Let $S = \Omega \cup \Omega'$, where $\Omega' \equiv \partial \Omega \cup (S \backslash \Omega)$. Translating notations in Associativity Theorem 2.22 to this context, we have $\Omega_L = \Omega$, $\Omega_R = \Omega'$, $A_L = \emptyset, A_R = \partial S$, and $C = \partial \Omega$. Because $\rho_S$ is an extreme point, it carries a *particular* label $s \in \mathcal{C}_{\partial S}$. Rewrite Eq. (2.21) for this particular label, we find:

$$1 = N^s(S) = \sum_{I \in \mathcal{C}_{\partial \Omega}} N^I(\Omega) N_I^s(\Omega'). \tag{2.22}$$

Here $N^s(S) = 1$ because $S$ is sectorizable. If $\rho_\Omega$ were not an extreme point of $\Sigma(\Omega)$, then there can only be two cases:

1. $\rho_{\partial \Omega} \equiv \mathrm{Tr}_{S \backslash \partial \Omega} \rho_S$ is a convex combination of more than one extreme point of $\Sigma(\partial \Omega)$. Let us say $\rho_{\partial \Omega} = p_1 \rho_{\partial \Omega}^{I_1} + p_2 \rho_{\partial \Omega}^{I_2} + \cdots$, where $p_1, p_2 > 0$. Then $N^{I_1}(\Omega)$, $N_{I_1}^s(\Omega')$, $N^{I_2}(\Omega)$ and $N_{I_2}^s(\Omega')$ are greater or equal to 1. This violates Eq. (2.22).
2. There is a certain label $I \in \mathcal{C}_{\partial \Omega}$, such that $\rho_\Omega \in \Sigma_I(\Omega)$ but $\rho_\Omega \notin \mathrm{ext}(\Sigma_I(\Omega))$. This implies $N^I(\Omega) \geq 2, N_I^s \geq 1$. Again, this violates Eq. (2.22).

This proves the case that $\Omega \cap \partial S = \emptyset$.

Suppose $\Omega \cap \partial S$ is nonempty, which happens when $\Omega$ share boundaries with $S$, for instance. It is possible to shrink $\Omega$ along its boundary and obtain $\Omega_-$ such that $\Omega_- = \Omega \backslash \partial \Omega$ and $\Omega_- \cap \partial S = \emptyset$. As the previous paragraph shows, the two statements of the proposition hold for $\Omega_-$. We then extend $\Omega_-$ back to $\Omega$ using a sequence of elementary steps of extensions. Both statement 1 and 2 still hold under each such step, and therefore they hold for the region $\Omega \subset S$. This completes the proof of the general case. $\square$

Table 1: Summary of the content of §3.

| Section | Physical data | Choice of regions |
|---|---|---|
| §3.1 | vacuum | ball, 3-sphere |
| §3.2 | superselection sectors | sectorizable regions |
| | point particles $\mathcal{C}_{\text{point}}$ | sphere shell |
| | pure-fluxes $\mathcal{C}_{\text{flux}}$ | solid torus |
| | Hopf excitations $\mathcal{C}_{\text{Hopf}}$ | torus shell |
| | shrinkable loops $\mathcal{C}_{\text{loop}}$ | torus shell with a constraint |
| §3.3 | fusion spaces (basic ones) | 3d regions with boundaries |
| | dimensional reduction | see Appendix E for details |
| §3.4 | knot multiplicity | knot complement |
| §3.5 | exotic fusion of fluxes | 3d regions with boundaries |

## 2.4 Subdivision-invariance and minimal diagram: tools for explicit data

A minimal diagram [17] is a means to avoid unnecessary complication when studying information convex sets for ground states of exactly solvable models of topological order. The idea is that the information convex set is a topological invariant in two senses: it depends only on the phase of matter represented by the reference state, and it depends only on the topology of the spatial region. Therefore we may choose the reference state to be a renormalization group fixed-point state which is not changed by subdivisions of the cell complex on which the model is defined. More concretely, starting from a fixed-point wave function, we can use entanglement renormalization [23, 24] to remove as many degrees of freedom as possible while preserving the topology of the region of interest. Therefore, we can determine its structure by considering the simplest available cell complex that discretizes the region of interest.

# 3 Information convex sets for various 3d topologies

In this section, we study the information convex set for regions of various 3d topologies, applying the entanglement bootstrap techniques summarized in §2. This allows us to identify a number of simplexes and finite-dimensional Hilbert spaces. Physically, these correspond to the superselection sectors of various excitation types of a 3d topological order and their fusion spaces. Some explicit data calculated from 3d quantum double models are presented for illustration purposes, the detailed calculation of which are shown in §4. The content of each subsection is summarized in Table 1.

## 3.1 Ball, 3-sphere, and the vacuum

In the entanglement bootstrap approach, we often start with a reference state on a ball instead of that on a closed manifold. The reason is that the density matrix on a ball is often an "economical" starting point: a ball is a subsystem of any manifold, and therefore it is something easy to obtain. Starting with a reference state on other manifolds is possible, but that is considered as a stronger input.

Nonetheless, the sphere completion lemma (Lemma 3.1) below indicates that the sphere is equally simple, in the sense that one can always complete the ball to a 3-sphere. The logic of the analysis applies in any space dimension.

**Lemma 3.1** (sphere completion lemma)**.** *Let $\sigma_Z$ be a reference state of a ball $Z$. Let $Z = ABCD$ as is shown in Fig. 7(a). There exists a pure state $|\psi\rangle$ on a 3-sphere $S^3 = \widetilde{A}BCD$ (Fig. 7(b)) such*

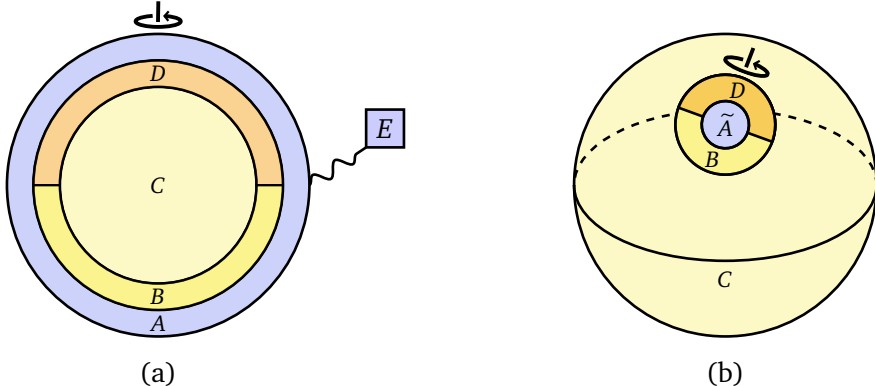

Figure 7: Sphere completion for 3d: completion of the reference state on a ball to a 3-sphere $S^3$. (a) A reference state on a ball $Z = ABCD$; $E$ is a finite-dimensional purifying system. (b) We map the system to a 3-sphere $S^3 = \widetilde{A}BCD$ by treating the union of $A$ and $E$ as a single site $\widetilde{A}$.

*that:*

1. *$|\psi\rangle$ is a reference state of $S^3$ if we treat $\widetilde{A}$ as a single site on the coarse-grained lattice. (Namely, the Hilbert space on each site is finite dimensional and that the 3d version of axioms **A0** and **A1** are satisfied on $|\psi\rangle$.)*

2. *$\mathrm{Tr}_{\widetilde{A}} |\psi\rangle\langle\psi| = \sigma_{BCD}$.*

*We shall call $|\psi\rangle$ as the completion of reference state $\sigma_Z$ onto the 3-sphere.*

*Proof.* Consider a partition of the ball $Z = ABCD$ as is shown in Fig. 7(a). Let $E$ be its purifying system with a finite dimensional Hilbert space, and $|\psi\rangle$ be a purification of $\sigma_Z$. We map $AE$ to a single site $\widetilde{A}$. The topology of the coarse-grained lattice is now a 3-sphere. It is easy to check that the state $|\psi\rangle$ satisfies the axioms that we would expect for a reference state on a 3-sphere. In particular, we have

$$\Delta(BD, \widetilde{A})_{|\psi\rangle} = 0 \quad \text{and} \quad \Delta(B, \widetilde{A}, D)_{|\psi\rangle} = 0, \tag{3.1}$$

among other relations. This completes the proof. $\square$

The important physical object associated to the ball and 3-sphere is the vacuum. To prepare the discussion of the vacuum, we recall the basic facts about information convex sets on these simple regions:

- The information convex set of a disk in 2d (ball in 3d) contains a unique element. The 2d case is Proposition 3.5 of [1], and the proof of the 3d case is a straightforward generalization.

- The information convex set on a sphere in 2d (3-sphere in 3d) contains a unique element, and that element is a pure state. The 2d case is Proposition 3.7 of [1], and the proof of the 3d case is a straightforward generalization.

The following lemma is the key to putting many of the anticipated properties of the vacuum into a firm footing. The logic of the proof works for general space dimensions.

**Lemma 3.2** (vacuum lemma)**.** *Let $\sigma$ be the reference state defined on a ball ( 3-sphere). $\Omega$ is a region embedded in the ball (3-sphere). Then the following are true:*

1. $\sigma_\Omega \in ext(\Sigma(\Omega))$.

2. Let $I \in \mathcal{C}_{\partial\Omega}$ be the label such that $\sigma_\Omega \in \Sigma_I(\Omega)$. The associated fusion space dimension satisfies $N_I(\Omega) = 1$.

*Proof.* First, we prove the case of a ball. A ball is a sectorizable region; moreover, the reference state is an extreme point. By applying Proposition 2.23, we prove the desired answer. Second, we prove the case of a 3-sphere. If $\Omega = S^3$, the result is true. If $\Omega \subsetneq S^3$, then $\Omega$ is embedded in a ball-shaped subsystem of $S^3$. By the previous argument, we prove the desired answer. $\square$

One implication is that vacuum is a well-defined superselection sector. Another implication is that vacuum fuse trivially among each other, as long as we consider fusion spaces associated with regions embedded in a ball (or a 3-sphere).

**Definition 3.3** (vacuum sector). Let $S$ be a sectorizable region embedded in a ball (3-sphere) for which the reference state $\sigma$ is defined. We define the vacuum sector 1 to be the label of the following unique extreme point:

$$\rho_S^1 \equiv \sigma_S. \tag{3.2}$$

In the rest of Section 3, all regions are embedded/immersed in either a ball or a 3-sphere, whichever is more convenient.

## 3.2 Sectorizable regions and superselection sectors

By the simplex theorem (Theorem 2.10), we can assign a set of superselection sectors to each sectorizable region. In 3d, sectorizable regions are diverse. Simple choices include sphere shell, solid torus, and torus shell; see Fig. 8. (The solid torus and torus shells can be knotted.) Another class of sectorizable regions is the genus-$g$ handlebodies for each $g > 1$; these play an important role in our companion study of braiding [20].

The superselection sectors associated with each region in Fig. 8 are summarized below. These superselection sectors are associated with different classes of point (loop) excitations. Each class of point (loop) excitations can be created by applying a suitable type of string (membrane) operator.[14] Intuitively, the view of the information convex set is "dual" to the view of excitations because it focuses on the complement of the excitations (when viewed on a 3-sphere). Each region in Fig. 8 is embedded in a ball (or 3-sphere), and therefore the vacuum sector is well-defined.

1. (Sphere shell) Let $X$ be a sphere shell. $\Sigma(X)$ is a simplex with extreme points $\mathrm{ext}(\Sigma(X)) \equiv \{\rho_X^a\}_{a \in \mathcal{C}_{\mathrm{point}}}$, where

$$\mathcal{C}_{\mathrm{point}} = \{1, a, b, \cdots\}. \tag{3.3}$$

These are the labels for the point particles, among which 1 is the vacuum sector.

For each $a \in \mathcal{C}_{\mathrm{point}}$, there is a unique antiparticle $\bar{a} \in \mathcal{C}_{\mathrm{point}}$. Furthermore, $\bar{1} = 1$, $\bar{\bar{a}} = a$ and $d_a = d_{\bar{a}} \geq 1$. Point particles can be created using a string operator connecting $a$ and $\bar{a}$. These are explained in Appendix E.

The *total quantum dimension* is defined as

$$\mathcal{D} \equiv \sqrt{\sum_{a \in \mathcal{C}_{\mathrm{point}}} d_a^2}. \tag{3.4}$$

---

[14]String and membrane operators have been considered extensively for exactly solvable models; see, for example, [4,25,26]. The existence of such operators can be established independently in entanglement bootstrap with the logic presented in Appendix H of [1]. The idea is to use the structure theorems of the information convex set and Uhlmann's theorem to constrain the general shape of the deformable operator.

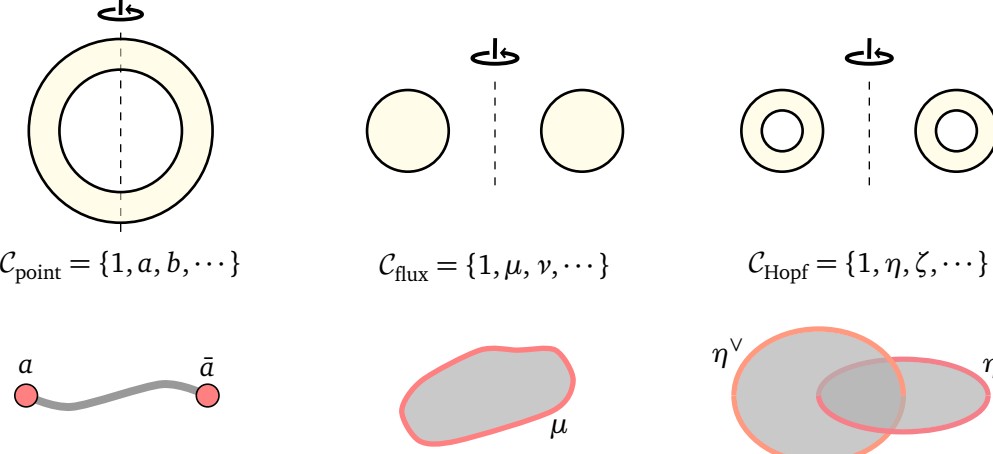

Figure 8: A list of basic sectorizable regions in 3d, the corresponding superselection sector types, and the string/membrane operators. Each region is embedded in a ball. (Left) sphere shell, the superselection sectors of point particles, and a string operator. (Middle) solid torus, the superselection sectors of pure-fluxes, and a membrane operator supported on a disk. (Right) Torus shell, the Hopf excitations, and a membrane operator bounded by a Hopf link. $\eta$ is a label of an extreme point of the information convex set of a torus shell in a neighborhood of one of the loops; $\eta^\vee$ is the label on a torus shell thickening the other. The relation between the loop labels $\eta^\vee$ and $\eta$ is analogous to the relation between particle and antiparticle, as explained in Footnote 17.

2. (Solid torus) Let $T$ be a solid torus. $\Sigma(T)$ is a simplex with extreme points $\{\rho_T^\mu\}_{\mu \in \mathcal{C}_{\text{flux}}}$, where

$$\mathcal{C}_{\text{flux}} = \{1, \mu, \nu, \cdots\}. \tag{3.5}$$

These are the labels for the pure-flux sectors, among which 1 is the vacuum sector. They correspond to excitations on closed loops that can be created by a single membrane operator supported on a disk; see Fig. 8.

For each $\mu \in \mathcal{C}_{\text{flux}}$ there is an anti-flux which we shall denote as $\bar{\mu} \in \mathcal{C}_{\text{flux}}$. Here are three ways to think about anti-fluxes:

- Rotating the flux excitation by $\pi$, about an in-plane axis. This maps the loop excitation back to the same position, and it maps $\mu$ to $\bar{\mu}$. Fig. 9(a).
- Deform the solid torus $T$, such that it rotates by $\pi$, about an in-plane axis, and maps back to itself. This generates an automorphism of the information convex set, such that $\rho_T^\mu \to \rho_T^{\bar{\mu}}$. Fig. 9(b).
- Consider an element in the information convex set of a genus-2 handlebody. If, after a partial trace, it reduces to 1 ($\mu$) on the outer (left) solid torus shown in Fig. 9(c), then it must reach the extreme point labeled by $\bar{\mu}$ on the third solid torus.[15]

Furthermore,

$$\bar{\bar{\mu}} = \mu, \quad d_{\bar{\mu}} = d_\mu \geq 1, \quad \forall \, \mu \in \mathcal{C}_{\text{flux}}. \tag{3.6}$$

We explain these conditions and say more about anti-fluxes when discussing dimensional reductions in Appendix E. The total quantum dimension, defined in Eq. (3.4), can be alternatively expressed as $\mathcal{D} = \sqrt{\sum_{\mu \in \mathcal{C}_{\text{flux}}} d_\mu^2}$; see Proposition 3.4.

---

[15]To make this mathematically accurate, we have fixed the sector label of the blue solid torus on the right by a "translation" of the labels of the left blue solid torus.

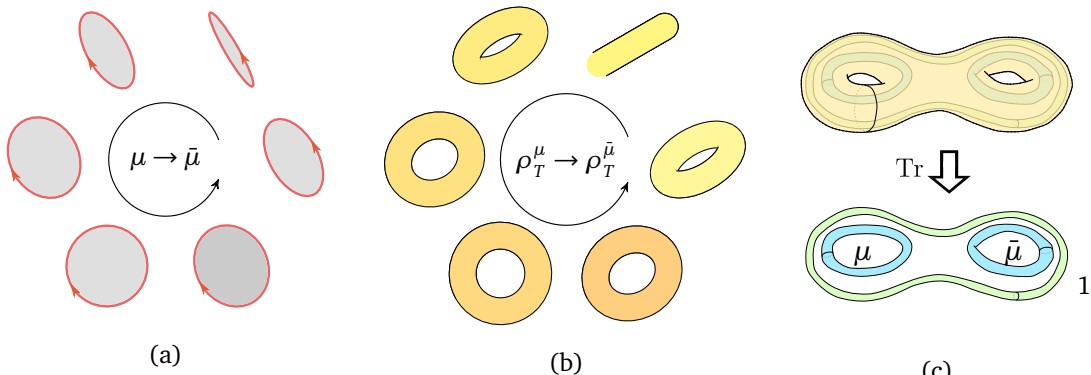

(a)           (b)           (c)

Figure 9: Three ways to think about anti-fluxes: (a) deform the excitation, (b) automorphism of the information convex set of the solid torus, and (c) do a partial trace for a genus-2 handlebody.

3. (Torus shell) Let $\mathbb{T}$ be a torus shell. $\Sigma(\mathbb{T})$ is a simplex with extreme points $\{\rho_{\mathbb{T}}^{\eta}\}_{\eta \in \mathcal{C}_{\text{Hopf}}}$, where

$$\mathcal{C}_{\text{Hopf}} = \{1, \eta, \zeta, \cdots\}. \tag{3.7}$$

We call them "Hopf sectors" (or "Hopf excitations") because all such superselection sectors can be realized by a loop that participates in a linked loop pair on a 3-sphere such that the pair of loop excitations form a Hopf link. See Fig. 8(c) for an illustration.[16] We emphasize that a Hopf superselection sector is associated with *a single* loop excitation (possibly linked with other excitations) in the sense that the sector can be detected from the tubular neighborhood of the loop. (This feature implies, for example, that a Hopf sector can be assigned to any single loop that is part of three linked Hopf fibers on $S^3$.)

As we shall show in Proposition 3.5, the set of Hopf excitations has a natural decomposition into a union of disjoint subsets (see Fig. 11 for the procedure):

$$\mathcal{C}_{\text{Hopf}} = \bigcup_{\mu \in \mathcal{C}_{\text{flux}}} \mathcal{C}_{\text{Hopf}}^{[\mu]}. \tag{3.8}$$

We shall develop a dimensional reduction understanding of $\mathcal{C}_{\text{Hopf}}^{[\mu]}$ in Appendix E; therein, each set is mapped to the set of anyons in a 2d entanglement bootstrap problem.

The information convex set $\Sigma(\mathbb{T})$ has nontrivial automorphisms. One obvious automorphism is achieved by deforming $\mathbb{T} \subset S^3$ in such a way that the Hopf link in its complement has its two constituent loops permuted.[17] This automorphism permutes the labels in $\mathcal{C}_{\text{Hopf}}$. We denote this map as $\nu : \mathcal{C}_{\text{Hopf}} \to \mathcal{C}_{\text{Hopf}}$, and $\nu(\eta) \equiv \eta^{\vee}$.

4. (Shrinkable loops) We define the set of "shrinkable loops" as $\mathcal{C}_{\text{loop}} \equiv \mathcal{C}_{\text{Hopf}}^{[1]}$. Each of these excitations can be realized by a loop that is not linked with other loops and thus can be continuously shrunk to a point. In fact, every shrinkable loop sector $l \in \mathcal{C}_{\text{loop}}$ can be created by a membrane operator supported on the side of a cylinder, together with another shrinkable loop:

---

[16]Note that a subset of excitations in $\mathcal{C}_{\text{Hopf}}$ can be created without exciting both of the loops. They are distinguished with genuine Hopf excitations that necessarily need both of the loops.

[17]Specifically, we can identify the two loops as part of a right-handed Hopf fibration of $S^3$ and deform them in such a way that they are thickened fibers of this fibration at any time. (A left-handed Hopf fibration corresponds to a different automorphism.) For this deformation, the region remains embedded. Viewed from the base space of the Hopf fibration, the image of the torus shell is an annulus, and this deformation swaps the two holes of the annulus. See Appendix E.3 for more exotic automorphisms making use of immersion.

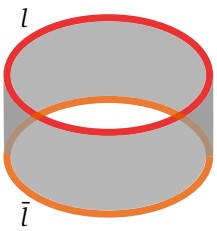

$$(3.9)$$

Each shrinkable loop has a well-defined anti-sector. (We denote the anti-sector of $l$ as $\bar{l}$.) This is because $\mathcal{C}_{\text{loop}}$ is mapped to the set of anyons in a 2d entanglement bootstrap problem, and anyons have anti-sectors. This is explained in Appendix E.

The set of fluxes $\mathcal{C}_{\text{flux}}$ is naturally embedded in $\mathcal{C}_{\text{loop}}$ as a subset.

$$\mathcal{C}_{\text{flux}} \overset{\varphi}{\hookrightarrow} \mathcal{C}_{\text{loop}} \subset \mathcal{C}_{\text{Hopf}}. \tag{3.10}$$

Here the embedding $\varphi : \mathcal{C}_{\text{flux}} \hookrightarrow \mathcal{C}_{\text{loop}}$ is defined by the following sequence of operations: (1) solid torus $T$ is embedded in a ball. Complete the ball to an $S^3$ if it is not already part of an $S^3$. (2) Let $T = BC$ where $B = \partial T$ and let $A = S^3 \setminus (BC)$. Deform $BC$ to $AB$ by a path. (Such a path exist because any pair of embedded solid tori on $S^3$ can be converted to one another by smooth deformations. There is, however, a possible flip. We chose one for concreteness.) By the isomorphism theorem, $\rho_T^\mu$ is deformed to an extreme point $\lambda_{AB}^\mu$. (3) Take a partial trace, and let $\widetilde{\rho}_B \equiv \text{Tr}_A \lambda_{AB}^\mu$. The label $\varphi(\mu) \in \mathcal{C}_{\text{loop}}$ is defined to be the label associated with the extreme point $\widetilde{\rho}_B$.

From this definition we further see that $\varphi(1) = 1$ and that the quantum dimension $d_{\varphi(\mu)} = d_\mu^2$.

Intriguingly, the set of point particles are naturally embedded in the set of shrinkable loops as well. As we explain in §6.2.2,

$$\mathcal{C}_{\text{point}} \overset{\phi}{\hookrightarrow} \mathcal{C}_{\text{loop}} \subset \mathcal{C}_{\text{Hopf}}, \tag{3.11}$$

such that $d_{\phi(a)} = d_a$.

**Remark.** The embedding of fluxes and point particles defined in Eq. (3.10) and Eq. (3.11) are natural in 3d quantum double models (with or without twist), where point particles are irreducible representations of the finite group $G$ and the fluxes are conjugacy classes. From our analysis, this structure holds on broader classes of systems, whose ground state satisfies axioms **A0** and **A1**.

We further notice a relation of the total quantum dimension:

**Proposition 3.4** (matching of total quantum dimension)**.**

$$\sum_{a \in \mathcal{C}_{\text{point}}} d_a^2 = \sum_{\mu \in \mathcal{C}_{\text{flux}}} d_\mu^2. \tag{3.12}$$

*Proof.* Make a solid torus $T$ from two balls $AB$ and $BC$ as in Fig. 10(a), where $B = B_1 B_2$. Merging the unique states in $\Sigma(AB)$ and $\Sigma(BC)$ produces the maximum-entropy state of $\Sigma(T)$,

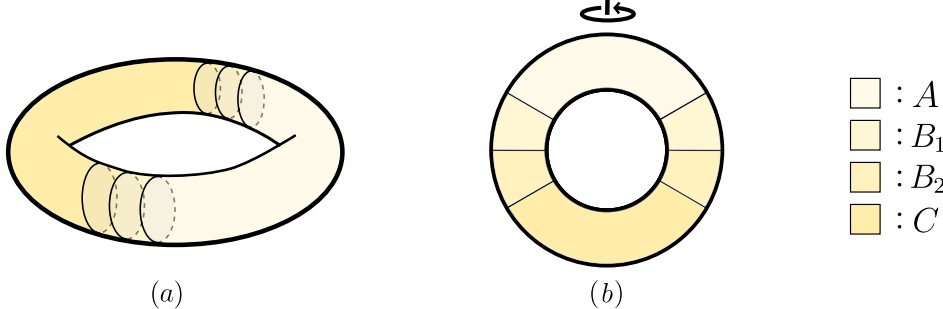

Figure 10: Making the maximum-entropy state of the information convex set of a solid torus (sphere shell) by merging the reduced density matrices of the reference state on balls.

denoted as $\rho_T^\star = \sum_\mu \frac{d_\mu^2}{\sum_\nu d_\nu^2} \rho_T^\mu$. This implies that $\rho_T^\star$ is a quantum Markov chain

$$
\begin{aligned}
0 &= I(A:C|B)_{\rho_T^\star} \\
&= -\ln\left(\sum_\mu d_\mu^2\right) + I(A:C|B)_{\sigma_T} \\
&= -\ln\left(\sum_\mu d_\mu^2\right) + 2\gamma.
\end{aligned}
\tag{3.13}
$$

The second line uses the orthogonality of the extreme points and the definition of the quantum dimension. $\gamma \equiv \frac{1}{2}I(A:C|B)_{\sigma_T}$ in the third line is the 3d version of Levin-Wen topological entanglement entropy.

Similarly, we can make a sphere shell $X$ from two balls $AB$ and $BC$; see Fig. 10(b). The merged state is $\rho_X^\star = \sum_a \frac{d_a^2}{\sum_b d_b^2} \rho_X^a$.

$$
\begin{aligned}
0 &= I(A:C|B)_{\rho_X^\star} \\
&= -\ln\left(\sum_a d_a^2\right) + I(A:C|B)_{\sigma_X}.
\end{aligned}
\tag{3.14}
$$

The second line follows from the orthogonality of the extreme points and the definition of the quantum dimension. The remaining thing is to show that $I(A:C|B)_{\sigma_X} = 2\gamma$. This is true according to Proposition D.11. This completes the proof. $\qquad\square$

**Proposition 3.5** (decomposition of $\mathcal{C}_{\text{Hopf}}$). *There is a well-defined map from $\mathcal{C}_{\text{Hopf}}$ to $\mathcal{C}_{\text{flux}}$ described by the following process. (By this map, we define the decomposition (3.8).) Let $\mathbb{T}$ be a torus shell, and $T \subset \mathbb{T}$ is a solid torus; see Fig. 11 below for an illustration. Let $\rho_{\mathbb{T}}^\eta \in \Sigma(\mathbb{T})$ be an extreme point. Then $\text{Tr}_{\mathbb{T}\backslash T}\, \rho_{\mathbb{T}}^\eta$ is an extreme point $\rho_T^\mu \in \Sigma(T)$.*

*Proof.* It follows from Proposition 2.23. To see this, we observe that $\mathbb{T}$ is a sectorizable region and that $\rho_{\mathbb{T}}^\eta$ is an extreme point. $\qquad\square$

**Remark.** The map identified in Proposition 3.5 is a special case of a class of maps we identify in §5. It is identified with $t_{(1,0)}$ among the maps $t_{(p,q)}$ defined in Eq. (5.19).

Below are a few examples. The explicit data presented here can be calculated by the procedure described in §4.

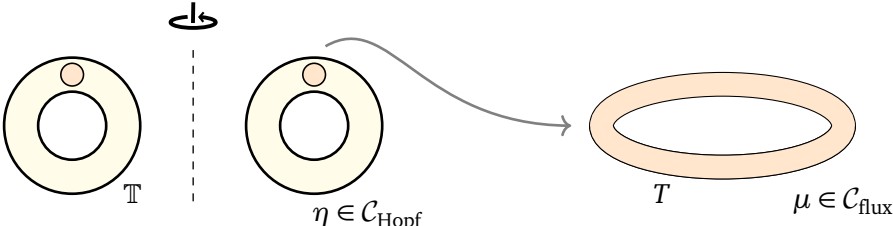

Figure 11: On the proof of the decomposition of $\mathcal{C}_{\text{Hopf}}$.

**Example 3.6** (3d toric code). 3d toric code is the 3d quantum double model with finite group $G = Z_2$. The data are

1. $\mathcal{C}_{\text{point}} = \{1, e\}$, with quantum dimension $d_1 = d_e = 1$.

2. $\mathcal{C}_{\text{flux}} = \{1, m\}$, with quantum dimension $d_1 = d_m = 1$.

3. $\mathcal{C}_{\text{loop}} = \{1, e, m, f\}$. We choose the same label as the 2d toric code model, because these can be identified with the 4 anyon types of the toric code model by a dimensional reduction.

4. $\mathcal{C}_{\text{Hopf}}$ contains 8 labels. Each of them has quantum dimension equals to 1. $\mathcal{C}_{\text{Hopf}} = \mathcal{C}_{\text{Hopf}}^{[1]} \cup \mathcal{C}_{\text{Hopf}}^{[m]}$. Both $\mathcal{C}_{\text{Hopf}}^{[1]}$ and $\mathcal{C}_{\text{Hopf}}^{[m]}$ contain 4 labels.

**Example 3.7** (3d $S_3$ quantum double). The finite group $S_3 = \{1, r, r^2, s, sr, sr^2\}$, with $r^3 = s^2 = 1$ and $sr = r^2 s$. It is the smallest non-Abelian group. This model has non-Abelian superselection sectors (whose quantum dimension is greater than 1). Without introducing heavy notations, we describe the number of excitations and the quantum dimensions.

1. $\mathcal{C}_{\text{point}}$ contains 3 labels, with $\{d_a\} = \{1, 1, 2\}$.

2. $\mathcal{C}_{\text{flux}}$ contains 3 labels, with $\{d_\mu\} = \{1, \sqrt{2}, \sqrt{3}\}$.

3. $\mathcal{C}_{\text{loop}}$ contains 8 labels, with $\{d_l\} = \{1, 1, 2, 2, 2, 2, 3, 3\}$.

4. $\mathcal{C}_{\text{Hopf}} = \cup_\mu \mathcal{C}_{\text{Hopf}}^{[\mu]}$ contains $8 + 9 + 4 = 21$ labels, with

$$\{d_\eta\} = \{1, 1, 2, 2, 2, 2, 3, 3\} \cup \{2, 2, 2, 2, 2, 2, 2, 2, 2\} \cup \{3, 3, 3, 3\}. \tag{3.15}$$

**Remark.** From these examples, we can explicitly see that $\sum_a d_a^2 = \sum_\mu d_\mu^2$. In fact, $\sum_{\eta \in \mathcal{C}_{\text{Hopf}}^{[\mu]}} d_\eta^2$ is independent of $\mu$; see Appendix E. Furthermore, both of the examples satisfy $|\mathcal{C}_{\text{flux}}| = |\mathcal{C}_{\text{point}}|$ and this is a general fact. An argument can be found in Ref. [27]. We shall provide an entanglement bootstrap derivation of this statement in [20].

## 3.3 Fusion spaces for 3d systems: basic ones

In the next three sections, we identify a variety of fusion spaces and fusion processes of the excitations types identified in §3.2. In this section, we start with a few basic ones that have been studied in literature by other approaches [28–33]. The key technique is the Hilbert space theorem (Theorem 2.14). An independent dimensional reduction method to understand these basic processes is presented in Appendix E. A few novel fusion spaces and processes are discussed in §3.4 and §3.5.

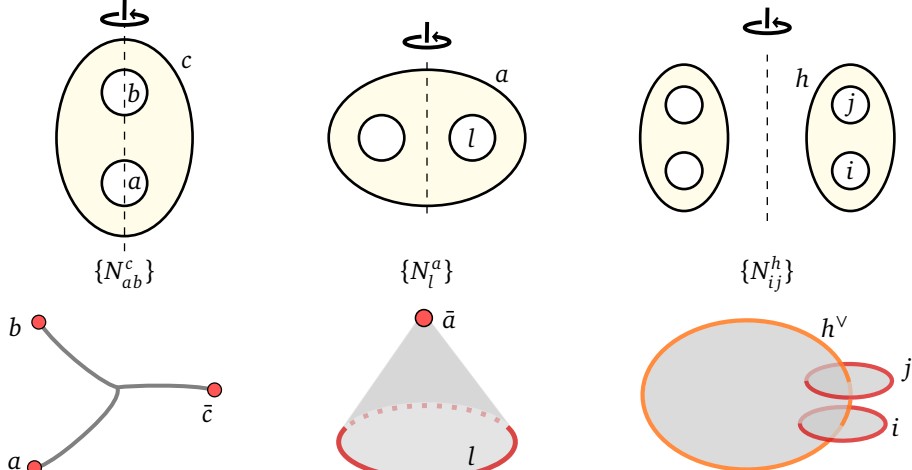

Figure 12: A list of regions, the corresponding fusion multiplicities, and the excitations with the associated string (membrane) operators. (left) The fusion of point particles. (Middle) For a shrinkable loop. (Right) Fusion of two loops in $\mathcal{C}_{\mathrm{Hopf}}^{[\mu]}$.

By the Hilbert space theorem (Theorem 2.14), information convex sets of 3d regions with boundaries can be associated with a set of fusion spaces.[18] Traditionally, the dimensions of fusion spaces appear in fusion equations. These are, roughly speaking, equations that specify the content before the fusion on the left and the possible outcomes of the fusion on the right. Below is a list of basic fusion processes in 3d. In Fig. 12 we illustrate the regions, and string (membrane) operators.

1. (ball minus two balls) The subsystem ball-minus-two-balls characterizes the fusion of two point particles[19]:

$$a \times b = \sum_c N_{ab}^c c, \quad a, b, c \in \mathcal{C}_{\mathrm{point}}. \qquad (3.16)$$

See the left of Fig. 12. The multiplicities $\{N_{ab}^c\}$ are characterized by the information convex set of ball-minus-2-balls: $N_{ab}^c = \dim \mathbb{V}_{ab}^c$(ball-minus-2-balls). These non-negative integers satisfy a set of consistency rules. We review these rules in Appendix E.

2. (ball minus unknotted solid torus) The complement of an unknot in a ball characterizes the shrinking of a shrinkable loop to a point:

$$l = \sum_a N_l^a a, \quad \text{with} \quad l \in \mathcal{C}_{\mathrm{loop}}, \ a \in \mathcal{C}_{\mathrm{point}}. \qquad (3.17)$$

We shall refer to this as the "shrinking rule". (By the Hilbert space theorem we should let the lower label of the multiplicity be $\eta \in \mathcal{C}_{\mathrm{Hopf}}$. We restrict the lower index to be $l \in \mathcal{C}_{\mathrm{loop}}$ for the reason that $N_\eta^a = 0$ for $\eta \notin \mathcal{C}_{\mathrm{loop}}$.) It is easy to see that

$$N_l^1 = \sum_{\mu \in \mathcal{C}_{\mathrm{flux}}} \delta_{l, \varphi(\mu)}, \qquad (3.18)$$

where $\varphi$ is the map defined in Eq. (3.10). The multiplicities $\{N_l^a\}$ and the consistency relations can be understood by dimensional reduction; see Appendix E.

---

[18]Sectorizable regions are special cases for which the fusion space dimensions are either 0 or 1.

[19]We will write formal equations like (3.16) indicating fusion rules by analogy with fusion rules for anyons in 2d. One precise meaning will be that replacing sector labels with an appropriate quantum dimension produces a true consistency relation.

3. (solid torus minus two solid tori) The fusion of Hopf excitations in each class $\mathcal{C}_{\text{Hopf}}^{[\mu]}$ is closed among themselves. It is tempting to write down a fusion equation for it:

$$h \times i = \sum_j N_{hi}^j \, j, \quad h, i, j \in \mathcal{C}_{\text{Hopf}}^{[\mu]}. \tag{3.19}$$

The precise meaning of multiplicities $\{N_{hi}^j\}$ are the dimensions of Hilbert spaces identified by the Hilbert space theorem (Fig. 12 top right). (They can be understood as the fusion spaces of a 2d system, by dimension reduction.) When $\mu = 1$, this corresponds to the fusion of shrinkable loops; see Refs. [28, 32, 33] for a discussion of the same phenomena. When $\mu \neq 1$, this corresponds to the fusion of two loops that are linked with a third loop. The configuration of the loop excitations is closely related to that involved in the 3-loop braiding statistics [29, 30].

**Remark.** These basic fusion spaces and the superselection sectors can be understood from a rigorous dimensional reduction point of view. This dimensional reduction is distinct from existing literature in the sense that it rigorously maps a 3d entanglement bootstrap problem to a 2d entanglement bootstrap problem. A detailed discussion is postponed to Appendix E. The idea is to look at the 2d region which becomes the 3d region under revolution. This applies to either a ball or a solid torus. The former case works for the vacuum, and the 2d system is adjacent to a gapped boundary; for the latter case, it is possible to apply the dimensional reduction to any flux. The idea is illustrated in the following figure.

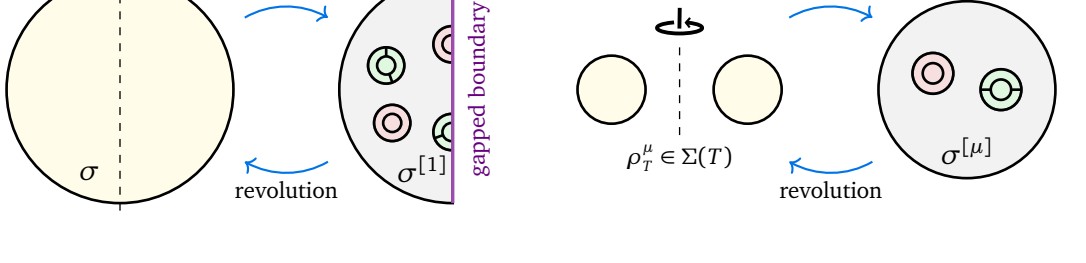

$$\tag{3.20}$$

**Example 3.8.** We provide a few basic examples of 3d quantum double, with finite group $G$, to familiarize the readers with our notations. The fusion processes described in this example are studied in Ref. [28].

- $\{N_{ab}^c\}$: The point excitations correspond to irreducible representations of the finite group $G$. $N_{ab}^c$ is the integers that appear in the tensor product of these irreducible representations.

  When $G = S_3 = \{1, r, r^2, s, sr, sr^2\}$, denote $1 = \text{Id}_{S_3}, a = \text{Sign}_{S_3}, b = \Pi_{S_3}$, where $\Pi_{S_3}$ is the 2-dimensional irreducible representation. Then the nontrivial fusion rules for point particles are

$$a \times a = 1,$$
$$a \times b = b,$$
$$b \times b = 1 + a + b.$$

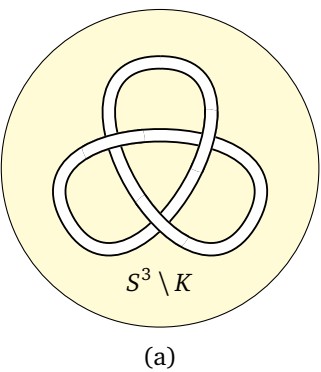

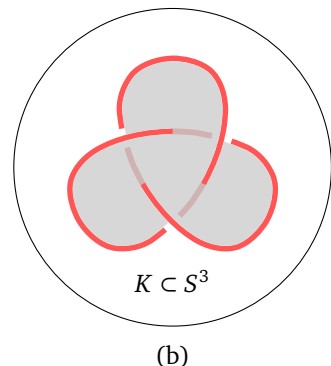

$S^3 \setminus K$          $K \subset S^3$

(a)              (b)

Figure 13: (a) The knot complement $S^3 \setminus K$. (b) The knot excitation located at $K$ and the associated (most general) membrane operator. Here $K$ can be any thickened knot. (For illustration purposes, $K$ is chosen to be a right-handed trefoil knot.)

- $\{N_l^a\}$: When $G = S_3$, there are 3 choices of $a$ and 8 choices of $l$. The multiplicities is a $3 \times 8$ table.

$$
\begin{array}{c|c|c|c|c|c|c|c|c}
\begin{array}{l} N_l^a \qquad\qquad d_l \\ \hline d_a \end{array} & 1 & 1 & 2 & 2 & 2 & 2 & 3 & 3 \\
\hline
1 & 1 & 0 & 0 & 1 & 0 & 0 & 1 & 0 \\
\hline
1 & 0 & 1 & 0 & 1 & 0 & 0 & 0 & 1 \\
\hline
2 & 0 & 0 & 1 & 0 & 1 & 1 & 1 & 1 \\
\end{array}
\tag{3.21}
$$

Here the first row and column are the vacuum sector. We, nonetheless, omit the detailed labels here. More details of the labeling can be found in Table 7 of Appendix C.2.

- $\{N_{hi}^j\}$ for $\mathcal{C}_{\text{Hopf}}^{[\mu]}$ are identical to the fusion multiplicities of anyons in a 2d quantum double model (depending on $\mu$) obtained by a dimensional reduction consideration. When $G = S_3$, the choice of flux $\mu$ corresponds to the three conjugacy classes $C_1$, $C_r$ and $C_s$. The multiplicities $\{N_{hi}^j\}$ for $\mathcal{C}_{\text{Hopf}}^{[C_1]}, \mathcal{C}_{\text{Hopf}}^{[C_r]}, \mathcal{C}_{\text{Hopf}}^{[C_s]}$ are the same as that for the 2d quantum double with group $G = S_3, \mathbb{Z}_3, \mathbb{Z}_2$ respectively.

## 3.4 Knot multiplicity

Knots are intriguing mathematical objects [34]. A knot is an embedding of the circle $S^1$ into a three-dimensional space. Knot complement plays an important role in the classification of knots. Here, we identify nontrivial fusion data associated with the information convex set of knot complements.

One may wonder why not consider the information convex set of a knot instead. The answer is twofold. First, the information convex set of a knotted solid torus does not provide now data, compared to that of an unknotted torus; see Corollary 2.6.2. Second, the information convex set of a knot complement physically characterizes loop excitations located on the knot.

For concreteness, we shall consider knot complements on a 3-sphere. See Fig. 13 for an illustration. Recall that the sphere completion lemma 3.1 indicates that even if we start from a reference state on a ball, we can recover from it a reference state on $S^3$. This mirrors a standard maneuver in the study of knots, where one can interchangeably study knot complements in $S^3$ and in $\mathbb{R}^3$. In the following we denote the knot complement on $S^3$ as

$$
\Omega_K \equiv S^3 \setminus K, \tag{3.22}
$$

where $K$ is a solid thickened knot.[20] The knot complement has one thickened boundary ($\partial\Omega_K$), which is a knotted and embedded torus shell. By the generalized isomorphism theorem (Theorem 2.5),

$$\mathcal{C}_{\partial\Omega_K} \simeq \mathcal{C}_{\mathrm{Hopf}}\,, \tag{3.23}$$

meaning that these two sets are identical up to a possible permutation of some kind. Below, we do not distinguish these two sets.

The information convex set $\Sigma(\Omega_K)$ is thus a convex hull of the subsets $\Sigma_\zeta(\Omega_K)$, where $\zeta \in \mathcal{C}_{\mathrm{Hopf}}$. In general, $\Sigma_\zeta(\Omega_K)$ is nonempty only for a subset of $\zeta \in \mathcal{C}_{\mathrm{Hopf}}$. This leads to the following theorem:

**Theorem 3.9** (knot excitation type). *The set of superselection sectors for knotted loop excitations that can exist on a knot K of a 3-sphere alone is a subset of $\mathcal{C}_{\mathrm{Hopf}}$.*

**Remark.** This implies that the knot excitation type is a small number on *any* knot, compared to the simplest link (Hopf link). However, as we shall see, excitations on knots are more coherent compared to excitations on a Hopf link. The former can be coherent, whereas the latter cannot. For example, $\Omega_{\mathrm{trefoil}}$ can encode quantum information; see Example 3.12.

By the Hilbert space theorem, $\Sigma_\zeta(\Omega_K) \cong \mathcal{S}(\mathbb{V}_\zeta(\Omega_K))$, where the finite-dimensional Hilbert space $\mathbb{V}_\zeta(\Omega_K)$ specifies the possible ways to put the loop excitation $\zeta$ on the knot $K$. The dimension of this Hilbert space will be referred to as the knot multiplicity.

**Definition 3.10** (Knot multiplicity). The knot multiplicity for knot $K$ is

$$N_\zeta(K) \equiv \dim \mathbb{V}_\zeta(\Omega_K)\,. \tag{3.24}$$

The knot multiplicity is nonnegative for any $\zeta \in \mathcal{C}_{\mathrm{Hopf}}$. Below are a few remarks on the physical meaning of $\mathbb{V}_\zeta(\Omega_K)$ and $N_\zeta(K)$. The observation is general and works for any knot including the unknot:

1. A vector $|\alpha\rangle \in \mathbb{V}_\zeta(\Omega_K)$, (up to the overall phase) represents the quantum information that can be encoded in the information convex set $\Sigma_\zeta(\Omega_K)$. This information can be decoded from the state on the knot complement $\Omega_K$. In fact, one can decode the information on any region $\widetilde{\Omega}_K \subset \Omega_K$ that can be continuously deformed to $\Omega_K$ by a sequence of extensions.

2. $\mathbb{V}_\zeta(\Omega_K)$ corresponds to the effective low energy Hilbert space associated with a single knotted excitation $\zeta$, that is robust under *any* perturbation of the excitation plus *any local* perturbations in other places. Note that the perturbation along the excitation does not have to be local. This is precisely the condition required by the authors of Ref. [32, 33].

3. On the other hand, to protect quantum information from decoherence to the environment, often a weaker statement is needed. We only need locally-indistinguishable states (i.e., the states are indistinguishable on balls with bounded radius). One may wonder if the protected degeneracy can be larger than the knot multiplicities. We do not have an answer to this question for the most general context. However, we would like to observe two things. First, when the excitation has an extra "smoothness", e.g., it behaves like a codimension-2 defect and satisfies a generalization of boundary axioms (a generalization of the version of **A0** and **A1** 2d gapped boundary), then there can be extra degeneracy; a related observation is made in Ref. [35]. The coherence of the extra degeneracy is, however, protected by the details in the vicinity of the excitation. Quantum information stored in this extra degeneracy (1) cannot be decoded in the knot complement, away from the excitation, and (2) is not robust to arbitrary perturbation along the excitation.

---

[20]When we consider a knot $K$ as a 3d region or its complement, we always think of a solid knot with a thickness large compared to the correlation length.

**Example 3.11** (unknot)**.** Let $K$ be the unknot. Then the knot multiplicity is

$$N_\eta(\text{unknot}) = \sum_{\mu \in \mathcal{C}_{\text{flux}}} \delta_{\eta, \varphi(\mu)}, \quad \forall \eta \in \mathcal{C}_{\text{Hopf}}. \tag{3.25}$$

**Example 3.12.** When the knot $K$ is a trefoil knot, the knot multiplicities and quantum dimensions for excitation types that can exist alone on the trefoil are as follows:
For the 3d toric code:

| $N_\zeta(K)$ | 1 | 1 |
|---|---|---|
| $d_\zeta$ | 1 | 1 |

$$\tag{3.26}$$

For the 3d $S_3$ quantum double model:

| $N_\zeta(K)$ | 1 | 1 | 2 | 1 |
|---|---|---|---|---|
| $d_\zeta$ | 1 | 2 | 3 | 3 |

$$\tag{3.27}$$

(Note that the data shown is identical for the left-handed trefoil and the right-handed trefoil for both models; if we included the labels $\zeta$ in the table, they need not be. The omitted data for the labeling of $\zeta$ can be found in Table 8.)

With a slight generalization, one can define the fusion space associated with a excitation located at the knot, labeled by $\zeta \in \mathcal{C}_{\text{Hopf}}$ and a point particle $a \in \mathcal{C}_{\text{point}}$. Because we are now allowed to put a point particle, the excitation type associated with the knot can now be a bigger set. Such excitation is detected by the information convex set of a ball with the knot removed; its thickened boundary has two components: an outer sphere shell labeled by a particle type and an inner torus shell boundary labeled by the knot excitation. According to the Hilbert space theorem, then, we can define

$$N_\zeta^a(K) \equiv \dim \mathbb{V}_\zeta^a(\text{ball} \setminus K). \tag{3.28}$$

The multiplicities depend on the choice of the knot and $N_\zeta(K) = N_\zeta^1(K)$. It is tempting to write a fusion rule for the shrinking of the knot into a particle, generalizing (3.17) in the case of the unknot:

$$\zeta \overset{?}{=} \sum_{a \in \mathcal{C}_{\text{point}}} N_\zeta^a(K) a. \tag{3.29}$$

The precise meaning of such a relation is not clear to us at the moment, and the formula obtained by naively replacing labels with quantum dimensions does not hold in this case (see Table 8).

### 3.4.1 Torus knots

Knots can be classified as torus knots, satellite knots and hyperbolic knots. Every knot falls into exactly one of the three categories [34]. It is interesting to ask whether it is possible to distinguish the three categories by looking at the fusion (braiding) data associated with them. We leave this as an open question. Another interesting question is whether there are reference states for which the data distinguishes a knot and its mirror image. For example, a trefoil is topologically distinct from its mirror image.

In this section, we study a specific property of torus knots. Torus knots are knots that can be put on the surface of an unknotted torus. A torus knot can be labeled by a pair $(p, q)$, where $p$ and $q$ are coprime integers. The $(p, -q)$ torus knot is the mirror image of the $(p, q)$ torus knot. The $(-p, -q)$ torus knot is equivalent to the $(p, q)$ torus knot except for the reversed orientation. The $(p, q)$ torus knot is equivalent to the $(q, p)$ torus knot. For example, the $(2, 3)$

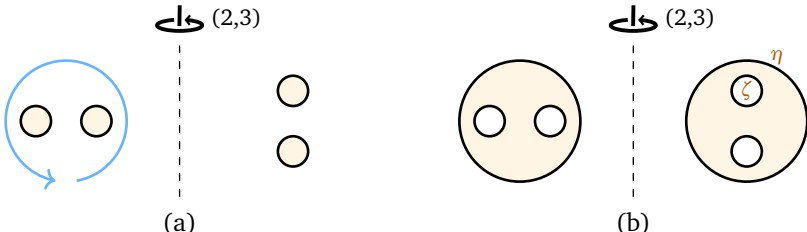

Figure 14: An illustration of $(p,q)$-type revolution: (a) A solid trefoil knot. (b) A solid torus with a trefoil knot removed; the labels are those used in (3.31).

torus knot and the $(2,-3)$ torus knot are trefoil knots with opposite chiralities; the $(p,1)$ torus knot, for any $p \in \mathbb{Z}$ is an unknot.

The number $(p,q)$ provides explicit instruction for constructing the torus knot. For our purpose, it is convenient to introduce a revolution of $(p,q)$-type; see Fig 14 for an illustration. Here $p$ is the number of times that a 2d region is rotated around the shown (vertical) axis, and $q$ is the number of times that the region is rotated around the circle located at the center of a solid torus. (The blue arrow in Fig. 14(a) illustrates the rotation around this second axis.) We allow $(p,q)$ to be $(1,0)$ or $(0,1)$ taking the obvious meaning.

For a torus knot, it is natural to consider the process of "fusing" a knot excitation to a Hopf excitation:

$$\zeta \overset{?}{=} \sum_\eta N^\eta_\zeta(p,q)\, \eta\,, \quad \zeta, \eta \in \mathcal{C}_{\text{Hopf}}\,. \tag{3.30}$$

More precisely, the Hilbert space theorem says that we can define a set of integers

$$N^\eta_\zeta(p,q) \equiv \dim \mathbb{V}^\eta_\zeta\left(T \setminus K_{p,q}\right)\,, \quad \zeta, \eta \in \mathcal{C}_{\text{Hopf}}\,, \tag{3.31}$$

where $K_{p,q}$ denotes the $(p,q)$ torus knot; see Fig. 14(b) for an illustration. $\eta$ is the label on the (unknotted) outer boundary and $\zeta$ is the label on the knot boundary. Another view of this fusion process is:

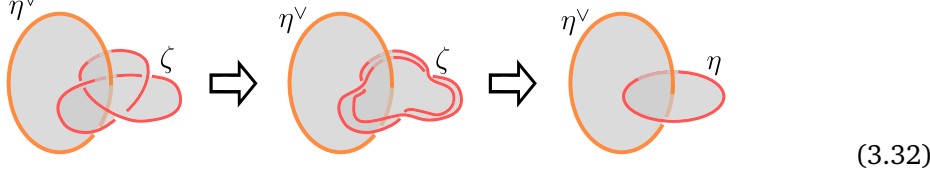

$$\tag{3.32}$$

Here the excitations are located in the complement of the region we consider, on a 3-sphere. (The difference between the second and third figures needs an explanation. We relabel $\zeta$ as $\eta$. This is allowed when either of the following happens. (1) The excitation on trefoil effectively becomes an unknot because the two strings become so close to each other that the distance between them is small compared to the correlation length. (2) We zoom out to a larger length scale compared to the distance between the two strings, which can be done by, e.g., coarse-grain the lattice further.)

## 3.5 Exotic fusion processes of flux-loops

The simplest class of loop excitations in 3d are the fluxes in $\mathcal{C}_{\text{flux}}$. We emphasize that, even for this simple set, the fusion processes can be very diverse, making use of the 3d space.[21] As

---

[21]Each of the following processes can be considered for shrinkable loops as well. The positions of the loop excitations are identical, but the membrane operators are different.

with previous examples, it is useful to look at the complement of the excitations on a $S^3$. The information convex sets know about the fusion processes.

We say some of the fusion processes below are "exotic" in the sense that: (1) for some cases, it is unclear if it is possible to assign a fusion space by applying the Hilbert space theorem; (2) for some cases, an intrinsic 3d view seems necessary, and we do not know any dimensional reduction understanding of them.

As a warm-up, we mention that the shrinking of flux loops is simple to understand. If we shrink a flux-loop to a point, it becomes the vacuum sector. This is because the membrane operator is supported on the disk. Shrinking the disk makes the operator local.

$$(3.33)$$

Below is a list of exotic cases we identify:

1. Two flux-loops can be fused on top of each other:

$$(3.34)$$

Note that the shape of the membrane operator matters, and from the shape of the membrane operator, we see that the possible fusion outcomes must be fluxes as well.

We say this case is exotic because it is unclear whether there exists a 3d region for which a set of integers labeled by three fluxes are defined:

$$N_{\mu\nu}^{\lambda} \stackrel{?}{=} \dim \mathbb{V}_{\mu\nu}^{\lambda}(\text{a certain 3d region}). \tag{3.35}$$

Here $\mathbb{V}_{\mu\nu}^{\lambda}$ is a fusion space defined by the Hilbert space theorem. It is therefore even less clear if an equation of the form:

$$\mu \times \nu \stackrel{?}{=} \sum_{\lambda} N_{\mu\nu}^{\lambda} \lambda, \quad \mu, \nu, \lambda \in \mathcal{C}_{\text{flux}}, \tag{3.36}$$

make sense physically. Nonetheless, a natural set of integer $N_{\mu\nu}^{\lambda}$ seems to be a candidate. For example, in the quantum double models, the set of fluxes is identified with the set of conjugacy classes. The "fusion" of conjugacy classes naturally provide a set of integers:

**Example 3.13** (fusion rule for conjugacy classes). For a general finite group $G$, consider the fusion of conjugacy classes:

$$C_{\mu} \times C_{\nu} = \sum_{C_{\lambda} \in (G)_{\text{cj}}} \mathcal{F}_{\mu\nu}^{\lambda} \cdot C_{\lambda}. \tag{3.37}$$

The precise definition for $\mathcal{F}_{\mu\nu}^{\lambda}$ is: for fixed group element $g_{\lambda} \in C_{\lambda}$, $\mathcal{F}_{\mu\nu}^{\lambda}$ is the number of ordered pairs $g_{\mu}, g_{\nu}$ with $g_{\mu} \in C_{\mu}, g_{\nu} \in C_{\nu}$ and $g_{\mu}g_{\nu} = g_{\lambda}$ (see, for example chapter 19 in [36]). We see $\mathcal{F}_{\mu\nu}^{\lambda}$ are non-negative integers and $\mathcal{F}_{\mu\nu}^{\lambda} = \mathcal{F}_{\nu\mu}^{\lambda}$.

For the finite group $S_3$, there are three conjugacy classes $C_1 = \{1\}$, $C_r = \{r, r^2\}$ and $C_s = \{s, sr, sr^2\}$. The fusion rules (algebra of classes) are:

$$\begin{aligned} C_1 \times C_1 &= C_1, & C_1 \times C_r &= C_r, & C_1 \times C_s &= C_s, \\ & & C_r \times C_r &= 2C_1 + C_r, & C_r \times C_s &= 2C_s, \\ & & & & C_s \times C_s &= 3C_1 + 3C_r. \end{aligned} \tag{3.38}$$

Can this set of integers play a physical role, whether or not Eq. (3.35) holds? We leave the solution to this puzzle as an open question.

It is worth noting a distinct fusion process: the fusion of two independently-created shrinkable loops, each of which is created on a membrane operator supported on the side of a cylinder:

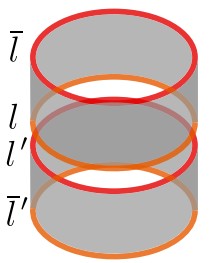

$$(3.39)$$

The fusion multiplicities of this process are understood in dimensional reduction picture and there is a well-defined set of integers $\{N_{ll'}^{l''}\}$. Here $l, l', l'' \in \mathcal{C}_{\text{loop}}$. (In the context of quantum double models, this is first discussed in Ref. [28].) Restricting the set of shrinkable loops to the subset $\{\varphi(\mu)\}$ (in both the incoming loops and the fusion outcomes), in general, provides a set of integers different from those obtained in Eq. (3.38). For example, it is impossible to have $N_{\varphi(\mu)\varphi(\nu)}^1 > 1$. We argue that this does not contradict what we discussed above because the two physical processes are different.

2. A flux-loop can be twisted spirally and then becomes a new flux-loop.

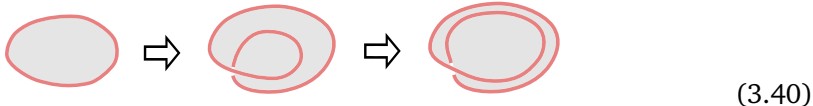

$$(3.40)$$

What is illustrated here is the spiral labeled by an integer $n = 2$. It is possible to generalize this to any integers.

**Example 3.14.** In the case of the quantum double with finite group $G$, the operation described in (3.40) acts by the group law of $G$ on a representative of the conjugacy class labeling the initial flux loop.

Interestingly, every $\mu \in \mathcal{C}_{\text{flux}}$ is mapped to a unique outcome. We shall explain a general proof of this fact in §5.1.1, which makes use of the spiral map that we introduce in §5.

3. A pair of fluxes can turn into a "graph excitation" by fusing part of the loops.

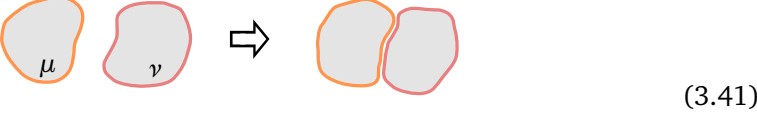

$$(3.41)$$

In general, the possible fusion outcomes for $\mu, \nu \in \mathcal{C}_{\text{flux}}$ can be multiple graph excitations. We will have more to say about these graph excitations and their creation by this fusion process in [20].

4. Two flux-loops can "collide" and become a new flux-loop as follows:

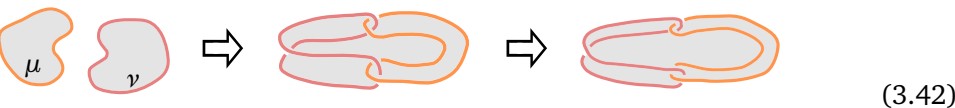

$$(3.42)$$

The final outcome of this process can be a non-vacuum flux sector only if both initial fluxes are non-vacuum ($\mu \neq 1$ and $\nu \neq 1$). However, we do not know if it is sensible to write an equation:

$$\mu \times \nu \stackrel{?}{=} \sum_{\lambda} N_{\mu\nu}^{\lambda} \lambda \,, \tag{3.43}$$

with an appropriate set of integers $\{N_{\mu\nu}^{\lambda}\}$.

We note that this fusion process is closely related to the rule for the crossing of string defects in a 3d ordered medium whose order parameter space has non-abelian fundamental group $\pi$ [37]. In that context, string types are labeled by conjugacy classes of $\pi$. When two string segments pass through each other, they leave behind a connecting string labeled by the group commutator $PQP^{-1}Q^{-1}$ of representatives of the respective conjugacy classes, $P$ and $Q$. (Though different choices of representatives $P$ and $Q$ can lead to different fusion outcomes, in this classical context, there is not really a non-abelian fusion rule; rather, the outcome depends on the details of the fusion process.)

**Example 3.15.** In the case of the 3d quantum double model, the outcome of the process in (3.42) is also constrained by the possible values of group commutators $PQP^{-1}Q^{-1}$, where $P$ and $Q$ are representatives of the conjugacy classes $\mu$ and $\nu$, respectively. In particular, therefore, the outcome is always trivial if $G$ is Abelian.

More generally, the outcome is constrained by the fusion multiplicity of the Borromean rings complement: let $Y$ be a solid torus surrounding the red and orange loops in the right figure of (3.42). Explicitly, $Y$ is the yellow solid torus in the following figure:

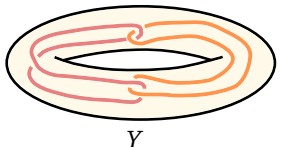 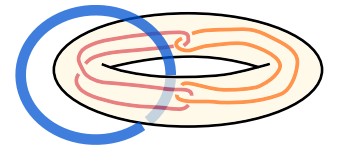

$$\tag{3.44}$$

Then the outcome of the fusion is measured by the state of a solid torus which is contained in the complement of $Y$, e.g., the blue solid torus in Eq. (3.44). $Y$ minus the thickened red and orange loops is a Borromean rings complement. Therefore, we will call this type of fusion of loops 'Borromean fusion'. However, the relation between this fusion process and the multiplicities associated with the information convex set of $Y$ We leave the precise relation as an open question. In Appendix C.5, we calculate Borromean ring multiplicities in quantum double examples, including the special cases relevant to the Borromean fusion of flux loops.

# 4 Examples from solvable models

In this section, we present explicit examples of the data we identified in §3. These examples come from a particular class of solvable models: the 3d quantum double model associated with a finite group $G$. It is a natural generalization of Kitaev's quantum double model [4], and at low energies reduces to lattice gauge theory with gauge group $G$. The ground state of this model is known to satisfy the two entanglement bootstrap axioms [38], and that is why it can serve as an example. The subdivision-invariance of the 3d quantum double model, i.e., the ability to add and remove qubits in a "smooth" way using entanglement renormalization [23, 24] makes it possible to reduce the calculation to a small lattice. This calculation approach may apply to other solvable models with subdivision-invariance, e.g., the 3d Dijkgraaf-Witten

models [39, 40] and the 3d Walker-Wang models with interesting boundary excitations and with or without deconfined bulk excitations [41, 42].

In §4.1 we review the 3d quantum double model. In §4.2 we review the "minimal diagram" technique [17], which is a way to calculate the information convex set of quantum double models. In §4.3, we present the minimal diagram for a few basic cases and the rules that lead to the explicit data.

## 4.1 3d quantum double model

The 3d quantum double is a lattice model with input: a finite group $G$ and a 3d lattice. The Hilbert space of the 3d quantum double model is a tensor product of local Hilbert spaces associated with each link of the 3d lattice ($\mathcal{H} = \otimes_e \mathcal{H}_e$). Each link of the 3d lattice is associated with a finite dimensional Hilbert space $\mathcal{H}_e = \text{span}\{|g\rangle | g \in G\}$. Here $\{|g\rangle\}$ is an orthonormal basis labeled by group elements. The orientation of each link can be chosen at will, but when the orientation is flipped, the basis vector is relabeled as $|g\rangle \rightarrow |g^{-1}\rangle$, where $g^{-1} \in G$ is the inverse of $g$. The physics of interest is insensitive to the detailed geometry of the lattice, and when we introduce the Hamiltonian, we shall consider the cubic lattice for concreteness.

The Hamiltonian for the 3d quantum double model is local, and it consists of two types of local terms:

$$H = -\sum_v A_v - \sum_p B_p. \tag{4.1}$$

Here $A_v$ is a vertex term acting on the (six) links adjacent to vertex $v$ (and is a sum of generators of gauge transformations for lattice gauge theory with gauge group $G$). $B_p$ is the plaquette term acting on (four) links that make up the boundary of a plaquette $p$ (and, in the language of lattice gauge theory, measures the flux through the plaquette). These operators are defined according to the following action on their supports:

- $A_v \equiv \frac{1}{|G|} \sum_{g \in G} A_v^g$, with

$$A_v^g \left| \begin{array}{c} \overset{e}{\underset{b}{\nearrow}} \overset{d}{\underset{v}{\longrightarrow}} c \\ a \overset{}{\underset{f}{\longleftarrow}} \end{array} \right\rangle = \left| \begin{array}{c} \overset{ge}{\underset{gb}{\nearrow}} \overset{gd}{\underset{v}{\longrightarrow}} gc \\ ga \overset{}{\underset{gf}{\longleftarrow}} \end{array} \right\rangle. \tag{4.2}$$

- $B_p$ is defined for a plaquette $p$ oriented $xy$, $yz$, $zx$-plane, respectively, as

$$B_p \left| \begin{array}{c} d \overset{c}{\nearrow} b \\ \overset{}{\underset{a}{p}} \end{array} \right\rangle = \delta_{1,abc^{-1}d^{-1}} \left| \begin{array}{c} d \overset{c}{\nearrow} b \\ \overset{}{\underset{a}{p}} \end{array} \right\rangle$$

$$B_p \left| \begin{array}{c} \overset{c}{\nearrow} b \\ d \overset{p}{\underset{a}{\nearrow}} \end{array} \right\rangle = \delta_{1,abc^{-1}d^{-1}} \left| \begin{array}{c} \overset{c}{\nearrow} b \\ d \overset{p}{\underset{a}{\nearrow}} \end{array} \right\rangle \tag{4.3}$$

$$B_p \left| \begin{array}{c} \overset{c}{\rightarrow} \\ d \uparrow \overset{p}{} \uparrow b \\ \overset{}{\underset{a}{\rightarrow}} \end{array} \right\rangle = \delta_{1,abc^{-1}d^{-1}} \left| \begin{array}{c} \overset{c}{\rightarrow} \\ d \uparrow \overset{p}{} \uparrow b \\ \overset{}{\underset{a}{\rightarrow}} \end{array} \right\rangle$$

Note that it does not matter on which one of the four vertices of the plaquette the product starts. All that matters is to take into account the arrows and the ordering.

## 4.2 Information convex set and minimal diagram

To compute the information convex set (defined in Definition 2.3) for various regions in the 3d quantum double model, we use the minimal diagram technique introduced in [17]. Note that in Ref. [17], the definition of information convex set uses a parent Hamiltonian rather than a reference state. In general, these two definitions of information convex set are inequivalent. Nevertheless, these two definitions are equivalent if the ground state satisfies the area law; see [43] (Theorem 10.1 of Chapter 10 in particular).

### 4.2.1 Choose a minimal diagram

Roughly speaking, a "minimal diagram" is a graph (more precisely, a cell complex) with a small number of links and a finite-dimensional Hilbert space, obtained from reducing links and constraints from a finite but arbitrarily large subsystem of the lattice model, using the subdivision-invariance of the solvable lattice model. This philosophy is familiar in the calculation of ground states of solvable models, e.g., the string-net model. The main difference is that the rules for minimal diagrams come from those for computing the information convex set. These rules are, in general, inequivalent to those for finding the set of ground states of a Hamiltonian on a small lattice. Without further ado, here is what we mean by a minimal diagram explicitly:

**Minimal diagram:** A minimal diagram, of 3d quantum double model, for a region $\Omega$ contains:

- a number of links

  - boundary links
  - bulk links

- a number of 2d faces

- a number of vertices

  - bulk vertices, not an endpoint of any boundary link
  - boundary vertices, an endpoint of a boundary link

We shall choose blue for boundary links and purple for bulk links. The local Hilbert space on each link is $|G|$-dimensional. The total Hilbert space for this minimal diagram, denoted by $\mathcal{H}_\Omega^{mini}$, is the tensor product of these local Hilbert spaces.

**Remark.** A few remarks are in order:

1. If the group element of a link is constrained to be the identity $1 \in G$ by a zero-flux constraint described in §4.2.2, then it is possible to go back and simplify the diagram by deleting some links and faces.

2. We do not require a minimal diagram to be the smallest choice. We want it to be small enough to do a calculation.

3. When we draw a minimal diagram, it is often useful to draw additional lines to illustrate the background topological space in which the diagram lives. Those lines are not part of the minimal diagram.

4. For all the examples in 3d bulk of which we are aware, we find that it is possible to choose one boundary vertex for each connected component of the boundary. However, this is not true in a broader context. For example, this stops being true when we consider regions attached to a gapped boundary.

5. If $\Omega$ is a closed manifold, the sets of boundary links and boundary vertices are empty.

### 4.2.2 Rules for calculation with a chosen minimal diagram

The goal of a minimal diagram calculation is to find a convex set of density matrices on $\mathcal{H}_\Omega^{mini}$, satisfying a few constraints. The convex set obtained this way is isomorphic to the information convex set $\Sigma(\Omega)$.

These constraints follow from the three types of constraints for the calculation of an information convex set of quantum double for an arbitrarily large subsystem: (1) the boundary links are block-diagonal, (2) the constraints from terms acting on the interior (zero-flux constraints and vertex projection), and (3) the invariance under conjugation by (truncated) boundary vertex terms.[22]

For a chosen minimal diagram, the explicit rules for the calculation are the following. We would like to find the set of all density matrices $\rho$ on $\mathcal{H}_\Omega^{mini}$ satisfying:

1. (block diagonal) $\rho$ is block-diagonal in the group element basis of the boundary links, namely

$$\rho = \sum_{\{g_i\}} p_{\{g_i\}} \rho_{\{g_i\}}, \qquad (4.4)$$

where $\{g_i\}$ is a set of group element labels for the boundary links, $\{p_{\{g_i\}}\}$ is a probability distribution and $\{\rho_{\{g_i\}}\}$ is a density matrix living in the subspace of $\mathcal{H}_\Omega^{mini}$ such that the boundary links are fixed to be the chosen group elements $\{g_i\}$.

2. (projectors within the bulk) There are two types. The first type is the projector for each face $f$ of the minimal diagram: $B_f \rho = \rho$. Here $B_f$ is the projector onto zero total flux on the links surrounding face $f$, an analog of Eq. (4.3); we call them zero-flux constraints. The second type is of the form $A_v \rho = \rho$ for each bulk vertex. $A_v$ is an analog of the vertex term in the 3d quantum double Hamiltonian.

3. (conjugation invariant) $A_v^g \rho A_v^{g\dagger} = \rho$, for every boundary vertex $v$ of the minimal diagram and $\forall\, g \in G$. Here the unitary operator $A_v^g$ is analogous to that defined in Eq. (4.2).

**Remark.** In practice, one can simplify the calculations further by using the technique reviewed in Appendix C.1. In particular, Theorem C.1 is useful in solving the conjugation constraint. When $\Omega$ is a closed manifold, only the second rule survives; the rules reduce to the familiar rules for calculating the convex set of (possibly mixed) ground states.

## 4.3 Explicit data

We provide the calculation of the information convex sets of 3d quantum double models, focusing on the simplest superselection sectors and one nontrivial fusion space (the knot multiplicity of trefoil). Additional cases are presented in Appendix C.

### 4.3.1 Superselection sectors

Below, we summarize the superselection sector data calculated from 3d quantum double models (with finite group $G$). The minimal diagram and the explicit constraints that lead to the calculation results are described.

---

[22]See equations (34), (35) and (36) of Ref. [17] for the explicit expressions in the context of 2d quantum double. The constraints for 3d quantum double are similar because both models have vertex terms and plaquette terms.

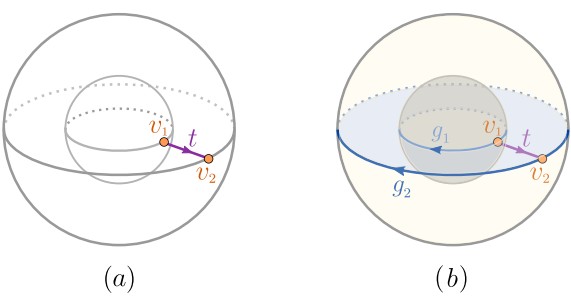

Figure 15: Minimal diagram for sphere shell.

**(Sphere shell and point excitations)** The superselection sectors of point excitations correspond to the irreducible representations of finite group $G$: $R \in (G)_{\text{ir}}$.

$$\mathcal{C}_{\text{point}} = \{1, a, \cdots\}, \quad \text{where } a = R \in (G)_{\text{ir}}. \tag{4.5}$$

Here, the label 1 corresponds to the 1-dimensional identity representation ($\text{Id}_G$). The quantum dimensions are

$$d_a = \dim R, \quad \text{for } a = R \in (G)_{\text{ir}}, \tag{4.6}$$

where $\dim R$ is the dimension of the irreducible representation $R$.

The minimal diagram calculation that leads to this result is as follows. We consider the minimal diagram for the sphere shell, shown in Fig. 15(a). It contains two vertices and a single link, which is a bulk link. (An alternative minimal diagram is Fig. 15(b), which contains 1 bulk link, 2 boundary links, 2 vertices and 5 faces. This one can be simplified to that in Fig. 15(a), according to the rule to remove faces and links.) The Hilbert space is $\mathcal{H}_{S^2 \times \mathbb{I}}^{mini} = \text{span}\{|t\rangle | t \in G\}$. Two constraints are associated with vertices $v_1$ and $v_2$:

- $A_{v_1}^g \rho A_{v_1}^{g\dagger} = \rho$, $\forall g \in G$, where $A_{v_1}^g$ is defined according to $A_{v_1}^g |t\rangle = |gt\rangle$.

- $A_{v_2}^g \rho A_{v_2}^{g\dagger} = \rho$, $\forall g \in G$, where $A_{v_2}^g$ is defined according to $A_{v_2}^g |t\rangle = |tg^{-1}\rangle$.

The goal is to find the convex set of density matrices supported on $\mathcal{H}_{S^2 \times \mathbb{I}}^{mini}$ that satisfy these two constraints. The end result is a simplex with extreme points labeled by $R \in (G)_{\text{ir}}$. The quantum dimension Eq. (4.6) is verified by calculating the entropy difference between the extreme points. We omit the details here since the problem is solved directly by applying Proposition C.1, using Eq. (C.6).

If $G = S_3$, the information convex set of the sphere shell has three extreme points. They correspond to three point particle types:

$$\mathcal{C}_{\text{point}} = \{\text{Id}_{S_3}, \text{Sign}, \Pi\}, \quad \text{with quantum dimensions} \quad \{d_a\} = \{1, 1, 2\}. \tag{4.7}$$

Here $\text{Id}_{S_3}$, Sign and $\Pi$ are the identity, sign and two dimensional irreducible representation of $S_3$ respectively.

**(Solid torus and pure fluxes)** The superselection sectors of pure fluxes correspond to the conjugacy classes of finite group $G$: $C \in (G)_{\text{cj}}$.

$$\mathcal{C}_{\text{flux}} = \{1, \mu, \cdots\}, \quad \text{where } \mu = C \in (G)_{\text{cj}}. \tag{4.8}$$

Here, 1 corresponds to the conjugacy class that contains the identity group element. The quantum dimensions are

$$d_\mu = \sqrt{|C|}, \quad \text{for } \mu = C \in (G)_{\text{cj}}, \tag{4.9}$$

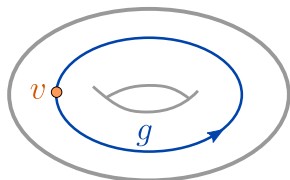

where $|C|$ is the number of group elements in conjugacy class $C$.

The minimal diagram calculation of this result is as follows. Take the minimal diagram of the solid torus shown in the following figure. It contains a single link living on the surface of the solid torus, a single vertex, and no face. (Accurately speaking, one may draw a diagram with additional faces and links, but then the faces are removed in the process of simplifying the diagram.) The Hilbert space associated with this minimal diagram is $\mathcal{H}_T^{mini} = \{|g\rangle|g \in G\}$. We consider the convex set of density matrices supported on $\mathcal{H}_T^{mini}$ that satisfy the following two conditions:

- $\rho = \sum_g p_g |g\rangle\langle g|$, with a probability distribution $\{p_g\}$.

- $\rho = A_v^h \rho A_v^{h\dagger}, \forall h \in G$.

The first condition comes from the fact that the link is a boundary link. The second condition comes from the vertex. Here the operator $A_v^h$ acts as $A_v^h |g\rangle = |hgh^{-1}\rangle$. The solution is a convex set with extreme points labelled by conjugacy classes $C$:

$$\rho^C = \frac{1}{|C|} \sum_{g \in C} |g\rangle\langle g|, \quad \forall C \in (G)_{\mathrm{cj}}. \tag{4.10}$$

This verifies Eq. (4.8). Furthermore, Eq. (4.9) is obtained by computing the entropy differences between these extreme points and the vacuum sector $\rho^{C_1}$, with $C_1 \equiv \{1\}$.

If $G = S_3$, the information convex set of the solid torus has three extreme points. Thus, there are three fluxes:

$$\mathcal{C}_{\mathrm{flux}} = \{C_1, C_r, C_s\}, \quad \text{with} \quad \{d_\mu\} = \{1, \sqrt{2}, \sqrt{3}\}. \tag{4.11}$$

Here $C_g$ is the conjugacy class of $G$ that contains element $g \in G$. Explicitly, for $S_3 = \{1, r, r^2, s, sr, sr^2\}$, where $r^3 = s^2 = 1$ and $sr = r^2 s$, the three conjugacy classes are $C_1 = \{1\}$, $C_r = \{r, r^2\}$ and $C_s = \{s, sr, sr^2\}$.

**(Torus shell and Hopf excitations)** The information convex set of the torus shell characterizes the Hopf excitations. For 3d quantum double models, these are

$$\boxed{\mathcal{C}_{\mathrm{Hopf}} = \{1, \eta, \cdots\}, \quad \text{where} \quad \eta = \left(C_{(g,h)}, R\right), \text{ and } gh = hg.} \tag{4.12}$$

Here $C_{(g,h)} \equiv \{(tgt^{-1}, tht^{-1})|t \in G\}, R \in (E_{(g,h)})_{\mathrm{ir}}$ and $E_{(g,h)} \equiv \{t \in G|(g,h) = (tgt^{-1}, tht^{-1})\}$. The quantum dimensions are

$$\boxed{d_\eta = \frac{|G|}{|E_{(g,h)}|} \cdot \dim R, \quad \text{for} \quad \eta = \left(C_{(g,h)}, R\right).} \tag{4.13}$$

$\eta \in \mathcal{C}_{\mathrm{Hopf}}^{[\mu]}$, for $\mu = C_g$.

The minimal diagram calculation that leads to this result is as follows. Again, we first draw a minimal diagram of the torus shell. The choice below contains 5 links, 2 vertices, and 3 faces; one of the links is a bulk link, and the others are boundary links.

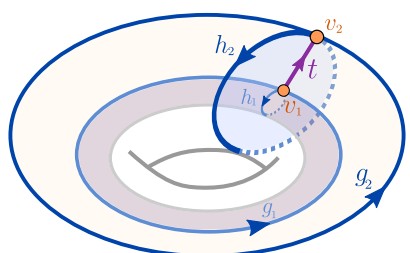

The Hilbert space associated with this minimal diagram is

$$\mathcal{H}_{\mathbb{T}}^{mini} = \{|g_1, h_1, g_2, h_2, t\rangle |g_1, h_1, g_2, h_2, t \in G\}. \tag{4.14}$$

(The association of the group elements with the links is specified in the figure.) The problem is to find the convex set of density matrices on $\mathcal{H}_{\mathbb{T}}^{mini}$ satisfying the following constraints:

- Constraints from the boundary links indicate that the relevant density matrices can be written in the diagonal basis as

$$\rho = \sum_{g_1, h_1, g_2, h_2, \lambda} p_{g_1, h_1, g_2, h_2}^{\lambda} |\{g_1, h_1, g_2, h_2\}; \lambda\rangle\langle\{g_1, h_1, g_2, h_2\}; \lambda|, \tag{4.15}$$

where $\{p_{g_1, h_1, g_2, h_2}^{\lambda}\}$ is a probability distribution and $\lambda$ is an abstract label such that

$$|\{g_1, h_1, g_2, h_2\}; \lambda\rangle = \sum_t c_\lambda(t, g_1, h_1, g_2, h_2)|g_1, h_1, g_2, h_2, t\rangle \tag{4.16}$$

is a normalized vector.

- Zero-flux constraints from the three faces are: $g_1 h_1 = h_1 g_1$, $h_2 = t^{-1} h_1 t$ and $g_2 = t^{-1} g_1 t$ ($g_2 h_2 = h_2 g_2$ is implied by these). This constraint is satisfied for all configurations with $c_\lambda(t, g_1, h_1, g_2, h_2) \neq 0$.

- Constraints from the vertices are:

$$A_{v_1}^g \rho A_{v_1}^{g\dagger} = \rho, \quad A_{v_2}^g \rho A_{v_2}^{g\dagger} = \rho, \quad \forall g \in G. \tag{4.17}$$

Here $A_{v_1}^g$ and $A_{v_2}^g$ act as

$$\begin{aligned}
A_{v_1}^g |g_1, h_1, g_2, h_2, t\rangle &= |g g_1 g^{-1}, g h_1 g^{-1}, g_2, h_2, g t\rangle, \\
A_{v_2}^g |g_1, h_1, g_2, h_2, t\rangle &= |g_1, h_1, g g_2 g^{-1}, g h_2 g^{-1}, t g^{-1}\rangle.
\end{aligned} \tag{4.18}$$

By solving this constraint problem we are able to verify the labeling of the superselection sectors Eq. (4.12) and the quantum dimensions Eq. (4.13). We omit the details but point out that Proposition C.1 is useful for solving this problem. This result agrees with the calculation using the minimal entangled states on $T^3$ [31].

**Example 4.1** (Hopf excitations). For the quantum double model with finite group $G = S_3 = \{1, r, r^2, s, sr, sr^2\}$, where $r^3 = s^2 = 1$ and $sr = r^2 s$, there are 3 point particles and 3 pure fluxes, as we have already described. The set of Hopf excitations, $\mathcal{C}_{\text{Hopf}} = \cup_\mu \mathcal{C}_{\text{Hopf}}^{[\mu]}$, contains 8+9+4=21 labels, summarized in Table 2.

Table 2: The Hopf excitations for the 3d $S_3$ quantum double, and their quantum dimensions.

| $\mu$ | $\{\eta\}_{\eta\in\mathcal{C}_{\mathrm{Hopf}}^{[\mu]}}$ | $\{d_\eta\}_{\eta\in\mathcal{C}_{\mathrm{Hopf}}^{[\mu]}}$ |
|---|---|---|
| $C_1$ | $(C_{(1,1)},\mathrm{Id}_{S_3}),(C_{(1,1)},\mathrm{Sign}),(C_{(1,1)},\Pi),$ $(C_{(1,r)},\mathrm{Id}_{Z_3}),(C_{(1,r)},\omega),(C_{(1,r)},\omega^2),$ $(C_{(1,s)},\mathrm{Id}_{Z_2}),(C_{(1,s)},\mathrm{Sign})$ | $\{1,1,2,2,2,2,3,3\}$ |
| $C_r$ | $(C_{(r,1)},\mathrm{Id}_{Z_3}),(C_{(r,1)},\omega),(C_{(r,1)},\omega^2),$ $(C_{(r,r)},\mathrm{Id}_{Z_3}),(C_{(r,r)},\omega),(C_{(r,r)},\omega^2),$ $(C_{(r,r^2)},\mathrm{Id}_{Z_3}),(C_{(r,r^2)},\omega),(C_{(r,r^2)},\omega^2),$ | $\{2,2,2,2,2,2,2,2,2\}$ |
| $C_s$ | $(C_{(s,1)},\mathrm{Id}_{Z_2}),(C_{(s,1)},\mathrm{Sign}),$ $(C_{(s,s)},\mathrm{Id}_{Z_2}),(C_{(s,s)},\mathrm{Sign}),$ | $\{3,3,3,3\}$ |

**(Shrinkable loops)** Because the set of shrinkable loops $\mathcal{C}_{\mathrm{loop}}$ is a subset of the Hopf excitations, we do not need a new minimal diagram. Instead, we take $g_1 = g_2 = 1$ in the minimal diagram for the torus shell, and find

$$\mathcal{C}_{\mathrm{loop}} = \{1, l, \cdots\}, \quad \text{where} \quad l = (C, R). \tag{4.19}$$

Here $C \in (G)_{\mathrm{cj}}$ is a conjugacy class, $R \in (E_h)_{\mathrm{ir}}$, and $E_h \equiv \{t \in G | h = tht^{-1}\}$ is the centralizer of a representative $h \in C$. The quantum dimensions are

$$d_l = |C| \cdot \dim R, \quad \text{for} \quad l = (C, R). \tag{4.20}$$

Eqs (4.19) and (4.20) should be familiar since they are the labels for the anyons of a 2d quantum double model with finite group $G$ [4, 26]. This is consistent with existing dimensional reduction statements in literature [28, 44, 45].

### 4.3.2 Knot multiplicity

In this section, we discuss the minimal diagram calculation of the knot multiplicity of a trefoil knot. Trefoil is a $(2, 3)$ torus knot. The strategy of the calculation generalizes straightforwardly to other torus knots. A notable feature is the role played by the knot group. The knot group of knot $K$ is the fundamental group of the knot complement $S^3 \setminus K$. For the trefoil knot

$$\pi_1(S^3 \setminus \text{trefoil}) = \langle \mathfrak{a}, \mathfrak{b} | \mathfrak{a}^2 = \mathfrak{b}^3 \rangle. \tag{4.21}$$

In other words, the fundamental group is a group with two generators $\mathfrak{a}$ and $\mathfrak{b}$ such that $\mathfrak{a}^2 = \mathfrak{b}^3$.

In the minimal diagram, Fig. 16(a), we have two bulk links (labeled by $a$ and $b$), two boundary links (labeled by $g$ and $h$) and a vertex $v$. The minimal diagram contains faces (not shown). The Hilbert space associated with it is $\mathcal{H}_{S^3\setminus K}^{mini} = \mathrm{span}\{|g, h, a, b\rangle | g, h, a, b \in G\}$. Again, the problem is to find the set of density matrices on this Hilbert space satisfying a few constraints.

The choice of the two bulk links is inspired by the structure of the knot group (4.21); as the labels $a$ and $b$ suggest, the group elements on these two links satisfy the knot group constraint $a^2 = b^3$ by the zero-flux constraints of the faces of the minimal diagram. ($a, b \in G$ are analogs of the generators $\mathfrak{a}$ and $\mathfrak{b}$ but they are restricted to the finite group $G$. The Hilbert space of the minimal diagram furnishes a representation of the knot group in $G$.) Because the minimal diagram has only one vertex and thus every link is a closed-loop, $a$ and $b$ determine the group element on all other links.

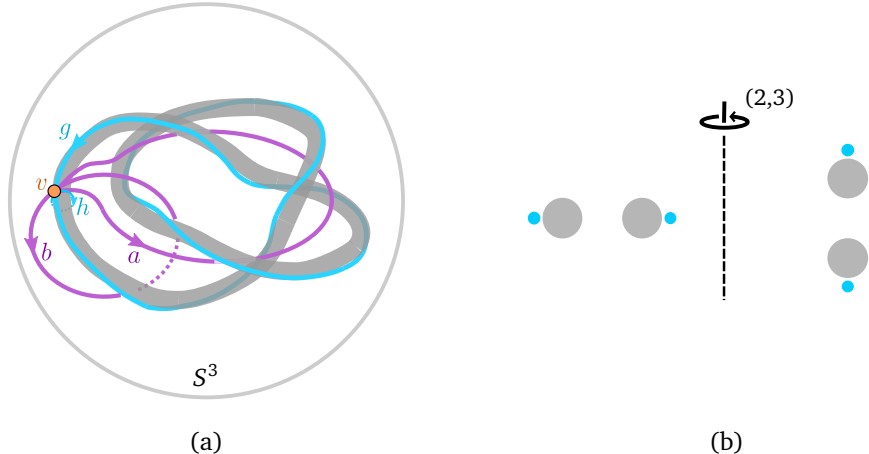

(a)          (b)

Figure 16: (a) The minimal diagram for a trefoil complement. The links labeled by group elements $g$ and $h$ live on the surface of the trefoil. The links labeled by $a$ and $b$ are bulk links corresponding to the generators of the knot group. (b) An illustration of the position relation between the removed trefoil (gray) and the link labeled by $g$ (blue), i.e., the choice of "framing" of the loop labeled by $g$.

**Remark.** This feature generalizes to the minimal diagram of all knots! For any knot, the minimal diagram contains a single vertex, two boundary links, and the number of bulk links can be chosen to have a one-to-one correspondence with knot group generators (with values restricted to finite group $G$).

Note that the choice for the loop labeled by $g$ is not unique since we can add a "Dehn twist" so that the framing changes. The choice of framing does affect the names of the $\eta$ labels without changing the physics. For a concrete calculation, however, it is crucial to keep track of the specific choice,[23] and this is illustrated in Fig. 16(b). For this choice, $\{h, g\}$ are determined by knot group generators $a$ and $b$ as follows:

$$g = a^2 = b^3, \quad h = a^{-1}b. \tag{4.22}$$

From this, we already see $gh = hg$, even before the explicit usage of the boundary face. The solution of the other two constraints leads to the formula:

$$\boxed{N_\zeta(\text{trefoil}) = \text{ the number of } R \in (E_{(g,h)})_{\text{ir}} \text{ contained in } \mathcal{H}_{(g,h)}.} \tag{4.23}$$

Here, $\zeta = (C_{(g,h)}, R)$ and the $\mathcal{H}_{(g,h)}$ is a representation of $E_{(g,h)}$ defined according to the following steps:

1. Pick a $C_{(g,h)}$ such that $g, h \in G$ and $gh = hg$.

2. Find the set of ordered pairs $\{a, b\}$, $a, b \in G$, such that Eq. (4.22) holds.

3. $\mathcal{H}_{(g,h)}$, as a Hilbert space, is defined as $\text{span}\{|a, b\rangle | \{a, b\} \text{ from step 2}\}$.

4. $\mathcal{H}_{(g,h)}$ is a representation of $E_{(g,h)}$ by the group action:

$$\Gamma(t)|a, b\rangle = |tat^{-1}, tbt^{-1}\rangle, \quad \forall t \in E_{(g,h)}. \tag{4.24}$$

---

[23]As a matter of fact, the choice of framing in Fig. 16 is consistent with the generalized isomorphism theorem in the following sense: There exists a path (with immersed regions in the intermediate steps) which turns the trefoil shell to an unknotted torus shell such that the framing becomes the usual framing for the unknotted torus shell.

Table 3: Trefoil knot multiplicities for 3d toric code, with $\mathbb{Z}_2 = \{1, s\}$.

| $\zeta$ | $(C_{(1,1)}, \mathrm{Id}_{\mathbb{Z}_2})$ | $(C_{(1,s)}, \mathrm{Id}_{\mathbb{Z}_2})$ |
|---|---|---|
| $N_\zeta(\text{trefoil})$ | 1 | 1 |
| $d_\zeta$ | 1 | 1 |

Table 4: Trefoil knot multiplicities for 3d $S_3$ quantum double. Only the excitation types with $N_\zeta(\text{trefoil}) \geq 1$ are shown. Note that the label of excitations can depend on the framing. This particular choice is based on that in Fig. 16.

| $\zeta$ | $(C_{(1,1)}, \mathrm{Id}_{S_3})$ | $(C_{(1,r)}, \mathrm{Id}_{\mathbb{Z}_3})$ | $(C_{(1,s)}, \mathrm{Id}_{\mathbb{Z}_2})$ | $(C_{(1,s)}, \mathrm{Sign}_{\mathbb{Z}_2})$ |
|---|---|---|---|---|
| $N_\zeta(\text{trefoil})$ | 1 | 1 | 2 | 1 |
| $d_\zeta$ | 1 | 2 | 3 | 3 |

The solution to this problem can be converted to to a calculation in terms of the characters. The character of representation $\mathcal{H}_{(g,h)}$ is

$$\chi_{\mathcal{H}_{(g,h)}}(t) = \sum_{\text{allowed}(a,b)} \delta_{a,tat^{-1}} \delta_{b,tbt^{-1}}, \quad \forall t \in E_{(g,h)}. \tag{4.25}$$

The expression in Eq. (4.23) is then translated into

$$N_\zeta(\text{trefoil}) = \frac{1}{|E_{(g,h)}|} \sum_{t \in E_{(g,h)}} \chi_R^*(t) \chi_{\mathcal{H}_{(g,h)}}(t). \tag{4.26}$$

The solution to this problem for specific finite groups are summarized in Table 3 and Table 4.

# 5 Spiral map and beyond

The set of pure fluxes $\mathcal{C}_{\text{flux}}$ is in an elementary position of the 3d theory. While they may be understood, to some extent, from the dimension reduction picture, we argue that an intrinsic 3d view of them is worthwhile.

In this section, we introduce a class of *spiral maps*, which maps pure fluxes to pure fluxes. The spiral maps are of intrinsic 3d nature; they arise because a solid torus (embedded in a solid ball) has nontrivial subsystems spirals within it that are deformable to the original solid torus. We further discuss a few generalizations to broader contexts.

## 5.1 Spiral map

We shall begin with the definition of the spiral map $\mathcal{T}_n$ on the information convex set of the solid torus:

**Definition 5.1** (Spiral map). The $n$th spiral map is

$$\mathcal{T}_n : \Sigma(T) \to \Sigma(T), \tag{5.1}$$

where $T$ is a solid torus embedded in a ball and $n \in \mathbb{Z}$ is an integer. It is defined by the following two steps, illustrated in Fig. 17:

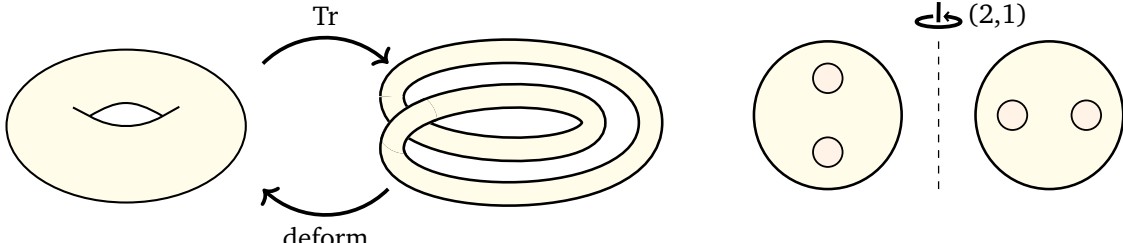

Figure 17: The steps involved in defining the spiral map. The case $\mathcal{T}_2$ is illustrated explicitly, while the strategy works for an arbitrary $\mathcal{T}_n$ for $n \neq 0$. The case $T_0$ is defined separately making use of the $(0,1)$ unknot, which is an unknot contained in a ball-shaped subsystem of the original solid torus.

1. Trace out the complement of a $(n,1)$ unknot contained in $T$.

2. Deform the $(n,1)$ unknot back to $T$ using the isomorphism theorem, along a particular path (fixed by convention) within the ball.

We are interested in the spiral map is that it has many nice properties:

1. (linearity) For any $n \in \mathbb{Z}$ and any collection of states $\rho_T^i \in \Sigma(T)$,

$$\mathcal{T}_n\left(\sum_i p_i \rho_T^i\right) = \sum_i p_i \mathcal{T}_n(\rho_T^i). \tag{5.2}$$

This is because each of the steps is linear in the input density matrix.

2. (preservation of vacuum) Let $\rho_T^1 \in \Sigma(T)$ be the vacuum sector, then

$$\mathcal{T}_n(\rho_T^1) = \rho_T^1, \quad \forall n \in \mathbb{Z}. \tag{5.3}$$

This is because every configuration along the deformation path is embedded in the ball on which the reference state is defined; the initial state $\rho_T^1$, after the partial trace, is (globally) consistent with the reference state. Thus any deformation step allowed by the isomorphism theorem cannot break this consistency.[24] Moreover,

$$\mathcal{T}_0(\lambda_T) = \rho_T^1, \quad \forall \lambda_T \in \Sigma(T). \tag{5.4}$$

3. (product rule) The spiral maps can be composed as:

$$\mathcal{T}_n \circ \mathcal{T}_m = \mathcal{T}_{mn}. \tag{5.5}$$

See Proposition 5.5.

4. (spiral map for fluxes) Extreme points are mapped to extreme points under the spiral map; see Corollary 5.6.1. Therefore, we can define the (induced) spiral map on pure flux labels:

$$t_n : \mathcal{C}_{\text{flux}} \to \mathcal{C}_{\text{flux}}, \quad \text{by} \quad \mathcal{T}_n(\rho_T^\mu) = \rho_T^{t_n(\mu)}. \tag{5.6}$$

Note that $t_n$ completely fixes the action of $\mathcal{T}_n$ by Eq. (5.2), and therefore it is an equivalent description of the spiral map.

---

[24]This argument uses embedding and thus it does not apply to paths containing immersed regions. It turns out that the statement remains true more generally, due to Proposition 5.4.

Table 5: The action of the first two nontrivial spiral maps on pure fluxes in the $\mathbb{Z}_2$ (left) and $S_3$ (right) quantum double models.

| $\mu$ | $C_1$ | $C_s$ |
|---|---|---|
| $t_2(\mu)$ | $C_1$ | $C_1$ |
| $t_3(\mu)$ | $C_1$ | $C_s$ |

| $\mu$ | $C_1$ | $C_r$ | $C_s$ |
|---|---|---|---|
| $t_2(\mu)$ | $C_1$ | $C_r$ | $C_1$ |
| $t_3(\mu)$ | $C_1$ | $C_1$ | $C_s$ |

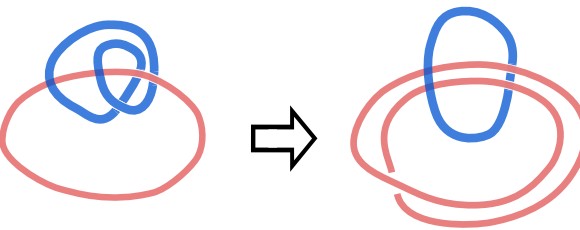

Figure 18: The first and last frames of the regular homotopy taking the region (blue) detecting the result of spiral fusion to the subregion of the solid torus defining the spiral map.

5. (monotonicity) The spiral map induces a monotonic decrease on the quantum dimension (and thus monotonic decrease of entropy of quantum states):

$$d_{t_n(\mu)} \leq d_\mu, \qquad \forall n \in \mathbb{Z}, \ \mu \in \mathcal{C}_{\text{flux}}. \tag{5.7}$$

See Proposition 5.8.

**Example 5.2.** For the quantum double model, the spiral map $t_n$ acts on the flux labelled $C_g$ (the conjugacy class with representative $g$) by $t_n : C_g \to C_{g^n}$.

So in the case of $G = \mathbb{Z}_2$ and $S_3$, the actions of $t_2$ and $t_3$ are given in Table 5.

### 5.1.1 Spiral fusion of fluxes

In §3.5 we introduced a notion of 'spiral fusion' of a pure flux loop. Here we use the spiral map to describe the outcome.

**Proposition 5.3.** *The process depicted in* (3.40) *takes* $\mu \in \mathcal{C}_{\text{flux}}$ *to* $t_n(\mu) \in \mathcal{C}_{\text{flux}}$.

*Proof.* This statement is best understood from Fig. 18. The idea is that the solid torus that detects the final flux can be deformed by regular homotopy to the $n$-fold-twisted solid torus defined in the first (partial trace) step of the spiral map definition. □

### 5.1.2 Proofs

**Proposition 5.4.** *Let* $\Omega$ *be an immersed region in a ball and* $\rho_\Omega \in \Sigma(\Omega)$. *Suppose* $K^0$ *and* $K^1$ *are two knots embedded in* $\Omega$, *that can be related by a path* $\{K^t\}_{t\in[0,1]}$ *immersed in* $\Omega$. *Then the following diagram commutes:*

$$
\begin{array}{ccc}
 & \rho_\Omega & \\
{}^{\text{Tr}_{\Omega\backslash K^0}}\swarrow & & \searrow^{\text{Tr}_{\Omega\backslash K^1}} \\
\rho_{K^0} & \underset{\Phi_{\{K^{1-t}\}}}{\overset{\Phi_{\{K^t\}}}{\rightleftarrows}} & \rho_{K^1}
\end{array}
\tag{5.8}
$$

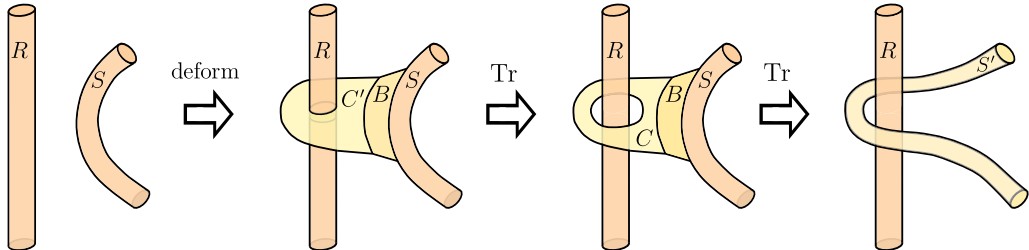

Figure 19: Illustrated is the strategy used in the proof for a basic move. All regions are within $\Omega$. The basic move is to deform $S$ to the left to obtain $S'$, where $RS = K^0$ and $RS' = K^1$. (Part of $R$ is not shown.) The first step is to enlarge the knot $K^0$ into immersed region $K^+ \equiv K^0 BC'$, where $BC'$, $B$, and $C'$ are disks, respectively. Here we require $K^+$ to contain $K^1$. The second step is to do a partial trace $K^+ \to K^0 BC$, where $C$ is an annulus and $K^0 BC$ is a subsystem containing $K^1$. The final step is to do a partial trace to reduce $BCS \to S'$ and thus $K^0 BC \to K^1$.

where $\Phi_{\{K^t\}} : \Sigma(K^0) \to \Sigma(K^1)$ *is the isomorphism associated with the path* $\{K^t\}$.

**Remark.** The application of this proposition is broad because it does not require $\Omega$ to be embedded in a ball. The proof does make use of a special property, namely that a knot looks like a solid cylinder locally (see Fig. 19). Pushing the sphere shell through itself is trickier to analyze. For example, we do not know if, in general, a *sphere eversion* [46–48] which turns a sphere shell inside out, takes the vacuum sector $1 \in \mathcal{C}_{\text{point}}$ back to itself.

*Proof.* Without loss of generality, we shall consider $K^0$ and $K^1$ that are related by the basic move described below. (More general cases are proved by applying the same strategy multiple times.) The basic move is to pass the solid torus through itself smoothly via a sequence of regions immersed in $\Omega$; in the depiction Fig. 19, the basic move is to deform $S \subset K^0$ smoothly to the left and obtain $S' \subset K^1$.

The strategy of the proof is illustrated in Fig. 19. Let the state on immersed region $K^+ = K^0 BC'$, obtained by an extension of $\rho_{K^0}$, as $\lambda_{K^+}$. It is easy to see that the both $\rho_{K^0}$ and $\Phi_{\{K^t\}}(\rho_{K^0})$ are consistent with $\lambda_{K^+}$. We want to further show $\Phi_{\{K^t\}}(\rho_{K^0}) = \rho_{K^1}$. This follows from two quantum Markov chain conditions.

First, for the state $\lambda$, we have

$$I(C : K^0 | B)_\lambda \leq I(C' : K^0 | B)_\lambda = 0 \,. \tag{5.9}$$

Second, for the state $\rho_\Omega \in \Sigma(\Omega)$, we have

$$I(C : K^0 | B)_\rho \leq (S_{BC} + S_{CD} - S_B - S_D)_\rho = 0 \,, \tag{5.10}$$

where $BD$ is a torus shell which covers the boundary of $C$ such that $D \cap K^0 = \emptyset$.

(Here, we have used the following result: Consider a partition of a solid torus $T$ into three regions $BCD$; see Fig. 20. Here $C$ is $T$ minus its thickened boundary, $BD$. $B$ is a ball connecting $C$ to the complement of $T$. Then, for the reference state $\sigma$, $(S_{BC} + S_{CD} - S_B - S_D)_\sigma = 0$. This statement is proved in Appendix D; see Proposition D.6.)

So, both $\rho$ and $\lambda$ are quantum Markov states with respect to the partition $C, B, K^0$. Moreover, they have the same marginals on $BC$ and $BK^0$. Therefore, $\rho_{BCK^0} = \lambda_{BCK^0}$. This implies the desired property and completes the proof. $\square$

**Corollary 5.4.1.** *The same map $\mathcal{T}_n$ is obtained if we replace the $(n, 1)$ unknot with any knot that can be deformed to the $(n, 1)$ unknot through a path within the solid torus.*

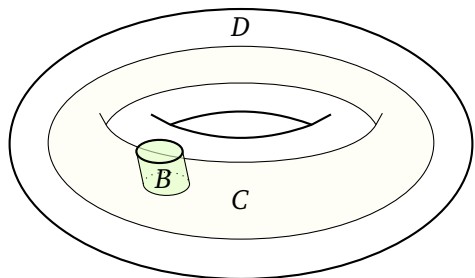

Figure 20: A partition of the solid torus, $T = BCD$.

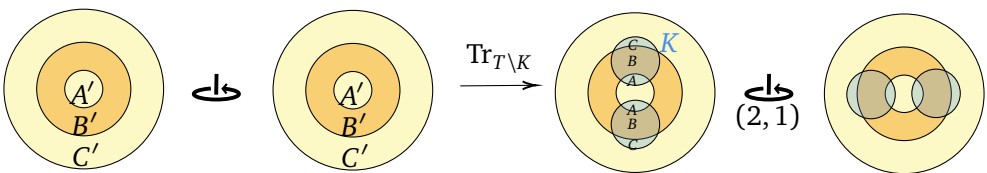

Figure 21: A partition of the solid torus $T$ and its intersection with a thickened torus knot $K$ contained in $T$. The case where $K$ is a $(2, 1)$ unknot is shown.

This implies, for example, that we obtain the same map if we use the $(n, p)$ torus knot instead of $(n, 1)$ torus knot, where $n$ and $p$ are relatively prime.

*Proof.* This follows from Proposition 5.4 directly. □

**Proposition 5.5** (Product rule)**.** *For any $m, n \in \mathbb{Z}$,*

$$\mathcal{T}_n \circ \mathcal{T}_m = \mathcal{T}_{mn}. \tag{5.11}$$

*Proof.* When both $m$ and $n$ are nonzero, this follows from Corollary 5.4.1 because the knot that is associated with the left-hand side of Eq. (5.5) is a satellite (un)knot of the $(n, 1)$ unknot (by a $(m, 1)$), and it is deformable to the $(mn, 1)$ unknot through a path within the solid torus.

When either $m$ or $n$ equals to 0, the relation holds. This is because the map defined on either side of the equation maps $\lambda_T \in \Sigma(T)$ to the vacuum sector $\rho_T^1$. □

**Proposition 5.6.** *Tracing out the complement of any knot or link $K$ contained in the solid torus $T$ maps extreme points of $\Sigma(T)$ to extreme points of $\Sigma(K)$.*

**Remark.** The scope of this statement goes beyond the context of the spiral map because we do not need the solid torus to be contained in a ball. We shall provide two proofs based on different ideas.

*Proof.* (The 1st proof.) Solid torus $T$ is a sectorizable region. Therefore, the desired answer follows from Proposition 2.23. □

*Proof.* (The 2nd proof.) The strategy is to subdivide the solid torus into three concentric regions, ($T = A'B'C'$ as in Fig. 21), by which we shall provide a proof for torus knots. The case of more general knots and links then follows from Proposition 5.4.

First, by the factorization property of extreme points, $I(A' : C')_\rho = 0$ if $\rho_T$ is an extreme point of $\Sigma(T)$. This implies, by SSA, that for any subsets $A \subset A', C \subset C'$ $I(A : C)_\rho = 0$ as well.

Now consider a torus knot. By definition, it can be embedded in an unknotted torus surface. Choose the torus to lie within $B'$ and let the thickened torus knot be $K$; see Fig. 21. We require $K$ to be thick enough so that $A = A' \cap K, C = C' \cap K$ are of the same topology as $K$ (namely,

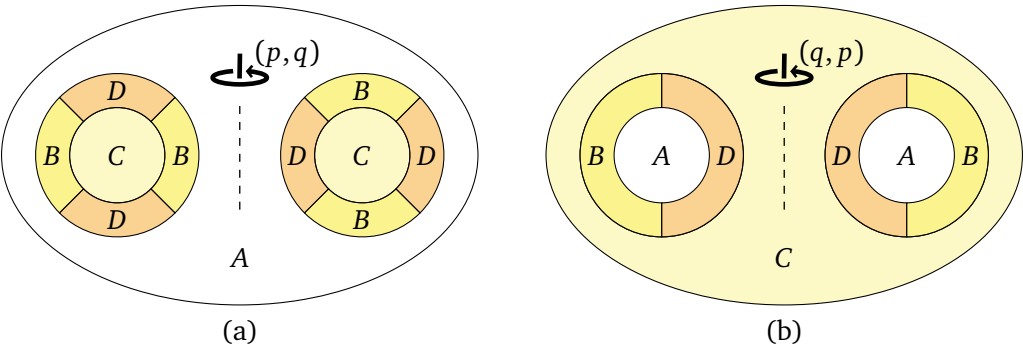

Figure 22: Solid torus $T = BCD$ is a subsystem of a ball, where $BD$ forms the thickened boundary of $T$ with rotation type $(p,q)$. Here $(p,q)$ are coprime integers. Completion on a sphere $S^3 = ABCD$ allows us to consider a pure reference state $|\psi_{S^3}\rangle$ and a "dual" solid torus $\widetilde{T} = BAD$, where the rotation type, show in (b) is $(q,p)$. The illustration is accurate when $(p,q) = (2,1)$.

they can be deformed back to $K$ by a sequence of extensions). We conclude that the reduced density matrix $\rho_K$ is an extreme point of $\Sigma(K)$; this is because $I(A:C)_\rho = 0$ is a necessary and sufficient condition for $\rho_K$ to be an extreme point of $\Sigma(K)$. □

**Corollary 5.6.1.** *The spiral map ($\mathcal{T}_n$, $\forall n \in \mathbb{Z}_{>0}$) maps extreme points to extreme points.*

*Proof.* Looking back at the definition of the spiral map (Definition 5.1), the only danger is the step involving the partial trace. This step is safe due to Proposition 5.6. □

**Lemma 5.7.** *Consider a solid torus $T$ that is a subsystem of a ball. The reference state of the ball is $\sigma$. Consider partition $T = BCD$, according to Fig. 22(a). Then*

$$(S_{BC} + S_{CD} - S_B - S_D)_\sigma = 0, \quad if \tag{5.12}$$

1. $(p,q) = (1,q)$ *for any integer $q$;*

2. $(p,q) = (p,1)$ *for any integer $p$.*

**Remark.** The basic intuition is that the $(1,q)$ and $(p,1)$ cases are dual to each other. This is why they can be solved simultaneously. In fact, the statement is true for all coprime $(p,q)$; the proof is more subtle, and it is presented in the appendix (Proposition D.7).

*Proof.* Because of the sphere completion lemma (Lemma 3.1), we only need to prove the statement (5.12) for the case where the solid torus is embedded in $S^3$. This case is illustrated in Fig. 22(a), where $A$ is the complement of the solid torus $BCD$ on $S^3$. Let $|\psi_{S^3}\rangle$ be the global reference state on $S^3$.

Because $|\psi_{S^3}\rangle$ is pure, Eq. (5.12) is equivalent to the statement

$$I(A:C|B)_{|\psi_{S^3}\rangle} = 0. \tag{5.13}$$

There are two views of the same subsystems, related by a rotation of the 3-sphere, namely Fig. 22(a) and its "dual" view Fig. 22(b).

When $(p,q) = (1,q)$, there is a way to smoothly deform $B$ to $BC$ by a sequence of enlargements (as in the proof of Lemma D.2 of [1]). When $(p,q) = (p,1)$, instead, there is a way to smoothly deform $B$ to $AB$ by a sequence of enlargements. This proves Eq. (5.13). The details of the enlargement are reviewed in the next paragraph for $B \to BA$ of the case $(p,q) = (p,1)$.

Consider a sequence of subsystems $\{BA_i\}_{i=1}^{M}$ where $A_0 = \emptyset$ and $A_M = A$. For any $i \in \{0, 1, \cdots M-1\}$, $\delta A_{i+1} \equiv A_{i+1} \setminus A_i$ is a small ball attached to $BA_i$ in a way that does not change the topology. Then for all $i \in \{0, 1, \cdots M-1\}$,

$$(S_{CBA_i} - S_{BA_i})_{|\psi_{S^3}\rangle} = (S_{CBA_{i+1}} - S_{BA_{i+1}})_{|\psi_{S^3}\rangle}. \tag{5.14}$$

Therefore $I(A:C|B)_{|\psi_{S^3}\rangle} = 0$. This completes the proof. $\qquad\square$

**Proposition 5.8** (Monotonicity of the spiral map). *For any pure flux sector $\mu \in \mathcal{C}_{\text{flux}}$ and any positive integer $p$,*

$$d_\mu \geq d_{t_p(\mu)}. \tag{5.15}$$

*Proof.* Consider a solid torus $T$ embedded in a ball. SSA says that for any state on $BCD$, the combination of entropies $(S_{BC} + S_{CD} - S_B - S_D) \geq 0$. By applying this result to a partition $T = BCD$ of type $(p, 1)$, as is shown in Fig. 22, and using the fact that $(S_{BC} + S_{CD} - S_B - S_D)_\sigma = 0$ for this partition (Lemma 5.7), we arrive at

$$\begin{aligned} 0 \leq{}& (S_{BC} + S_{CD} - S_B - S_D)_{\rho^\mu} - (S_{BC} + S_{CD} - S_B - S_D)_\sigma \\ ={}& \left(2\ln d_\mu + 2\ln d_\mu - 2\ln d_{t_p(\mu)} - 2\ln d_{t_p(\mu)}\right) \\ ={}& 4\left(\ln d_\mu - \ln d_{t_p(\mu)}\right). \end{aligned} \tag{5.16}$$

Here $\mu \in \mathcal{C}_{\text{flux}}$ and $\rho_T^\mu$ is an extreme point of $\Sigma(T)$. To arrive at the second line, we have used the fact that the reduced density matrices of $\rho_T^\mu$ on $B$ and $D$ each contribute $2\ln d_{t_p(\mu)}$ entropy difference with the reference state. Therefore, $d_\mu \geq d_{t_p(\mu)}$ and this completes the proof. $\qquad\square$

## 5.2 Generalizations

A similar class of maps can be generalized to other regions. We provide two such examples.

The first generalization is to a genus $g$ handlebody $X_g$. We can define a map from $\Sigma(X_g)$ to itself. A first step is to take the trace over $X \setminus X_-$ in this figure (for $g = 2$):

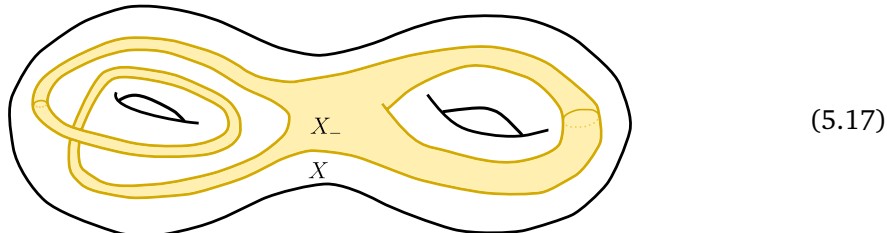

$$\tag{5.17}$$

Then deform $X_-$ along a specific path back to $X$. This map takes extreme points to extreme points by Proposition 2.23.

The second generalization involves the torus shell $\mathbb{T}$; we require the torus shell to be embedded in a ball. (See Fig. 23 for the illustration of the idea.) For each pair of coprime positive integers $(p, q)$ we define

$$\mathcal{T}_{(p,q)} : \Sigma(\mathbb{T}) \to \Sigma(T) \tag{5.18}$$

by:

1. Trace out the complement of a thickened $(p, q)$ torus knot contained within the torus shell $\mathbb{T}$.

2. Use the generalized isomorphism theorem to deform the knot to a reference solid torus $T$.



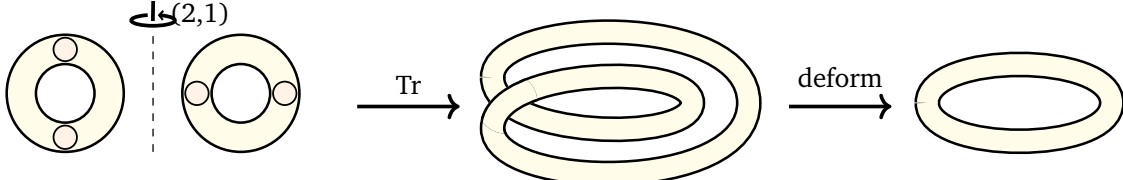

Figure 23: The idea behind the definition of $\mathcal{T}_{(p,q)}$. Being illustrated is $(p,q) = (2,1)$.

Similar to the spiral maps, $\mathcal{T}_{(p,q)}$ maps extreme points to extreme points (by Proposition 2.23). Thus, it induces a map

$$t_{(p,q)} : \mathcal{C}_{\text{Hopf}} \to \mathcal{C}_{\text{flux}}, \quad \text{according to} \quad \mathcal{T}_{(p,q)}(\rho_{\mathbb{T}}^{\eta}) = \rho_T^{t_{(p,q)}(\eta)}. \tag{5.19}$$

$t_{(p,q)}$ maps the vacuum sector to the vacuum sector. (Note that, $\mathcal{T}_{(p,q)}$ cannot be composed because it maps a torus shell to a solid torus.)

**Remark.** In defining each of these spiral maps, we have chosen a particular path from the outcome of the partial trace to the final reference region. It will be interesting to study the space of ambiguities avoided by this arbitrary choice, namely the homotopy groups of the space of such deformation paths.

# 6 Consistency Relations

In this section, we study the consistency relations among the fusion data of various 3d regions. The focus is on the consistency relation for the knot multiplicities of torus knots because they are simple and escape (known) dimensional reduction descriptions. (Many other consistency relations of 3d data can be derived by applying the dimensional reduction technique we developed. These relations and a few that are beyond dimensional reduction can be found in Appendix E.) These consistency relations come from a standard entanglement bootstrap analysis: computing the entropy of a certain maximum-entropy state of an information convex set in two different ways and comparing them. The relations we identify below are of two types. Conditions of the first type follow directly from the associativity theorem 2.22; these are consistency relations among multiplicities. Relations of the second type involve both multiplicities and quantum dimensions.

## 6.1 Associativity constraints

The associativity theorem 2.22 is a powerful statement. It indicates that the fusion spaces of a few simple regions can be used as "building blocks" to construct more complex fusion spaces. In 3d, we do not know the complete list of such building blocks. Nonetheless, the fusion space associated with a large class of regions can be obtained this way. One example is:

**Proposition 6.1.** *Let $\Omega(m,n)$ be a ball with $m$ balls removed and $n$ (unlinked) unknots removed. The fusion multiplicities associated with $\Omega(m,n)$ are $\{N_{b_1\cdots b_m l_1\cdots l_n}^a\}$ with $a, b_1, \cdots, b_m \in \mathcal{C}_{\text{point}}$ and $l_1, \cdots, l_n \in \mathcal{C}_{\text{loop}}$. These multiplicities are determined by two sets of basic multiplicities: $\{N_{ab}^c\}_{a,b,c\in\mathcal{C}_{\text{point}}}$, the multiplicities for $\Omega(2,0)$ and $\{N_l^a\}_{l\in\mathcal{C}_{\text{loop}}}$, the multiplicities for $\Omega(0,1)$.*

*Proof.* First, the multiplicities associated with $\Omega(m+n,0)$ are determined entirely by $\{N_{ab}^c\}$. This is because one can glue a boundary of $\Omega(k-1,0)$ and $\Omega(2,0)$ to make a $\Omega(k,0)$. Next, we can glue $\Omega(k+1,l)$ with $\Omega(0,1)$ to obtain $\Omega(k,l+1)$. For example, for $k=0$, we get

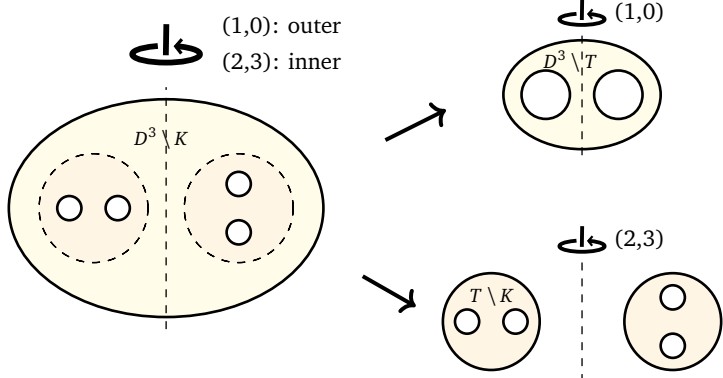

Figure 24: A partition of a ball minus a torus knot ($D^3 \setminus K$) for the associativity proof. Here $K$ is a $(p,q)$ torus knot. $(p,q) = (2,3)$ is illustrated accurately while the same idea works for general $(p,q)$.

$N^a_{l_1 l_2} = \sum_b N^a_{b l_1} N^b_{l_2}$. By induction, we see that the multiplicities of $\Omega(m,n)$ are determined by $\{N^c_{ab}\}_{a,b,c \in \mathcal{C}_{\mathrm{point}}}$ and $\{N^a_l\}_{l \in \mathcal{C}_{\mathrm{loop}}}$. A precise formula can be worked out for the general case, but we omit it. $\qquad\square$

**Remark.** This statement can be understood from the dimensional reduction picture. The idea is that balls and unlinked unknots can be arranged along a line segment. The whole region $\Omega(m,n)$ then becomes the revolution of a 2d region with a boundary.

**Proposition 6.2.** *Let $K$ be a $(p,q)$ torus knot. Then, the knot multiplicities satisfy:*

$$N_\zeta(K) = \sum_{\mu \in \mathcal{C}_{\mathrm{flux}}} N_\zeta^{\varphi(\mu)}(p,q), \quad \forall \zeta \in \mathcal{C}_{\mathrm{Hopf}}. \tag{6.1}$$

*Here, $\{N_\zeta^{\varphi(\mu)}(p,q)\}$ is a subset of the multiplicities defined in Eq. (3.31). More generally, the multiplicities defined in Eq. (3.28) satisfy*

$$N_\zeta^a(K) = \sum_{l \in \mathcal{C}_{\mathrm{loop}}} N_\zeta^l(p,q) N_l^a, \qquad \forall \zeta \in \mathcal{C}_{\mathrm{Hopf}}, \quad \forall a \in \mathcal{C}_{\mathrm{point}}. \tag{6.2}$$

*Proof.* We first derive Eq. (6.2). Let $D^3 \supset T \supset K$, where $D^3$ is a ball, $T$ is a solid torus. $K$ is the $(p,q)$ torus knot that is described by a $(p,q)$ type rotation along the same axis that defines $T$ and $D^3$. The boundaries of these three regions do not touch each other, as is illustrated in Fig. 24. Thus $D^3 \setminus K$ is divided into halves by the boundary of $T$. By the associativity theorem,

$$N_\zeta^a(K) = \sum_{\eta \in \mathcal{C}_{\mathrm{Hopf}}} N_\zeta^\eta(p,q) N_\eta^a, \quad \forall a \in \mathcal{C}_{\mathrm{point}}.$$

Only shrinkable loops contribute because $N_\eta^a = 0$ if $\eta \notin \mathcal{C}_{\mathrm{loop}}$. Therefore, we arrive at Eq. (6.2). To derive Eq. (6.1), we recall that $N_l^1 = \sum_\mu \delta_{l,\varphi(\mu)}$ and that $N_\zeta^1(K) = N_\zeta(K)$. We plug these in Eq. (6.2). $\qquad\square$

**Remark.** In fact, this idea can be extended to provide insight into a certain type of satellite knots. Let $K$ be the $(p,q)$ torus knot discussed in Fig. 24. Let $K' \subset K$ be a knot such that $K \setminus K'$ can be deformed to $T \setminus K$ by a path (formed by immersed regions), under which process $K'$ becomes $K$. It is easy to see $K'$ is a (special type of) satellite knot. The set of regions $K' \subset K \subset T \subset D^3$ is useful for deriving a consistency relation.

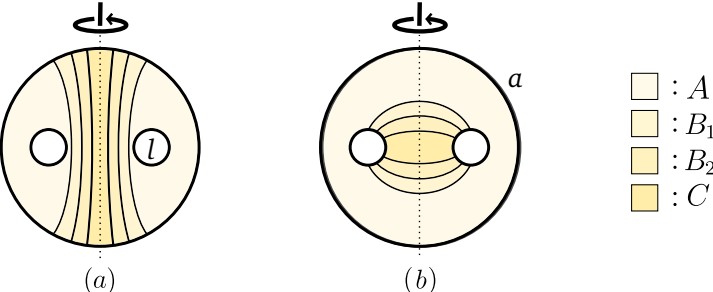

Figure 25: Make a ball minus unknot ($D^3 \setminus T$) by two different merging processes. (a) The newly formed boundary is the sphere boundary. We have the freedom to choose $l \in \mathcal{C}_{\text{loop}}$. (b) The newly formed boundary is the torus boundary. We have the freedom to choose $a \in \mathcal{C}_{\text{point}}$.

## 6.2 Consistency relations on shrinking rules

Below, we study a few consistency relations related to the multiplicities $\{N_l^a\}$ associated with the shrinking rule. The region in question is the ball minus a solid torus $D^3 \setminus T$. While it is not impossible to continue the analysis of cutting $D^3 \setminus T$ by a hypersurface contained in the interior of $D^3 \setminus T$, there are simpler partitions that involve cuts that intersect the boundary. We prove two such conditions. A general feature is that when the cut intersects with the boundary, quantum dimensions appear in the consistency relation. We further establish an embedding $\phi : \mathcal{C}_{\text{point}} \hookrightarrow \mathcal{C}_{\text{loop}}$ which preserves the quantum dimension.

### 6.2.1 Consistency with quantum dimensions

The two consistency relations we discuss here can be understood by dimensional reduction (Appendix E, Proposition E.1), which maps them to a 2d consistency in the presence of a gapped boundary. Nonetheless, the 3d analysis is simple and pleasant. We present them for pedagogical purposes.

**Proposition 6.3** (for shrinking rules)**.**

$$d_l = \sum_{a \in \mathcal{C}_{\text{point}}} N_l^a d_a, \qquad \forall l \in \mathcal{C}_{\text{loop}}. \tag{6.3}$$

$$d_a = \frac{1}{\mathcal{D}^2} \sum_{l \in \mathcal{C}_{\text{loop}}} N_l^a d_l, \quad \forall a \in \mathcal{C}_{\text{point}}. \tag{6.4}$$

*Proof.* We first prove Eq. (6.3). Consider the partition shown in Fig. 25(a). $D^3 \setminus T = ABC$, where $B = B_1 B_2$. For each choice of $l \in \mathcal{C}_{\text{loop}}$, there is a well-defined merging process. Let the merged state be $\tau_{ABC}^{\star, l}$. Note that $\tau_{ABC}^{\star, 1} = \sigma_{ABC}$. (In other words, $I(A : C|B)_\sigma = 0$ for the partition in Fig. 25(a). For reader's convenience, we review this fact in Lemma 6.4.) The entropy different $S(\tau_{ABC}^{\star, l}) - S(\sigma_{ABC})$ can be expressed in two different ways. First, by analyzing the structure of $\Sigma(D^3 \setminus T)$ we have $S(\tau_{ABC}^{\star, l}) - S(\sigma_{ABC}) = \ln(\sum_a N_l^a d_a) + \ln d_l$. Second, by considering the entropy difference of the marginals, we derive $S(\tau_{ABC}^{\star, l}) - S(\sigma_{ABC}) = 2 \ln d_l$. Therefore, $d_l = \sum_a N_l^a d_a$, which is precisely Eq. (6.3).

Next, we prove Eq. (6.4). Now we choose an extreme point of the sphere shell, labeled by $a \in \mathcal{C}_{\text{point}}$, and consider the merging process in Fig. 25(b). Let the merged state be $\lambda_{ABC}^{\star, a}$. This time, note that $\lambda_{ABC}^{\star, 1}$ is *not* the same with $\sigma_{ABC}$, although its marginals on $AB$ and $BC$

are identical with that of the reference state. (The reason is that $I(A : C|B)_\sigma = 2\ln\mathcal{D}$ by Lemma 6.4 while $I(A : C|B)_{\lambda^{\star,a}} = 0$ for any $a$.) Again, we have two ways to compute the entropy difference $S(\lambda^{\star,a}_{ABC}) - S(\sigma_{ABC})$. By analyzing the structure of $\Sigma(D^3 \setminus T)$ we have $S(\lambda^{\star,a}_{ABC}) - S(\sigma_{ABC}) = \ln(\sum_l N^a_l d_l) + \ln d_a$. On the other hand, by looking at the marginals we see $S(\lambda^{\star,a}_{ABC}) - S(\lambda^{\star,1}_{ABC}) = 2\ln d_a$ and that $S(\lambda^{\star,1}_{ABC}) - S(\sigma_{ABC}) = 2\ln\mathcal{D}$. This leads to the constraint $\ln d_a + \ln(\sum_l N^a_l d_l) = 2\ln d_a + \ln\mathcal{D}^2$. By simplifying this, we get Eq. (6.4). $\square$

**Remark.** It is interesting to observe the existence of an extra $1/\mathcal{D}^2$ factor on the right-hand side of the second equation. This makes shrinkable loops and point particles manifestly different. This contribution comes from a factor of the form $e^{-I(A:C|B)_\sigma}$. This contribution will appear in a broader context in [20].

**Lemma 6.4.** *For the ABC partition in Fig. 25(a), $I(A : C|B)_\sigma = 0$. For the ABC partition in Fig. 25(b), $I(A : C|B)_\sigma = 2\ln\mathcal{D}$.*

*Proof.* Consider the *ABC* partition of $D^3 \setminus T$ in Fig. 25(a). $I(A : C|B)_\sigma \le I(AT : C|B)_\sigma$ by SSA. Next, we show $I(AT : C|B)_\sigma = 0$. We provide two independent ways to see it. Here is the first method. Suppose $I(AT : C|B)_\sigma > 0$; we can take the marginals $\sigma_{TAB}$ and $\sigma_{BC}$ and merge them; the resulting state has $I(AT : C|B) = 0$ and it is an element of $\Sigma(D^3)$. However, this would contradict with the fact that $\Sigma(D^3)$ has a unique element. The second method is to consider Fig. 26:

$$I(AT : C|B)_\sigma \le \Delta(B, AT, E)_\sigma \tag{6.5}$$
$$= 0.$$

The first line follows from SSA, and the second line follows from Proposition 5.7.

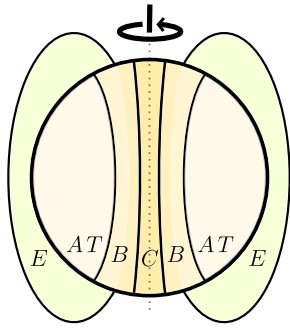

Figure 26: With a region $E$.

For the *ABC* partition of $D^3 \setminus T$ in Fig. 25(b), $I(A : C|B)_\sigma$ is the topological entanglement entropy $2\gamma$; see Proposition D.11, the fourth case. According to the analysis that is done around Fig. 10, we have $\gamma = \ln\mathcal{D}$. Therefore, $I(A : C|B)_\sigma = 2\ln\mathcal{D}$. $\square$

### 6.2.2 Embedding $\phi : \mathcal{C}_{\text{point}} \hookrightarrow \mathcal{C}_{\text{loop}}$

We establish a natural embedding of the set of point particles to the set of shrinkable loops.

Consider the arrangement of regions in Fig. 27. Here $A$ is a ball minus an unknot and $AB$ is a sphere shell. Because $AB$ is a sphere shell, the extreme points of $\Sigma(AB)$ are labeled by $a \in \mathcal{C}_{\text{point}}$. By a partial trace, we can reduce any extreme point $\rho^a_{AB} \in \Sigma(AB)$ to $A$. According to Proposition 2.23, the result is an isolated extreme point, namely

$$\rho^a_{AB} \overset{\text{Tr}_B}{\to} \rho^a_A \in \Sigma^a_{\phi(a)}(A), \tag{6.6}$$

where $\phi(a) \in \mathcal{C}_{\text{loop}}$, defined by this process, satisfies $N^a_{\phi(a)} = 1$.

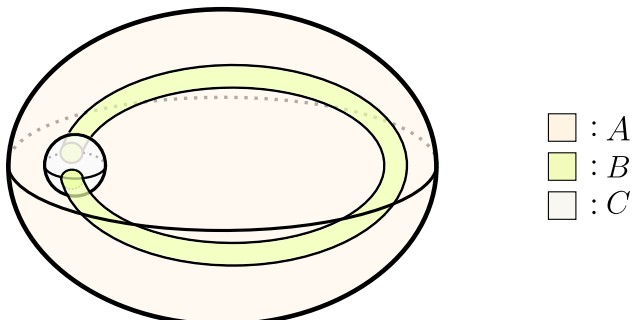

Figure 27: Identify a ball minus an unknot ($D^3 \setminus T$) as a subsystem of a sphere shell. Here $ABC = D^3$ is a ball, $B$ and $C$ are two balls and $BC = T$ is an unknotted solid torus. $A = D^3 \setminus T$ is a ball minus an unknot, and $AB$ is a sphere shell.

**Proposition 6.5.** *The map $\phi : \mathcal{C}_{\text{point}} \to \mathcal{C}_{\text{loop}}$ defined by Eq. (6.6) is injective. Furthermore,*

$$d_{\phi(a)} = d_a, \quad N_{\phi(a)}^b = \delta_{a,b}. \tag{6.7}$$

For this reason, we shall call $\phi$ an embedding, and denote it as $\phi : \mathcal{C}_{\text{point}} \hookrightarrow \mathcal{C}_{\text{loop}}$.

*Proof.* For the partition in Fig. 27, we consider an extreme point $\rho_{AB}^a$. Not only can we take a partial trace to obtain a state in $\Sigma(A)$, we can also reverse the partial trace by a quantum channel (associated with an appropriate merging) that is independent of the sector $a \in \mathcal{C}_{\text{point}}$. Thus, the entropy difference is preserved: $\ln d_a + \ln d_{\phi(a)} = 2 \ln d_a$. Therefore, $d_{\phi(a)} = d_a$. Plugging this into Eq. (6.3) we see that $d_a = \sum_b N_{\phi(a)}^b d_b$. Noticing that multiplicities and quantum dimensions are non-negative (and $N_{\phi(a)}^a = 1$), we see that $N_{\phi(a)}^b = \delta_{a,b}$. $\qquad\square$

**Remark.** Does the same logic work when the union of the two balls $B$ and $C$ is a knot, instead of the unknot shown in Fig. 27? In that case, we can identify a map

$$\widetilde{\phi}_K : \mathcal{C}_{\text{point}} \to \mathcal{C}_{\text{Hopf}}, \tag{6.8}$$

which depends on the choice of knot $K$. Furthermore,

$$d_{\widetilde{\phi}_K(a)} = d_a, \quad N_{\widetilde{\phi}_K(a)}^a(K) = 1. \tag{6.9}$$

This observation is consistent with the data shown in Table 8. The analog of the second equation in (6.7) would require the analog of (6.3) for knots, the naive version of which is not true. Nevertheless, we expect that $\widetilde{\phi}_K$ is an embedding for general knots $K$.

## 6.3 Torus knots: a constraint with quantum dimensions

We shall consider a $(p,q)$ torus knot $K \subset S^3$. We derive a consistency relation that involves the knot multiplicities, the quantum dimensions, and the spiral maps. As a corollary, we derive a universal upper bound for the knot multiplicities for all torus knots in terms of the total quantum dimension.

**Proposition 6.6.** *Let $K$ be a $(p,q)$ torus knot. Then*

$$\boxed{\sum_{\zeta \in \mathcal{C}_{\text{Hopf}}} N_\zeta(K) d_\zeta = \sum_{\substack{\mu,\nu \in \mathcal{C}_{\text{flux}}:\\ t_p(\mu)=t_q(\nu)}} \frac{d_\mu^2 d_\nu^2}{d_{t_p(\mu)}^2}.} \tag{6.10}$$

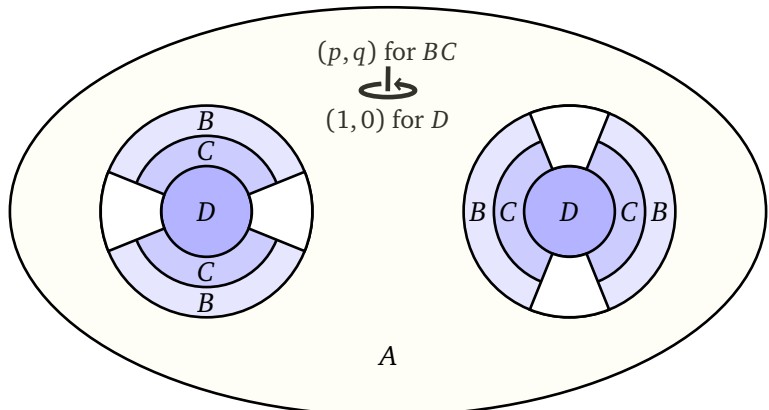

Figure 28: A decomposition of the knot complement of a $(p,q)$ torus knot. $S^3 \setminus K = ABCD$. $(p,q) = (2,3)$ is illustrated accurately. Note that $A$ is the solid torus such that the rotation axis is contained entirely in $A$ (although this is not obvious in a 2d Euclidean illustration).

*Proof.* Consider the decomposition of the complement of a $(p,q)$ torus knot, $\Omega_K \equiv S^3 \setminus K$ depicted in Fig. 28. $\Omega_K = ABCD$, where $ABC$ and $BCD$ are both solid tori. The maximum entropy state $\rho^\star_{\Omega_K}$ of the information convex set $\Sigma(\Omega_K)$ and the reference state on $S^3$ ($\sigma$) both have $I(A : D|BC) = 0$. To see that the former statement is true, we observe that if $\rho^\star_{ABCD}$ were not a quantum Markov state, we would be able to obtain a state with higher entropy by merging the marginals back. The latter case follows from Proposition D.7.

Below we compute the entropy difference $S(\rho^\star_{\Omega_K}) - S(\sigma_{\Omega_K})$ in two different ways.

1. First, by the structure theorem of $\Sigma(\Omega_K)$:

$$S(\rho^\star_{\Omega_K}) - S(\sigma_{\Omega_K}) = \ln \left( \sum_{\zeta \in \mathcal{C}_{\text{Hopf}}} N_\zeta(K) d_\zeta \right). \tag{6.11}$$

2. Second, we use the quantum Markov state condition of $\rho^\star$ and solve a maximization problem on the marginals $ABC$, $BC$ and $BCD$. The final answer is

$$S(\rho^\star_{\Omega_K}) - S(\sigma_{\Omega_K}) = \sum_{\substack{\mu,\nu \in \mathcal{C}_{\text{flux}}: \\ t_p(\mu) = t_q(\nu)}} \frac{d_\mu^2 d_\nu^2}{d_{t_p(\mu)}^2}. \tag{6.12}$$

By comparing these two equations, one derives the desired result. The rest of the proof is devoted to the proof of Eq. (6.12).

To prove Eq. (6.12), we first consider what the spiral map does on probability distribution $\{p_\mu\}_{\mu \in \mathcal{C}_{\text{flux}}}$. Let us define $f_p(\{p_\mu\}) \equiv \{r_\mu\}_{\mu \in \mathcal{C}_{\text{flux}}}$, where

$$r_\nu = \sum_{\mu : t_p(\mu) = \nu} p_\mu. \tag{6.13}$$

We further denote $F(\{p_\mu\}) \equiv \sum_\mu p_\mu \ln \left( \frac{d_\mu^2}{p_\mu} \right)$.

The problem is to solve

$$\max_{\substack{\{p_\mu\}, \{q_\nu\}: \\ f_p(\{p_\mu\}) = f_q(\{q_\nu\})}} \left( F(\{p_\mu\}) + F(\{q_\nu\}) - F(f_p(\{p_\mu\})) \right). \tag{6.14}$$

This is because this is the general expression for the entropy difference $S(\lambda_{\Omega_K}) - S(\sigma_{\Omega_K})$ for any state $\lambda$ obtained by merging states in $\Sigma(ABC)$ and $\Sigma(BCD)$. (Here, the two states are $\sum_\mu p_\mu \rho_{ABC}^\mu$ and $\sum_\nu q_\nu \rho_{BCD}^\nu$ respectively.) The condition $f_p(\{p_\mu\}) = f_q(\{q_\nu\})$ is the necessary and sufficient condition for two states be merged. The solution leads to Eq. (6.12). □

**Example 6.7.** Here we check the equality for trefoil knot, $(p, q) = (2, 3)$.

1. For the 3d toric code model, the prediction from (3.26) is $\sum_\zeta N_\zeta(K) = 2$. This is consistent with the action of the spiral map, since only $(\mu, \nu) = (C_1, C_1)$ and $(C_s, C_1)$ contribute to the sum.

2. For the 3d $S_3$ quantum double model $\sum_\zeta N_\zeta(K) d_\zeta = 12$ from the data shown in (3.27). This is consistent with the value in terms of the spiral map from Table 5.

**Corollary 6.7.1** (Universal bound for torus knots). *For any torus knot $K$:*

$$\boxed{\sum_{\zeta \in \mathcal{C}_{\text{Hopf}}} N_\zeta(K) d_\zeta \leq \mathcal{D}^4.} \tag{6.15}$$

*Proof.*

$$\sum_{\substack{\mu, \nu \in \mathcal{C}_{\text{flux}}: \\ t_p(\mu) = t_q(\nu)}} \frac{d_\mu^2 d_\nu^2}{d_{t_p(\mu)}^2} \leq \sum_{\mu, \nu \in \mathcal{C}_{\text{flux}}} \frac{d_\mu^2 d_\nu^2}{d_{t_p(\mu)}^2} \leq \sum_{\mu, \nu \in \mathcal{C}_{\text{flux}}} d_\mu^2 d_\nu^2 = \mathcal{D}^4. \tag{6.16}$$

□

**Remark.** The upper bound says that the knot multiplicity for any torus knot cannot be too large for a given value of $\mathcal{D}$. (This is because $d_\zeta \geq 1$; see Appendix E.) We do not know if this universal bound can be improved further, i.e., replacing $\mathcal{D}^4$ by $\mathcal{D}^\alpha$ with $\alpha < 4$. Because $\gamma = \ln \mathcal{D}$ is the topological entanglement entropy of the 3d system, this bound can be thought of as bound in terms of the topological entanglement entropy [6, 7, 49]. Furthermore, it is an interesting question if any hyperbolic knot or satellite knot may violate this bound.

With a slight generalization of the method one can derive a similar constraint for the multiplicities $N_\zeta^\eta(p, q)$ (defined in Eq. (3.31)); see §6.4. Therein, we shall present a proof using a slightly different method.

## 6.4   A consistency relation for unknot minus a torus knot

**Proposition 6.8.** *Let $K$ be a $(p, q)$ torus knot, then*

$$\boxed{\sum_{\zeta \in \mathcal{C}_{\text{Hopf}}} N_\zeta^\eta(p, q) d_\zeta = \sum_{\substack{\mu \in \mathcal{C}_{\text{flux}}: \\ t_p(\mu) = t_{(p,q)}(\eta)}} \frac{d_\mu^2 d_\eta}{d_{t_p(\mu)}^2}.} \tag{6.17}$$

*Proof.* Consider the region shown in Fig. 29. It has three boundaries, and the associated multiplicities are $\{N_{\zeta\eta'}^\eta\}$. Let $\{N_{\zeta\varphi^\vee(\mu)}^\eta\}$ be a subset of these multiplicities. Here $\varphi^\vee : \mathcal{C}_{\text{flux}} \hookrightarrow \mathcal{C}_{\text{Hopf}}$ is an embedding defined as $\varphi : \mathcal{C}_{\text{flux}} \hookrightarrow \mathcal{C}_{\text{loop}} \subset \mathcal{C}_{\text{Hopf}}$ followed by the transformation $\eta \to \eta^\vee$. By the associativity theorem,

$$N_\zeta^\eta(p, q) = \sum_{\mu \in \mathcal{C}_{\text{flux}}} N_{\zeta\varphi^\vee(\mu)}^\eta. \tag{6.18}$$

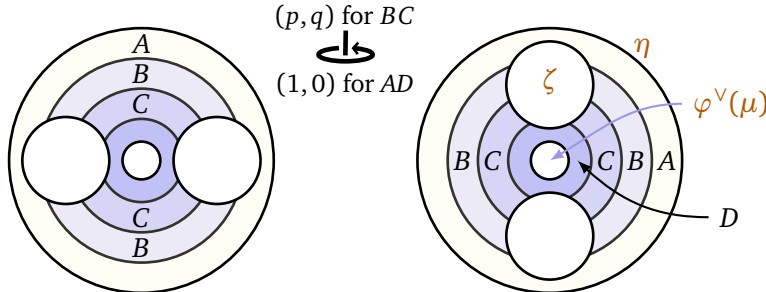

Figure 29: A decomposition of a torus shell minus a $(p,q)$ torus knot. The figure is precise for $(p,q) = (2,3)$, i.e., when the torus knot is a trefoil.

Next, we show

$$\sum_{\zeta \in \mathcal{C}_{\text{Hopf}}} N^{\eta}_{\zeta \varphi^{\vee}(\mu)} d_{\zeta} = \frac{d_{\mu}^2 d_{\eta}}{d_{t_p(\mu)}^2} \delta_{t_p(\mu), t_{(p,q)}(\eta)} \,. \tag{6.19}$$

Suppose $\eta$ and $\varphi^{\vee}(\mu)$ can label the two boundaries in Fig. 29 for some state in $\Sigma(ABCD)$. Then we must have $t_p(\mu) = t_{(p,q)}(\eta)$. This is because, that is the condition for the marginal on $ABC$ (completed determined by $\eta$) and the marginal on $BCD$ (completely determined by $\varphi^{\vee}(\mu)$) to match on the overlapping region $BC$.

If $\delta_{t_p(\mu) \neq t_{(p,q)}(\eta)}$ then $N^{\eta}_{\zeta \varphi^{\vee}(\mu)} = 0$ for any $\zeta$. This is consistent with Eq. (6.19).

If $\delta_{t_p(\mu) = t_{(p,q)}(\eta)}$ we are able to merge $\rho^{\eta}_{ABC}$ and $\rho^{\varphi^{\vee}(\mu)}_{BCD}$. The existence of a merged state, which we denote as $\tau^{\eta\mu}_{ABCD}$, implies that $\sum_{\zeta} N^{\eta}_{\zeta \varphi^{\vee}(\mu)} > 0$. To derive an equality, we consider the entropy difference $S(\tau^{\eta\mu}_{ABCD}) - S(\tau^{11}_{ABCD})$ and note that $\tau^{11}_{ABCD} = \sigma_{ABCD}$.

With the structure of $\Sigma(ABCD)$, we have

$$S\left(\tau^{\eta\mu}_{ABCD}\right) - S\left(\tau^{11}_{ABCD}\right) = \ln d_{\eta} + \ln d_{\varphi^{\vee}(\mu)} + \ln\left(\sum_{\zeta} N^{\eta}_{\zeta \varphi^{\vee}(\mu)} d_{\zeta}\right). \tag{6.20}$$

On the other hand, by looking at the marginals, we get

$$S\left(\tau^{\eta\mu}_{ABCD}\right) - S\left(\tau^{11}_{ABCD}\right) = 2\ln d_{\eta} + 2\ln d_{\varphi^{\vee}(\mu)} - 2\ln d_{t_p(\mu)} \,. \tag{6.21}$$

By comparing these two equations, we get:

$$\sum_{\zeta} N^{\eta}_{\zeta \varphi^{\vee}(\mu)} d_{\zeta} = \frac{d_{\eta} d_{\mu}^2}{d_{t_p(\mu)}^2} \,. \tag{6.22}$$

Here, we have plugged in $d_{\varphi^{\vee}(\mu)} = d_{\mu}^2$. This verifies Eq. (6.19) and completes the proof. $\qquad \square$

# 7 Discussion and open questions

We have extended the entanglement bootstrap approach to 3d gapped phases. Starting with the two axioms on the entanglement entropy, we are able to make concrete statements of the general structures of the theory. The focus of this work is on a detailed analysis of the diverse excitation types and the fusion spaces and processes that appear in these systems. This also set up the foundation for future investigation of braiding statistics [20].

There are many approaches to studying gapped quantum many-body systems in three spatial dimensions. Under some assumptions that rule out fracton phases, one expects the extreme low-energy physics to be governed by a suitable topological field theory [32, 33, 50, 51]. Given the exception represented by fracton phases, one may wonder precisely when such a framework applies. One of the goals of the entanglement bootstrap approach is to answer this question.

The logic we use is completely independent from these prior studies. Here is a physical summary of what we learned:

1. We are able to obtain the vacuum state on the 3-sphere from a vacuum state (i.e., the reference state) on a ball. This "sphere completion" is the first step toward a bigger goal: constructing closed space manifolds (and possibly spacetime manifolds) of different topologies from "local" knowledge of a ground state [20].

2. The set of point particles is nontrivial whenever the set of loop excitations is nontrivial. In other words, it is impossible to have a 3d topological order with only loop excitations, nor can there be a 3d topological order with only particle excitations. In particular, the total quantum dimension computed from particles equals that computed from flux loops (Proposition 3.4).

3. The simplest class of loop excitations we identify are the flux loops. Fluxes are both simple and nontrivial. For example, there is a way to fuse the flux loops spirally into a new flux loop. The spiral map provides an understanding of this process. Interestingly, this spiral map seem to escape all available dimensional reductions, e.g. in §3.3 and in Appendix E.

4. Loop excitations can be linked and knotted. The excitations on Hopf links provide the most general superselection sectors on a closed loop.[25] The set of loop excitations that can exist alone on a single knot $K \subset S^3$ must be a subset of Hopf sectors.

5. For a non-Abelian theory (where quantum dimensions differ from unity), the knot complement can serve as a quantum memory; this is in the same manner as, e.g., the complement of a few Fibonacci anyons on a sphere in 2d. One notable difference is that we need only one knot. The fusion space associated with the knot excitations are identified in §3.4, and the coherence is verified by examples in §4.3.

6. For torus knots, we are able to verify consistency relations for their knot multiplicities (§6). In particular, there is a universal bound for their knot multiplicities, (6.15). It would be interesting to see if any hyperbolic knot or satellite knot may violate this bound.

7. We assumed that the global Hilbert space is the tensor product of local Hilbert spaces. This assumption limits our analysis to systems made of bosons. However, the topological point particle excitations can be either emergent bosons or emergent fermions. It would be interesting to (1) come up with a way to tell which emergent particles are bosons and which emergent particles are fermions and (2) modify this assumption to allow the study of fermionic systems.

8. We assumed that the reference state is defined on a ball, and the ball is "smooth" in the sense that both axioms are satisfied on *all* bounded-radius balls contained in that ball. On the other hand, some of the statements we prove require a weaker assumption (e.g., Proposition 5.4 and 5.6). Statements like these can be adapted to 3d systems with codimension-1, 2, and 3 defects.

---

[25]Here, closed loops are embedded circles. On a graph, there can be other excitation types.

9. In 2d, no known system with nonzero chiral central charge obeys the precise version of the axioms. One may wonder if we miss a certain exotic class of gapped phases of matter by assuming the exact axioms. We do not have a concrete no-go theorem. In 2d, one hope is to develop a robust version of entanglement bootstrap to accommodate chiral phases. While a robust version is currently unavailable, a piece of supporting evidence is that related tools are shown to be useful in guessing a new formula for chiral central charge [11]. In 3d, axiom **A1** does exclude fractons, which violate this axiom with a linear term. It is interesting to ask if there are other interesting 3d gapped phases that escape the description (meaning that no representative wave function of the phase satisfies the two axioms precisely).

We conclude with some other outstanding open questions.

1. The isomorphism between embedded regions gives rise to a well-defined notion of particle types. For a chosen disk, a universal "reference frame" for comparing anyon types exists in 2d (Lemma 4.3 of [1]). The basic intuition here is that an embedded annulus cannot flip within a disk. The same observation generalizes to the sphere shell in 3d. A nontrivial automorphism does exist for loop excitations because a torus can flip within a ball. The sphere completion lemma provides extra flexibility: on a sphere, an annulus can be turned inside through a sequence of embedded intermediate configurations. This provides a nice way to think about antiparticles.

   Immersion provides additional flexibility for the choice of regions as well as the ways regions can deform. For example, the torus shell can be mapped to itself in many different ways through immersion. One simple thing that we already learned from immersion is the product rule of the spiral maps. On the other hand, many open questions remain, suggesting that we may be able to learn more from immersion. Instead of providing a long list of such questions, we mention a simple one. In 3d, a sphere shell can be flipped inside out through immersion [46–48]; this can be done within a ball rather than the $S^3$ obtained from sphere completion. This generates a permutation of the labels in $\mathcal{C}_{\text{point}}$. Is it true that this permutation always maps the vacuum sector $1 \in \mathcal{C}_{\text{point}}$ back to itself?

   It will be interesting to understand the homotopy properties of the space of (immersed) paths between regions, for example, in the definition of the spiral map.

2. We have studied two types of dimensional reduction in Appendix E.1. The idea is to identify 2d regions related to the 3d region by a revolution. A couple of open questions remain. One question is if the dimensional reduction for any nontrivial flux is a 2d system that allows a gapped boundary. Another question is if inequivalent dimensional reduction arises by considering a revolution of type $(p, q)$.

3. Are sectorizable regions always of the form $\mathcal{M} \times \mathbb{I}$, i.e., a manifold (possibly with boundary) times an interval?

4. The naive analog of the shrinking rule (6.3) for knots other than the unknot is not true. Perhaps the correct generalization can be found by combining the consistency conditions for ball minus torus and for torus minus knot. Such a generalization would help us understand the correct interpretation of fusion equations such as (3.30).

5. In the quantum double examples, we can see an intimate relationship between the information convex set of a knot complement and its fundamental group, the knot group, and more specifically with the Wirtinger presentation of the knot group. While ground states of the quantum double on the knot complement with a particular gapped boundary condition are in one-to-one correspondence with *representations* of the knot group

in the gauge group $G$ (modulo conjugation) [52], the elements of the information convex set are instead density matrices. This seems to be a new mathematical structure on which the knot group can act; it is interesting to ask about the ability of such actions to distinguish knots from each other.

6. In this paper, we have focused on torus knots. It is an open question whether the structure of the information convex set shows some dramatic difference for satellite knots or hyperbolic knots. We conjecture that the bounds (6.15) of §6.3 can be violated for more general knots.

**Acknowledgments.** We are grateful to Meng Cheng, Tarun Grover, Jeongwan Haah, Chao-Ming Jian, Isaac Kim, Xiang Li, Shu-Heng Shao, Xiao-Gang Wen and Xueda Wen for helpful discussions and comments. This work was supported in part by funds provided by the U.S. Department of Energy (D.O.E.) under the cooperative research agreement DE-SC0009919, by the University of California Laboratory Fees Research Program, grant LFR-20-653926, and by the Simons Collaboration on Ultra-Quantum Matter, which is a grant from the Simons Foundation (652264).

# A  Glossary of notation

| notation | meaning | definition appears in |
|---|---|---|
| $\sigma$ | the reference state | |
| $I(A:C)$ | mutual information: $S_A + S_C - S_{AC}$ | |
| $I(A:C\|B)$ | conditional mutual information: $S_{AB} + S_{BC} - S_B - S_{ABC}$ | |
| $\Delta(B,C)$ | $S_C + S_{BC} - S_B$ | (1.2) |
| $\Delta(B,C,D)$ | $S_{BC} + S_{CD} - S_B - S_D$ | (1.2) |
| $\Gamma(\Omega)$ | the set of bounded-radius balls on the region $\Omega$ on which we impose the axioms | §2.1 |
| $\widetilde{\Gamma}(\Omega_+)$ | the set of balls arising as preimages in $\Omega_+$ under $\mathfrak{p}$ of balls in $\Gamma(\mathfrak{p}(\Omega_+))$ | §2.1 |
| $\mathbb{I}$ | interval | §2.2.1 |
| $\partial\Omega$ | the thickened boundary of region $\Omega$ | §2.2.2 |
| $\Sigma(\Omega)$ | the information convex set of region $\Omega$ | §1, Def. 2.3 |
| $\Sigma_I(\Omega)$ | the subset of $\Sigma(\Omega)$ in sector $I$ on $\partial\Omega$ | (2.11) |
| $\mathbb{V}_I(\Omega)$ | fusion space of $\Omega$ with $I$ boundary conditions | Theorem 2.14 |
| $\mathcal{S}(\mathbb{V})$ | density matrices on $\mathbb{V}$ | Theorem 2.14 |
| $N_I(\Omega)$ | $\dim\mathbb{V}_I(\Omega)$ | §2.3 |
| $d_I$ | quantum dimension of excitation type $I$ | Def. 2.11 |
| $\rho \bowtie \sigma$ | the result of merging the states $\rho$ and $\sigma$ | Lemma 2.6 |
| $\mathrm{ext}(\Sigma(\Omega))$ | the set of extreme points of $\Sigma(\Omega)$ | §3.2 |
| $X$ | sphere shell | §3.2 |
| $T$ | solid torus | §3.2 |
| $\mathbb{T}$ | torus shell | §3.2 |
| $\mathcal{C}_{\mathrm{point}}$ | the set of labels on $\mathrm{ext}(\Sigma(X))$ | (3.3) |
| $\mathcal{C}_{\mathrm{flux}}$ | the set of labels on $\mathrm{ext}(\Sigma(T))$ | (3.5) |
| $\mathcal{C}_{\mathrm{Hopf}}$ | the set of labels on $\mathrm{ext}(\Sigma(\mathbb{T}))$ | (3.7) |
| $\mathcal{C}_{\mathrm{loop}}$ | the set of shrinkable loops | (3.10) |
| $\varphi$ | embedding of $\mathcal{C}_{\mathrm{flux}}$ into $\mathcal{C}_{\mathrm{loop}}$ | (3.10) |
| $\phi$ | embedding of $\mathcal{C}_{\mathrm{point}}$ into $\mathcal{C}_{\mathrm{loop}}$ | (3.11), §6.2.2 |
| $\Omega_K$ | complement of the knot $K$ in $S^3$ | (3.22) |
| $(G)_{\mathrm{ir}}$ | irreps of $G$ | §4 |
| $(G)_{\mathrm{cj}}$ | conjugacy classes of $G$ | §4 |
| $E_g$ | the centralizer of $g \in G$ | §4 |
| $C_g$ | the conjugacy class of $g \in G$ | §4 |
| $E_{(g,h)}$ | $E_g \cap E_h$ | §4 |
| $C_{(g,h)}$ | $\{(tgt^{-1}, tht^{-1})\|t \in G\}$ | §4 |
| $\mathcal{T}_n$ | $n$th spiral map on $\Sigma(T)$ | (5.1) |
| $t_n$ | $n$th spiral map on fluxes | (5.6) |

# B  Proof of associativity theorem

In this appendix, we present the proof of the associativity theorem (Theorem 2.22).

## B.1  Proof of Lemma 2.20

Below is the proof of Lemma 2.20.

*Proof.* The first statement is simple to prove. The fact that $\rho_{\Omega_L}$ and $\lambda_{\Omega_R}$ can be merged follows

from (1) $I(A_L B_L : C_R | C_L)_\rho = 0$ and $I(C_L : B_R A_R | C_R)_\lambda = 0$, and (2) $\rho_{\Omega_L}$ and $\rho_{\Omega_R}$ are consistent on $C$.

To prove the second statement, it is enough to verify the extreme point criterion stated in Lemma 2.13, namely $(S_\Omega + S_{\Omega \setminus \partial \Omega} - S_{\partial \Omega})_\tau = 0$, where $\tau$ is the merged state. This identity follows algebraically from the following entropy conditions on $\tau$:

$$\begin{aligned}
S_{\Omega_L} + S_{B_L} - S_{A_L} - S_C &= 0, \\
S_{\Omega_R} + S_{B_R} - S_{A_R} - S_C &= 0, \\
I(A_L B_L : A_R B_R | C) &= 0, \\
S_{BC} + S_C - S_B &= 0, \\
I(B_L : B_R) &= 0, \\
I(A_L : A_R) &= 0.
\end{aligned} \tag{B.1}$$

The first and the second lines are the factorization property for $\rho_{\Omega_L}$ and $\lambda_{\Omega_R}$ respectively. The third line follows from the quantum Markov chain property of the merged state. The fourth, fifth, and sixth lines are consequences of the factorization property of extreme points and SSA; in more details, the fourth line needs $\tau$ restricted to $C$ to be an extreme point whereas the fifth and sixth lines use the fact that $\tau$ restricted to $B_L$ and $A_L$ are extreme points respectively.

To see explicitly the actual algebra that leads to the final answer, we rewrite each line of Eq. (B.1) and see, for the state $\tau$:

$$\begin{aligned}
0 &= -S_{\Omega_L} - S_{B_L} + S_{A_L} + S_C, \\
0 &= -S_{\Omega_R} - S_{B_R} + S_{A_R} + S_C, \\
S_\Omega &= S_{\Omega_L} + S_{\Omega_R} - S_C, \\
S_{\Omega \setminus \partial \Omega} &= S_B - S_C, \\
0 &= -S_B + S_{B_L} + S_{B_R}, \\
-S_{\partial \Omega} &= -S_{A_L} - S_{A_R}.
\end{aligned} \tag{B.2}$$

By adding each side of all the six lines, we arrive at $(S_\Omega + S_{\Omega \setminus \partial \Omega} - S_{\partial \Omega})_\tau = 0$. Thus the merged state $\tau$ is an extreme point of $\Sigma(\Omega)$. This completes the proof. $\square$

## B.2 Proof of Theorem 2.22 (associativity)

Below is the proof of the associativity theorem (Theorem 2.22). This theorem relates the dimensions of the fusion spaces of a region obtained by merging two subregions along a whole boundary component to those of the subregions being merged. In the writing of the proof, we omit the subsystem labels, and denote $N^{a_R}_{a_L}(\Omega)$, $N^i_{a_L}(\Omega_L)$ and $N^{a_R}_i(\Omega_R)$ as $N^{a_R}_{a_L}$, $N^i_{a_L}$ and $N^{a_R}_i$ for simplicity.

*Proof.* If $N^{a_R}_{a_L} = 0$, then we must have $\sum_{i \in \mathcal{C}_C} N^i_{a_L} N^{a_R}_i = 0$. If this were not the case, it would imply that $\Sigma^i_{a_L}(\Omega_L)$ and $\Sigma^{a_R}_i(\Omega_R)$ were both nonempty. Then it would be possible to take an element from each set and merge them; the end result would be an element in $\Sigma^{a_R}_{a_L}(\Omega)$. This would contradict the statement that $N^{a_R}_{a_L} = 0$.

If $N^{a_R}_{a_L} \geq 1$, then $\Sigma^{a_R}_{a_L}(\Omega)$ is nonempty. This implies that $\sum_{i \in \mathcal{C}_C} N^i_{a_L} N^{a_R}_i \geq 1$. On general grounds,

$$\ln N^{a_R}_{a_L} = S\left(\rho_\Omega^{(a_L, a_R)_{\max}}\right) - S\left(\rho_\Omega^{(a_L, a_R)_{\min}}\right), \tag{B.3}$$

where $\rho_\Omega^{(a_L, a_R)_{\max}}$ is the maximum-entropy state of $\Sigma^{a_R}_{a_L}(\Omega)$ and $\rho_\Omega^{(a_L, a_R)_{\min}}$ is an extreme point of $\Sigma^{a_R}_{a_L}(\Omega)$.

The maximum-entropy state $\rho_\Omega^{(a_L, a_R)_{\max}}$ obeys a quantum Markov chain condition:

$$I(A_L B_L : A_R B_R | C) = 0. \tag{B.4}$$

This follows from Lemma 2.21; if the maximum-entropy state were not Markov, we could merge the marginals to a state with larger entropy. In addition, while not every extreme point (i.e., minimum-entropy state) satisfies Eq. (B.4) in general, it is possible to choose an extreme point with this property. Consider a superselection sector $i_0 \in \Sigma(C)$ such that $N_{a_L}^{i_0} N_{i_0}^{a_R} \geq 1$. Choose an extreme point from $\Sigma_{a_L}^{i_0}(\Omega_L)$ and $\Sigma_{i_0}^{a_R}(\Omega_R)$ respectively and merge them; in this way, we obtain an extreme point of $\Sigma_{a_L}^{a_R}(\Omega)$ that does obey Eq. (B.4); see Lemma 2.20. In the following, $\rho_\Omega^{(a_L, a_R)_{\min}}$ refers to the *specific* extreme point associated with the choice $i_0$.

Quantum Markov chains saturate SSA and therefore (B.3) becomes

$$\ln N_{a_L}^{a_R} = (S_{\Omega_L} + S_{\Omega_R} - S_C)_{\rho^{(a_L, a_R)_{\max}}} - (S_{\Omega_L} + S_{\Omega_R} - S_C)_{\rho^{(a_L, a_R)_{\min}}}. \tag{B.5}$$

Now consider three partial traces of $\rho_\Omega^{(a_L, a_R)_{\max}}$:

$$
\begin{aligned}
\mathrm{Tr}_{A_R B_R}\, \rho_\Omega^{(a_L, a_R)_{\max}} &= \sum_i p_i^{(a_L, a_R)} \rho_{\Omega_L}^{(a_L, i)_{\max}}, \\
\mathrm{Tr}_{A_L B_L}\, \rho_\Omega^{(a_L, a_R)_{\max}} &= \sum_i p_i^{(a_L, a_R)} \rho_{\Omega_R}^{(i, a_R)_{\max}}, \\
\mathrm{Tr}_{AB}\, \rho_\Omega^{(a_L, a_R)_{\max}} &= \sum_i p_i^{(a_L, a_R)} \rho_C^i.
\end{aligned}
\tag{B.6}
$$

Here the sum over $i$ runs over $\mathcal{C}_C$, the set of superselection sectors for sectorizable region $C$. $\{p_i^{(a_L, a_R)}\}$ is a probability distribution, and it is the same in each of these expressions because the marginals must agree. We set $p_i^{(a_L, a_R)} = 0$ when $N_{a_L}^i N_i^{a_R} = 0$ for the same reason.

Next, we determine $\{p_i^{(a_L, a_R)}\}$ by maximizing the entropy difference. The consistency relations associated with the optimal choice reveals the associtivity. Using our knowledge of the structure of the information convex sets of each of these regions, we can evaluate each of the differences on the RHS of (B.5).

$$
\begin{aligned}
\delta S_{\Omega_L} &= \sum_i p_i^{(a_L, a_R)} \left( \ln \frac{d_i}{d_{i_0}} + \ln N_{a_L}^i - \ln p_i^{(a_L, a_R)} \right), \\
\delta S_{\Omega_R} &= \sum_i p_i^{(a_L, a_R)} \left( \ln \frac{d_i}{d_{i_0}} + \ln N_i^{a_R} - \ln p_i^{(a_L, a_R)} \right), \\
\delta S_C &= \sum_i p_i^{(a_L, a_R)} \left( 2 \ln \frac{d_i}{d_{i_0}} - \ln p_i^{(a_L, a_R)} \right).
\end{aligned}
\tag{B.7}
$$

Therefore

$$
\begin{aligned}
\ln N_{a_L}^{a_R} &= \max_{\{p_i^{(a_L, a_R)}\}} \left[ \sum_i p_i^{(a_L, a_R)} \left( \ln(N_{a_L}^i N_i^{a_R}) - \ln p_i^{(a_L, a_R)} \right) \right] \\
&= \ln\left( \sum_i N_{a_L}^i N_i^{a_R} \right),
\end{aligned}
\tag{B.8}
$$

where the maximum is achieved by $p_i^{(a_L, a_R)} = \frac{N_{a_L}^i N_i^{a_R}}{\sum_{i'} N_{a_L}^{i'} N_{i'}^{a_R}}$. Therefore

$$N_{a_L}^{a_R} = \sum_i N_{a_L}^i N_i^{a_R}.$$

This completes the proof. $\qquad\square$

Table 6: The regions and the multiplicities we consider and where to find them.

| Region | Multiplicities | Subsection |
|---|---|---|
| Ball minus unknot | $\{N_l^a\}$ | Appendix C.2 |
| Ball minus trefoil | $\{N_\zeta^a(\text{trefoil})\}$ | Appendix C.3 |
| Solid torus minus trefoil | $\{N_\zeta^\eta(2,3)\}$ | Appendix C.4 |
| Borromean ring complement | $\{N_{\eta_1\eta_2\eta_3}\}$ | Appendix C.5 |

# C  Calculation details for 3d quantum double

In this appendix, we provide some details of the calculation of 3d quantum double models as well as additional examples of regions. The essential tools to understand the calculations are finite groups and their representations; Appendix C.1. The remaining parts are examples of regions to illustrate the fusion spaces and consistency relations.

## C.1  Group theory notation and useful properties

We start by reviewing some basic notation of finite groups and representations. Also reviewed are a few properties that will be handy in doing the calculation.

### C.1.1  Notation and facts

Let $g \in G$ be an element of finite group $G$. We denote its inverse as $g^{-1}$. 1 is the identity of the group. $|G|$ is the number of group elements in $G$. $G \times H$ is the tensor product of finite groups $G$ and $H$, namely $G \times H = \{(g,h)|g \in G, h \in H\}$. The group multiplication is: $(g_1, h_1)(g_2, h_2) = (g_1 g_2, h_1 h_2)$.

We shall denote the set of conjugacy classes of $G$ as $(G)_{\text{cj}}$. $C_g$ is the conjugacy class of $G$ that contains $g$. $C_{(g,h)} \equiv \{(tgt^{-1}, tht^{-1})|t \in G\}$, where $g, h \in G$. $E_g = \{t \in G \mid tgt^{-1} = g\}$ is the centralizer of $g$. $E_{(g,h)} \equiv \{t \in G|(g,h) = (tgt^{-1}, tht^{-1})\} = E_g \cap E_h$.

We shall only consider unitary representations. $(G)_{\text{ir}}$ is the set of irreducible representations of $G$. $\dim R$ is the dimension of representation $R$.

$\Gamma_R(g)$ is the unitary matrix of representation $R$ for the group element $g \in G$. For each representation $R$, there is a dual representation $\bar{R}$ defined such that $\Gamma_{\bar{R}}(g) = \bar{\Gamma}_R(g)$, where $\bar{\Gamma}_R(g)$ is the complex conjugation of $\Gamma_R(g)$. The character of representation $R$ is $\chi_R(g) = \text{Tr}\,\Gamma_R(g) = \sum_{i=1}^{\dim R} \Gamma_R^{ii}(g)$.

Here is a list of facts:

1. Orthogonality:

$$\frac{1}{|G|} \sum_{g \in G} \Gamma_R^{ab}(g)\Gamma_{R'}^{a'b'*}(g) = \frac{1}{\dim R} \delta_{R,R'} \delta_{a,a'} \delta_{b,b'}, \quad \text{for } R, R' \in (G)_{\text{ir}}. \tag{C.1}$$

2. Character $\chi_R(g)$ is a function of conjugacy class. Furthermore, $\chi_R(1) = \dim R$, $\chi_R(g^{-1}) = \chi_R^*(g) = \chi_{\bar{R}}(g)$. Denote $\langle \chi_R|\chi_{R'}\rangle \equiv \frac{1}{|G|}\sum_{g \in G} \chi_R(g)\chi_{R'}^*(g)$, then

$$\langle \chi_R|\chi_{R'}\rangle = \delta_{i,j}, \quad \text{for } R, R' \in (G)_{\text{ir}}. \tag{C.2}$$

Therefore, for any representation $\mathcal{H}$ of $G$, and $R \in (G)_{\text{ir}}$:

$$\langle \chi_R|\chi_{\mathcal{H}}\rangle = \text{the number of } R \text{ contained in } \mathcal{H}. \tag{C.3}$$

3. The group $G \times H$ has $(G \times H)_{\text{ir}} = \{R \times \pi | R \in (G)_{\text{ir}}, \pi \in (H)_{\text{ir}}\}$, where

$$\dim(R \times \pi) = \dim R \cdot \dim \pi \quad \text{and} \quad \chi_{R \times \pi} = \chi_R \cdot \chi_\pi . \tag{C.4}$$

4. Consider the Hilbert space $\mathcal{H}_G \equiv \text{span}\{|g\rangle | g \in G\}$, with $\langle g|k\rangle = \delta_{g,k}$. Under the unitary operator of $G$: $\Gamma(g)|h\rangle = |gh\rangle$, for $g, h \in G$, $\mathcal{H}_G$ decomposes into irreducible representations of $G$ as:

$$\mathcal{H}_G = \bigoplus_{R \in (G)_{\text{ir}}} \dim R \cdot R . \tag{C.5}$$

Under the unitary operation of $G \times G$: $\Gamma(m, n)|h\rangle = |mhn^{-1}\rangle$, for $(m, n) \in G \times G$ and $h \in G$, $\mathcal{H}_G$ decomposes into irreducible representations of $G \times G$ as:

$$\mathcal{H}_G = \bigoplus_{R \in (G)_{\text{ir}}} R \otimes \bar{R} . \tag{C.6}$$

### C.1.2 Conjugation-invariant density matrices

We prove a general statement about conjugation-invariant density matrices in a group theoretical context. This statement will be useful in solving the conjugation constraints of minimal diagrams.

**Proposition C.1** (Conjugation-invariant density matrices). *Let* $\mathcal{H} = \bigoplus_{R \in (G)_{\text{ir}}} n_R R$, *with* $n_R \in Z_{\geq 0}$, *be a representation of* $G$. *We denote the unitary group operation on* $\mathcal{H}$ *as* $\Gamma_{\mathcal{H}}(g)$, $g \in G$. *Then, the set of density matrices* $\rho$ *on* $\mathcal{H}$ *which satisfy*

$$\Gamma_{\mathcal{H}}(g) \rho \, \Gamma_{\mathcal{H}}^\dagger(g) = \rho , \quad \forall g \in G , \tag{C.7}$$

*forms a convex set* $\Sigma(\mathcal{H})$, *which is the convex hull of orthogonal subsets* $\Sigma_R(\mathcal{H})$ *of the form*

$$\Sigma_R(\mathcal{H}) = \left\{ \rho^R \, \middle| \, \rho^R \in \mathcal{S}(\mathbb{V}_R) \otimes \frac{\text{Id}_{\dim R}}{\dim R} \right\}, \quad R \in (G)_{\text{ir}} , \tag{C.8}$$

*where* $\mathcal{S}(\mathbb{V}_R)$ *is the state space of* $n_R$ *dimensional Hilbert space* $\mathbb{V}_R$, *and* $\text{Id}_{\dim R}$ *is the identity operator on a* $\dim R$ *dimensional Hilbert space.*

We shall say a density $\rho$ on $\mathcal{H}$ is conjugation invariant if it satisfies Eq. (C.7).

*Proof.* It is easy to verify that if two density matrices $\rho_1, \rho_2$ are conjugation invariant, their convex combination $p_1\rho_1 + (1-p)\rho_2, p \in [0, 1]$ is also conjugation invariant. Therefore, $\Sigma(\mathcal{H})$ is a convex set.

Next we need to show its extreme points are elements of $\mathcal{S}(\mathbb{V}_R) \otimes \frac{\text{Id}_{\dim R}}{\dim R}$ for $R \in (G)_{\text{ir}}$. We prove this by analyzing the possible forms of the extreme points.

Consider the density matrices in the form $\rho = \frac{1}{|G|} \sum_{g \in G} \Gamma_{\mathcal{H}}(g) \lambda \Gamma_{\mathcal{H}}^\dagger(g)$, where $\lambda$ is an arbitrary density matrix. We see $\rho$ is conjugation invariant. Furthermore, the set of all such $\rho$ is precisely $\Sigma(\mathcal{H})$.

Since $\lambda$ can be written as $\lambda = \sum_\alpha p_\alpha |\alpha\rangle \langle\alpha|$, and $\frac{1}{|G|} \sum_{g \in G} \Gamma_{\mathcal{H}}(g) |\alpha\rangle \langle\alpha| \Gamma_{\mathcal{H}}^\dagger(g)$ for each $\alpha$ is conjugation invariant, all the extreme points are elements of the set

$$S \equiv \left\{ \frac{1}{|G|} \sum_{g \in G} \Gamma_{\mathcal{H}}(g) |\alpha\rangle \langle\alpha| \Gamma_{\mathcal{H}}^\dagger(g), \quad |\alpha\rangle \in \mathcal{H} \right\} . \tag{C.9}$$

This is because extreme points cannot be expressed as convex combination of two other points with positive coefficients.

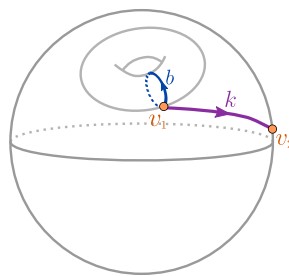

Figure 30: The minimal diagram for a ball minus an unknot.

We consider following orthonormal basis of $\mathcal{H}$:

$$\mathcal{H} = \text{span} \{|R, \mu, a\rangle \,|\, R \in (G)_{\text{ir}}, \ \mu = 1, \cdots, n_R, \ a = 1, \cdots, \dim R\} . \tag{C.10}$$

(We omit $R$ for which $n_R = 0$.) Moreover, we shall write $|R, \mu, a\rangle$ as $|R, \mu\rangle \otimes |a\rangle$ when needed. Let $|\alpha\rangle = \sum_{R,\mu,a} C_{R\mu a} |R, \mu, a\rangle$, then

$$\frac{1}{|G|} \sum_{g \in G} \Gamma_{\mathcal{H}}(g) |\alpha\rangle \langle \alpha| \Gamma_{\mathcal{H}}^{\dagger}(g)$$

$$= \sum_{R\mu a} \sum_{R',\mu',a'} C_{R\mu a} C_{R'\mu'a'}^{*} \left( \frac{1}{|G|} \sum_{g \in G} \Gamma_{\mathcal{H}}(g) |R, \mu, a\rangle \langle R', \mu', a'| \Gamma_{\mathcal{H}}(g)^{\dagger} \right)$$

$$= \sum_{R,\mu,a,b} \sum_{R',\mu',a',b'} \left( \frac{1}{|G|} \sum_{g \in G} \Gamma_R^{ab}(g) \Gamma_{R'}^{a'b'*}(g) \right) C_{R\mu a} C_{R'\mu'a'}^{*} |R, \mu, b\rangle \langle R', \mu', b'|$$

$$= \sum_{R,\mu,a,b} \sum_{R',\mu',a',b'} \left( \frac{1}{\dim R} \delta_{R,R'} \delta_{a,a'} \delta_{b,b'} \right) C_{R\mu a} C_{R'\mu'a'}^{*} |R, \mu, b\rangle \langle R', \mu', b'|$$

$$= \sum_{R \in (G)_{\text{ir}}} \left[ \sum_{\mu=1}^{n_R} \sum_{\mu'=1}^{n_R} \left( \sum_a C_{R\mu a} C_{R\mu'a}^{*} \right) |R, \mu\rangle \langle R, \mu'| \otimes \left( \frac{1}{\dim R} \sum_{b=1}^{\dim R} |b\rangle \langle b| \right) \right] .$$

The summands in the last expression are orthogonal for different $R \in (G)_{\text{ir}}$. Therefore the extreme points are elements of $\mathcal{S}(\mathbb{V}_R) \otimes \frac{\text{Id}_{\dim R}}{\dim R}$ for $R \in (G)_{\text{ir}}$. $\qquad \square$

## C.2 Ball minus unknot

Here we calculate the multiplicities for the shrinking rule: $\{N_l^a\}$ defined in (3.17). Here $a \in \mathcal{C}_{\text{point}}$ and $l \in \mathcal{C}_{\text{loop}}$. As a reminder, the relevant rules for minimal diagram are reviewed in §4.2. The content of the next few sections are essentially a continuation of §4.3.

The region that characterizes the multiplicities $\{N_l^a\}$ is a ball minus an unknot ($D^3 \setminus T$). We consider the minimal diagram shown in Fig. 30. This minimal diagram contains a boundary link (labeled by $b$), a bulk link (labeled by $k$), and two vertices $v_1$ and $v_2$. There is no face in this diagram.

By solving the three types of constraints described in §4.2.2, we arrive at a closed-form formula for any finite group $G$.

$$\boxed{N_l^a = \text{ the number of } R_l \times \bar{R}_a \in (E_g \times G)_{\text{ir}} \text{ contained in } \mathcal{H}_b.} \tag{C.11}$$

Here, $l = (C_b, R_l)$, $R_l \in (E_b)_{\text{ir}}$ and $a = R_a \in (G)_{\text{ir}}$; $\mathcal{H}_b$ is a representation of $E_b \times G$ defined according to the following steps:

Table 7: For 3d $S_3$ quantum double. Multiplicities $\{N_l^a\}$ associated with a ball minus an unknot. The columns are point particle types ($a \in \mathcal{C}_{\text{point}}$), and the rows are shrinkable loop types ($l \in \mathcal{C}_{\text{loop}}$). $\omega \equiv e^{2\pi i/3}$ and $\omega^2$ label the nontrivial irreps of $\mathbb{Z}_3$.

| $N_l^a$ \qquad $a$ <br> $l$ | $\text{Id}_{S_3}$ | $\text{Sign}_{S_3}$ | $\Pi_{S_3}$ |
|---|---|---|---|
| $(C_1, \text{Id}_{S_3})$ | 1 | 0 | 0 |
| $(C_1, \text{Sign}_{S_3})$ | 0 | 1 | 0 |
| $(C_1, \Pi_{S_3})$ | 0 | 0 | 1 |
| $(C_r, \text{Id}_{\mathbb{Z}_3})$ | 1 | 1 | 0 |
| $(C_r, \omega_{\mathbb{Z}_3})$ | 0 | 0 | 1 |
| $(C_r, (\omega^2)_{\mathbb{Z}_3})$ | 0 | 0 | 1 |
| $(C_s, \text{Id}_{\mathbb{Z}_2})$ | 1 | 0 | 1 |
| $(C_s, \text{Sign}_{\mathbb{Z}_2})$ | 0 | 1 | 1 |

1. Choose a $C_b$ that we want to study.

2. $\mathcal{H}_b$, as a Hilbert space, is defined as $\text{span}\{|k\rangle | k \in G\}$.

3. $\mathcal{H}_b$ is a representation of $E_b \times G$ by the group action:

$$\Gamma(t)|k\rangle = |t_1 k t_2^{-1}\rangle, \quad \forall t = (t_1, t_2) \in E_b \times G. \tag{C.12}$$

The final result has a simple closed-form in terms of characters:

$$N_l^a = \frac{1}{|E_b|} \sum_{g \in E_b} \chi_{R_l}(g) \chi_{R_a}^*(g). \tag{C.13}$$

Here we used the fact that $E_b$ is a subgroup of $G$. In words, $N_l^a$ equals to the number of $R_l$ contained in $R_a$, treating $R_a$ as a representation of the subgroup $E_b \subset G$.

For the group $G = S_3$, an explicit calculation leads to Table 7. By recalling the quantum dimensions $d_a = \dim R_a$ and $d_l = |C_b| \cdot \dim R_l$ and checking each column of table 7, we can verify consistency relations:

$$
\begin{aligned}
d_a &= \frac{1}{\mathcal{D}^2} \sum_l N_l^a d_l, \\
d_l &= \sum_a N_l^a d_a,
\end{aligned}
\tag{C.14}
$$

where $\mathcal{D} = \sqrt{\sum_a d_a^2} = \sqrt{6}$. This is Proposition 6.2.1.

## C.3 Ball minus a trefoil

We now compute the multiplicities $\{N_\zeta^a(K)\}$ defined in §3.4, for $K$ a trefoil knot. This multiplicity is associated with region $D^3 \setminus K$, i.e., a ball minus a trefoil.

The minimal diagram is shown in Fig. 31. This minimal diagram has three bulk links (labeled by $a$, $b$ and $k$), two boundary links (labeled by $g$ and $h$), and two vertices $v_1$ and $v_2$. Faces are not shown. Note that, this diagram share similarity with the one for the knot complement (Fig. 16). In particular, we take advantage of the structure of the knot group $\pi_1(D^3 \setminus K) = \pi_1(S^3 \setminus K)$. For a trefoil, the knot group is $\pi_1(D^3 \setminus \text{trefoil}) = \langle \mathfrak{a}, \mathfrak{b} | \mathfrak{a}^2 = \mathfrak{b}^3 \rangle$.

Sci|Post                                    SciPost Phys. 14, 141 (2023)

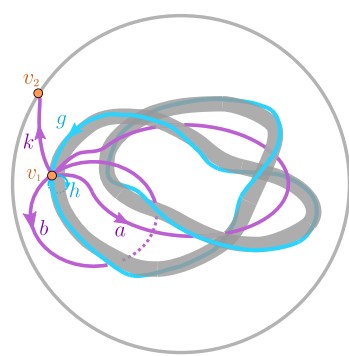

Figure 31: The minimal diagram for a ball minus a trefoil knot.

Table 8: For the 3d $S_3$ quantum double model. Multiplicities associated with a ball minus a trefoil. The columns are point particle types ($a \in \mathcal{C}_{\text{point}}$), and the rows are trefoil excitation types ($\zeta \in \mathcal{C}_{\text{Hopf}}$). Only nonzero multiplicities are shown. Note that the label of $\zeta$ can depend on the framing of the $g$.

| $N_\zeta^a(\text{trefoil})$ $\diagdown$ $a$  $\zeta$ | $\text{Id}_{S_3}$ | $\text{Sign}_{S_3}$ | $\Pi_{S_3}$ |
|---|---|---|---|
| $(C_{(1,1)}, \text{Id}_{S_3})$ | 1 | 0 | 0 |
| $(C_{(1,1)}, \text{Sign}_{S_3})$ | 0 | 1 | 0 |
| $(C_{(1,1)}, \Pi_{S_3})$ | 0 | 0 | 1 |
| $(C_{(1,r)}, \text{Id}_{\mathbb{Z}_3})$ | 1 | 1 | 0 |
| $(C_{(1,r)}, \omega_{\mathbb{Z}_3})$ | 0 | 0 | 1 |
| $(C_{(1,r)}, (\omega^2)_{\mathbb{Z}_3})$ | 0 | 0 | 1 |
| $(C_{(1,s)}, \text{Id}_{\mathbb{Z}_2})$ | 2 | 1 | 3 |
| $(C_{(1,s)}, \text{Sign}_{\mathbb{Z}_2})$ | 1 | 2 | 3 |

By solving the minimal diagram for this problem, we find the general solution:

$$N_\zeta^a(\text{trefoil}) = \text{ the number of } R_\zeta \times \bar{R}_a \in (E_{(g,h)} \times G)_{\text{ir}} \text{ contained in } \mathcal{H}_{(g,h)}. \quad (\text{C.15})$$

Here, $\zeta = (C_{(g,h)}, R_\zeta)$, $R_\zeta \in (E_{(g,h)})_{\text{ir}}$ $a = R_a \in (G)_{\text{ir}}$ and the $\mathcal{H}_{(g,h)}$ is a representation of $E_{(g,h)} \times G$ defined according to the following steps:

1. Choose a $E_{(g,h)}$ that we want to study. Here $g, h \in G$ and $gh = hg$.

2. Find the set of ordered pairs $\{a, b\}$, $a, b \in G$, such that Eq. (4.22) holds.

3. $\mathcal{H}_{(g,h)}$, as a Hilbert space, is defined as $\text{span}\{|a, b, k\rangle | \{a, b\}$ from step 2, and $k \in G\}$.

4. $\mathcal{H}_{(g,h)}$ is a representation of $E_{(g,h)} \times G$ by the group action:

$$\Gamma(t)|a, b, k\rangle = |t_1 a t_1^{-1}, t_1 b t_1^{-1}, t_1 k t_2^{-1}\rangle, \quad \forall t = (t_1, t_2) \in E_{(g,h)} \times G. \quad (\text{C.16})$$

The solution to this problem for finite group $S_3$ is summarized in Table 8. A formula involving characters analogous to (4.26) can summarize these rules.

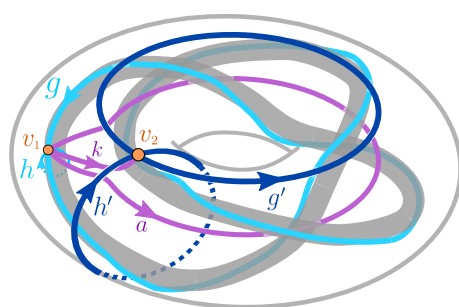

Figure 32: The minimal diagram for a solid torus minus a trefoil. We use it to calculate the multiplicities $\{N_\zeta^l(2,3)\}$.

### C.4 Solid torus minus trefoil

Below, we calculate a subset of $\{N_\zeta^\eta(2,3)\}$ (defined in §3.4.1) by setting $\eta \in \mathcal{C}_{\text{loop}}$. We shall denote this subset as $\{N_\zeta^l(2,3)\}$. We restrict to this set for two reasons. First, these are the multiplicities we need, in order to verify the consistency relation $N_\zeta^a(\text{trefoil}) = \sum_l N_\zeta^l(2,3)N_l^a$, i.e., a special case of Eq. (6.2). Second, the minimal diagram for calculating $\{N_\zeta^l(2,3)\}$ is simpler than the minimal diagram that can handle the general calculation of $\{N_\zeta^\eta(2,3)\}$.

The region associated with this calculation is an (unknotted) solid torus minus a trefoil $T \setminus K$, below $K$ is a trefoil. The minimal diagram is shown in Fig. 32. It contains 4 boundaries links (labeled by $g, h, g', h'$), and 2 bulk links (labeled by $\{a, k\}$). It has two vertices $v_1$ and $v_2$ (and faces which are not shown). We emphasis that this minimal diagram is designed to handle the calculation for $g' = 1$, which corresponds to restricting $\eta$ to shrinkable loops. Because $g' = 1$, the knot group $\pi_1(S^3 \setminus K) = \langle \mathfrak{a}, \mathfrak{b} | \mathfrak{a}^2 = \mathfrak{b}^3 \rangle$ is still useful in this problem. (Note that $\pi_1(T \setminus K)$ is different.) Here, as the label suggests, $a$ corresponds to the knot group generator $\mathfrak{a}$. $b \equiv kh'^{-1}k^{-1}$ play the role of the other generator $\mathfrak{b}$.

The multiplicities $\{N_\zeta^l(2,3)\}$ are specified by two labels of the superselection sectors. $\zeta = (C_{(g,h)}, R_\zeta)$ for the excitation on the trefoil and $l = \{C_{h'}, R_l\}$ for the excitation on the unknotted torus.

The knot group constraints give

$$g = a^2 = b^3, \quad h = a^{-1}b, \quad \text{where} \quad b \equiv kh'^{-1}k^{-1}. \tag{C.17}$$

The solution to the other two constraints leads to the final answer:

$$\boxed{N_\zeta^l(2,3) = \text{the number of } R_\zeta \times \bar{R}_l \in (E_{(g,h)} \times E_{h'})_{\text{ir}} \text{ contained in } \mathcal{H}_{(g,h,h')}.} \tag{C.18}$$

Here, $\zeta = (C_{(g,h)}, R_\zeta)$ and $l = (C_{h'}, R_l)$. $\mathcal{H}_{(g,h,h')}$ is a representation of $E_{(g,h)} \times E_{h'}$ defined according to the following steps:

1. Choose the $E_{(g,h)}$ and $E_{h'}$ that we want to study. $g, h, h' \in G$ are the representatives specified by the lower indices, $gh = hg$.

2. Find the set of ordered pair $\{a, k\}$, $a, k \in G$ such that Eq. (C.17) holds.

3. $\mathcal{H}_{(g,h,h')}$, as a Hilbert space, is defined as $\text{span}\{|a,k\rangle|\{a,k\}$ from step 2$\}$.

4. $\mathcal{H}_{(g,h,h')}$ is a representation of $E_{(g,h)} \times E_{h'}$ by the group action:

$$\Gamma(t)|a,k\rangle = |t_1 a t_1^{-1}, t_1 k t_2^{-1}\rangle, \quad \forall t = (t_1, t_2) \in E_{(g,h)} \times E_{h'}. \tag{C.19}$$

Table 9: Multiplicities $N_\zeta^l(2,3)$ associated with a solid torus minus a trefoil. The columns are shrinkable loop types ($l \in \mathcal{C}_{\mathrm{loop}}$), and the rows are trefoil excitation types ($\zeta \in \mathcal{C}_{\mathrm{Hopf}}$). Only the nonzero multiplicities are shown.

| $N_\zeta^l(2,3)$    $l$ <br> $\zeta$ | $(C_1, \mathrm{Id}_{S_3})$ | $(C_1, \mathrm{Sign}_{S_3})$ | $(C_1, \Pi_{S_3})$ |
|---|---|---|---|
| $(C_{(1,1)}, \mathrm{Id}_{S_3})$ | 1 | 0 | 0 |
| $(C_{(1,1)}, \mathrm{Sign}_{S_3})$ | 0 | 1 | 0 |
| $(C_{(1,1)}, \Pi_{S_3})$ | 0 | 0 | 1 |
| $(C_{(1,s)}, \mathrm{Id}_{\mathbb{Z}_2})$ | 1 | 0 | 1 |
| $(C_{(1,s)}, \mathrm{Sign}_{\mathbb{Z}_2})$ | 0 | 1 | 1 |

| $N_\zeta^l(2,3)$    $l$ <br> $\zeta$ | $(C_r, \mathrm{Id}_{\mathbb{Z}_3})$ | $(C_r, \omega_{\mathbb{Z}_3})$ | $(C_r, (\omega^2)_{\mathbb{Z}_3})$ |
|---|---|---|---|
| $(C_{(1,r)}, \mathrm{Id}_{\mathbb{Z}_3})$ | 1 | 0 | 0 |
| $(C_{(1,r)}, \omega_{\mathbb{Z}_3})$ | 0 | 0 | 1 |
| $(C_{(1,r)}, (\omega^2)_{\mathbb{Z}_3})$ | 0 | 1 | 0 |
| $(C_{(1,s)}, \mathrm{Id}_{\mathbb{Z}_2})$ | 1 | 1 | 1 |
| $(C_{(1,s)}, \mathrm{Sign}_{\mathbb{Z}_2})$ | 1 | 1 | 1 |

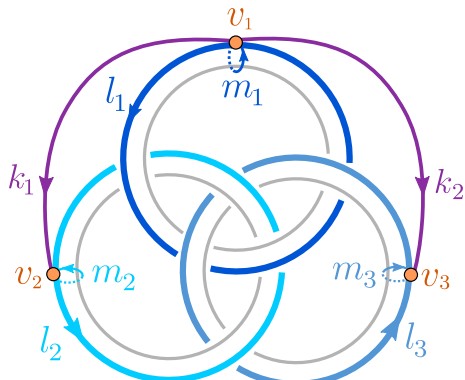

Figure 33: Minimal diagram for Borromean rings complement.

The solution to this problem for the specific finite group $S_3$ is summarized in Table 9.

From Tables 8,7,9, it is not difficult to check the consistency relations between different multiplicities:

$$N_\zeta^a(\text{trefoil}) = \sum_{l \in \mathcal{C}_{\mathrm{loop}}} N_\zeta^l(2,3) N_l^a .  \tag{C.20}$$

This is Eq. (6.2) of the main text.

## C.5 Complement of Borromean Rings in $S^3$

Here, we study the information convex set of the complement of the Borromean rings in $S^3$. We shall denote this region as $\Omega_{\mathrm{Bor}}$. Unlike previous studies of a related topic [53], our physical context is to have three loop excitations on the three Borromean rings. We highlight a feature of non-Abelian models, i.e., nontrivial fusion multiplicities associated with this arrangement of three loops. Below we denote this set of multiplicities as $\{N_{\eta_1,\eta_2,\eta_3} \equiv \dim \mathbb{V}_{\eta_1,\eta_2,\eta_3}(\Omega_{\mathrm{Bor}})\}$.

As with our study of knot complements, we find that the fundamental group of the Borromean rings complement [54] is useful:[26]

$$\pi_1(\Omega_{\text{Bor}}) = \left\{ \mathfrak{m}_1, \mathfrak{m}_2, \mathfrak{m}_3 | [\mathfrak{m}_1, [\mathfrak{m}_2^{-1}, \mathfrak{m}_3]] = 1, [\mathfrak{m}_2, [\mathfrak{m}_3^{-1}, \mathfrak{m}_1]] = 1, [\mathfrak{m}_3, [\mathfrak{m}_1^{-1}, \mathfrak{m}_2]] = 1 \right\}. \quad \text{(C.21)}$$

Here $[\alpha, \beta] \equiv \alpha\beta\alpha^{-1}\beta^{-1}$, which is a particular useful abbreviation to use in this section.

The minimal diagram shown in Fig. 33 contains 6 boundary links (labeled by $m_1, l_1, m_2, l_2, m_3, l_3$), 2 bulk links (labeled by $k_1, k_2$) and 3 vertices $v_1, v_2$ and $v_3$. Faces are not shown.

The zero-flux constraints from the faces in $\Omega_{\text{Bor}}$ then imply:

$$[m_1, [\tilde{m}_2^{-1}, \tilde{m}_3]] = 1, \qquad [\tilde{m}_2, [\tilde{m}_3^{-1}, m_1]] = 1, \qquad [\tilde{m}_3, [m_1^{-1}, \tilde{m}_2]] = 1, \quad \text{(C.22)}$$

$$l_1 = [\tilde{m}_2^{-1}, \tilde{m}_3], \qquad \tilde{l}_2 = [\tilde{m}_3^{-1}, m_1], \qquad \tilde{l}_3 = [m_1^{-1}, \tilde{m}_2], \quad \text{(C.23)}$$

$$\text{where} \quad \tilde{m}_2 \equiv k_1 m_2 k_1^{-1}, \qquad \tilde{m}_3 \equiv k_2 m_3 k_2^{-1}, \qquad \tilde{l}_2 \equiv k_1 l_2 k_1^{-1}, \qquad \tilde{l}_3 \equiv k_2 l_3 k_2^{-1}. \quad \text{(C.24)}$$

The purpose of introducing $\tilde{\phantom{m}}$ is to bring the variables to the same base point. It is easy to see, $\{m_1, \tilde{m}_2, \tilde{m}_3\}$ plays the role of the generators $\{\mathfrak{m}_1, \mathfrak{m}_2, \mathfrak{m}_3\}$, whereas the former set are group elements in $G$.

The solution to this problem is

$$\boxed{\begin{array}{c} N_{\eta_1\eta_2\eta_3} = \text{the number of } R_{\eta_1} \times R_{\eta_2} \times R_{\eta_3} \in (E_{(l_1, m_1^{-1})} \times E_{(l_2, m_2^{-1})} \times E_{(l_3, m_3^{-1})})_{\text{ir}} \\ \text{contained in } \mathcal{H}_{(l_1, m_1, l_2, m_2, l_3, m_3)}. \end{array}} \quad \text{(C.25)}$$

Here, $\eta_i = (C_{(l_i, m_i^{-1})}, R_{\eta_i})$ and $R_{\eta_i} \in (E_{(l_i, m_i^{-1})})_{\text{ir}}$. $\mathcal{H}_{(l_1, m_1, l_2, m_2, l_3, m_3)}$ is a representation of $E_{(l_1, m_1^{-1})} \times E_{(l_2, m_2^{-1})} \times E_{(l_3, m_3^{-1})}$ defined according to the following steps:

1. Choose the $\{E_{(l_i, m_i^{-1})}\}_{i=1}^3$ that we want to study. Here we require $[l_i, m_i] = 1$.

2. Find the set of ordered pair $\{k_1, k_2\}$, $k_1, k_2 \in G$ such that Eqs. (C.22), (C.23) and (C.24) hold.

3. $\mathcal{H}_{(l_1, m_1, l_2, m_2, l_3, m_3)}$, as a Hilbert space, is defined as $\text{span}\{|k_1, k_2\rangle | \{k_1, k_2\} \text{ from step 2}\}$.

4. $\mathcal{H}_{(l_1, m_1, l_2, m_2, l_3, m_3)}$ is a representation of $E_{(l_1, m_1^{-1})} \times E_{(l_2, m_2^{-1})} \times E_{(l_3, m_3^{-1})}$ by the group action:

$$\Gamma(t)|k_1, k_2\rangle = |t_1 k_1 t_2^{-1}, t_1 k_2 t_3^{-1}\rangle, \quad \forall t = (t_1, t_2, t_3) \in E_{(l_1, m_1^{-1})} \times E_{(l_2, m_2^{-1})} \times E_{(l_3, m_3^{-1})}. \quad \text{(C.26)}$$

One can write down a more explicit form of Eq. (C.25) using characters, as

$$N_{\eta_1, \eta_2, \eta_3} = \left( \prod_{i=1}^3 \frac{1}{|E_{(l_i, m_i^{-1})}|} \right) \sum_{\substack{t_i \in E_{(l_i, m_i^{-1})} \\ i=1,2,3}} \chi_{R_{\eta_1}}^*(t_1) \chi_{R_{\eta_2}}^*(t_2) \chi_{R_{\eta_3}}^*(t_3) \chi_{\mathcal{H}}(t_1, t_2, t_3), \quad \text{(C.27)}$$

where

$$\chi_{\mathcal{H}}(t_1, t_2, t_3) = \sum_{\text{allowed } (k_1, k_2)} \delta_{k_1, t_1 k_1 t_2^{-1}} \delta_{k_2, t_1 k_2 t_3^{-1}} \quad \text{(C.28)}$$

is the character for $\mathcal{H}_{(l_1, m_1, l_2, m_2, l_3, m_3)}$ as a representation of $E_{(l_1, m_1^{-1})} \times E_{(l_2, m_2^{-1})} \times E_{(l_3, m_3^{-1})}$.

**Abelian case:** If the group $G$ is Abelian, the set of constraints are translated into

$$l_1 = l_2 = l_3 = 1. \quad \text{(C.29)}$$

---

[26]Comparing to Ref. [54], Eq. (C.21) has a minor difference due to the convention of ordering that we adopt.

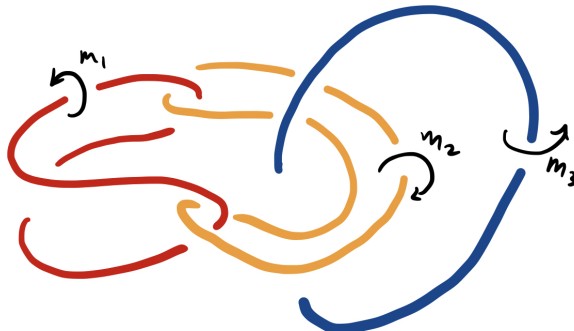

Figure 34: A demonstration that $m_3$ is trivial in the arrangement of loops relevant to the Borromean fusion of pure fluxes of (3.42).

Table 10: Multiplicities relevant to the Borromean fusion process. The data is for the 3d $S_3$ quantum double model. $n_{\mu\nu}^{\lambda} \equiv N_{\varphi(\mu)\varphi(\nu)\varphi^{\vee}(\lambda)}$.

| $\sum_{\lambda} n_{\mu\nu}^{\lambda}\lambda$ \ $\mu$    $\nu$ | $C_1$ | $C_r$ | $C_s$ |
|---|---|---|---|
| $C_1$ | $C_1$ | $C_1$ | $C_1$ |
| $C_r$ | $C_1$ | $2C_1$ | $C_r$ |
| $C_s$ | $C_1$ | $C_r$ | $C_1 + C_r$ |

Meanwhile, $m_1, m_2, m_3, k_1, k_2 \in G$ and they can be chosen independently. The problem can be solved easily, and the intuition is simple. For Abelian models, fluxes can be put on the three rings independently, as if they are "transparent". We can put three pure fluxes $(\mu_1, \mu_2, \mu_3)$ on the three loops, respectively. We can also create three particles $a_1, a_2, a_3$ such that $N_{a_1,a_2,a_3} = 1$. We fuse the point particles onto each flux. So each loop is associated with a pair $(\mu_i, a_i)$; this describes a shrinkable loop sector. (Note that this is only true for Abelian models.) In summary, for Abelian models:

- $N_{\eta_1 \eta_2 \eta_3} = 1$ when $\eta_1, \eta_2, \eta_3 \in \mathcal{C}_{\text{loop}}$ and $R_{\eta_1} \times R_{\eta_2} \times R_{\eta_3} = \text{Id}_G$.

- $N_{\eta_1 \eta_2 \eta_3} = 0$, otherwise.

**non-Abelian case:** The multiplicities are more interesting for non-Abelian models. As usual, we take $S_3$ quantum double as an illustration.

First, let us study the case relevant to the Borromean fusion process of flux loops introduced in §3.5. In that case the generator $m_3$ is trivial (see Fig. 34). The relations (C.22) are then automatically satisfied. The condition relating the inputs of the fusion and its output is the third relation in (C.23): $\tilde{l}_3 = [m_1^{-1}, \tilde{m}_2]$. This is the relation claimed in our discussion in Example 3.15. (Since the inputs are pure fluxes, $l_1$ and $l_2$ are trivial as well, consistent with the first two equations of (C.23)). The multiplicity can be solved by the general procedure described above. In particular, (C.25) can be evaluated by calculating the characters (C.27).

In Table 10, we are interested in a subset of $\{N_{\eta_1, \eta_2, \eta_3}\}$ which are relevant to the Borromean fusion process of flux loops, introduced in Eq. (3.42). For this purpose, we set $\{\eta_1, \eta_2, \eta_3\} = \{\varphi(\mu), \varphi(\nu), \varphi^{\vee}(\lambda)\}$, where $\mu, \nu, \lambda \in \mathcal{C}_{\text{flux}}$. The map $\varphi^{\vee} : \mathcal{C}_{\text{flux}} \hookrightarrow \mathcal{C}_{\text{Hopf}}$ is defined to be $\varphi$ followed by the operation $\eta \to \eta^{\vee}$. In the table, we choose a short-hand notation $n_{\mu\nu}^{\lambda} \equiv N_{\varphi(\mu)\varphi(\nu)\varphi^{\vee}(\lambda)}$. Although the relation between these multiplicities to the Borromean fusion is indirect, one thing can be said concretely. If $n_{\mu\nu}^{\lambda} = 0$ then $\lambda$ cannot be the fusion outcome of $\mu$ and $\nu$ in (3.42).

We note that the entries in table 10 pass a consistency check:

$$\sum_{\lambda \in \mathcal{C}_{\text{flux}}} n^\lambda_{\mu\nu} = \sum_{a \in \mathcal{C}_{\text{point}}} N^a_{\varphi(\mu)} N^{\bar{a}}_{\varphi(\nu)}. \tag{C.30}$$

Here $\{N^a_l\}$ are the multiplicities for the shrinking rule. We omit the derivation since it is a simple variation of Proposition. 6.1.

Finally, let us list the multiplicities when none of $m_1, m_2, m_3$ is the identity group element, for the case of $G = S_3$. For these cases, each of the three rings is occupied by a certain genuine loop excitation.

1. When $\{(l_1, m_1), (l_2, m_2), (l_3, m_3)\} = \{(1, r), (1, r), (1, r)\}$:

$$N_{\eta_1, \eta_2, \eta_3} = \begin{cases} 4, & R_1 = R_2 = R_3 = \text{Id}_{\mathbb{Z}_3}, \\ 2, & R_i = \text{Id}_{\mathbb{Z}_3}, R_{i+1}, R_{i+2} \in \{\omega_{\mathbb{Z}_3}, \omega^2_{\mathbb{Z}_3}\}, \text{ for } i = 1, 2, 3, \\ 0, & \text{otherwise.} \end{cases}$$

Here we set $R_4 = R_1, R_5 = R_2$.

2. When $\{(l_1, m_1), (l_2, m_2), (l_3, m_3)\} = \{(1, s), (r, r), (r, r)\}$, for all choices of $R_1, R_2$ and $R_3$, we have $N_{\eta_1, \eta_2, \eta_3} = 1$.

3. When $\{(l_1, m_1), (l_2, m_2), (l_3, m_3)\} = \{(1, s), (r, r^2), (r, r^2)\}$, for all choices of $R_1, R_2$ and $R_3$, we have $N_{\eta_1, \eta_2, \eta_3} = 1$.

4. When $\{(l_1, m_1), (l_2, m_2), (l_3, m_3)\} = \{(1, s), (1, s), (1, s)\}$

$$N_{\eta_1, \eta_2, \eta_3} = \begin{cases} 1, & R_1 = R_2 = R_3 = \text{Id}_{\mathbb{Z}_2}, \\ 1, & R_i = \text{Id}_{\mathbb{Z}_2}, R_{i+1} = R_{i+2} = \text{Sign}_{\mathbb{Z}_2}, \text{ for } i = 1, 2, 3, \\ 0, & \text{otherwise.} \end{cases}$$

These exhaust all nontrivial cases for $G = S_3$ when each of the three loops is occupied by a genuine loop excitation.

# D  Useful entropy combinations

The main goal of this appendix is a coherent summary of useful entropy conditions on the reference states. We introduce the following short-hand notations:

$$\Delta(B, C)_\rho \equiv (S_{BC} + S_C - S_B)_\rho, \tag{D.1}$$
$$\Delta(B, C, D)_\rho \equiv (S_{BC} + S_{CD} - S_B - S_D)_\rho. \tag{D.2}$$

In Sec. D.1, we identify a set of conditions of the reference state $\sigma$ that will be useful to prove things; as corollaries, the value for other extreme points can be inferred. In Sec. D.2, we discuss a few equivalent definitions of topological entanglement entropy coming from various partitions.

We recall the following useful conditions, which follow from SSA:

$$\begin{aligned} \Delta(B, C) &\geq 0, \\ \Delta(B, C) &\geq \Delta(BB', C), \\ \Delta(B, CE) &\geq \Delta(BE, C), \\ \Delta(B, C, D) &\geq 0, \\ \Delta(B, C, D) &\geq \Delta(BB', C, DD'), \\ \Delta(B, CEF, D) &\geq \Delta(BE, C, DF). \end{aligned} \tag{D.3}$$

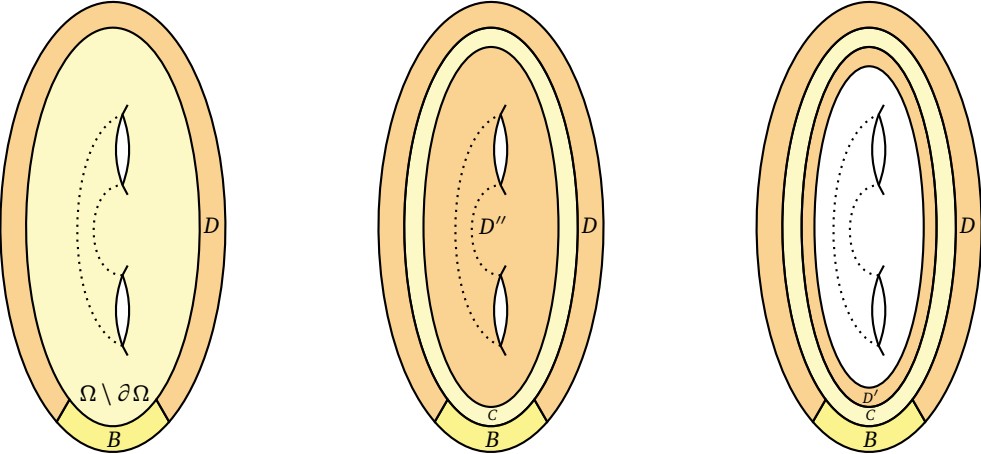

Figure 35: Regions showing the idea of proving Eq. (D.4) of the decoupling lemma. $\partial\Omega = BD$, $\partial(\Omega\setminus(BD)) = C$, $D'' = \Omega\setminus(BCD)$ and $D' = \partial D''$. The partition $\partial\Omega$ into $BD$ is arbitrary. In general, $\partial\Omega$ is allowed to have multiple connected components.

## D.1 Useful entropy combinations

The main purpose of this section is to derive and summarize a set of conditions satisfied by the reference state $\sigma$, of the form $\Delta(B,C,D)_\sigma = 0$ and $\Delta(B,C)_\sigma = 0$. Since the combinations of entropies are of similar form to the axioms **A0** and **A1**, this can be regarded as analogs of the axioms on more diverse topologies. While the conditions studied here are less fundamental than the axioms in that they are derived properties, they are handy in proofs.

Finally, we derive the entropy combination for all extreme points as a corollary. A certain condition of this form can serve as an alternative definition of quantum dimension, which does not require the existence of a vacuum sector.

We start with the *decoupling lemma* for entropy conditions, which uncovers some connections between these conditions on different configurations.

**Lemma D.1** (Decoupling lemma). *Let $\Omega$ be an immersed region. Let $\rho_\Omega^{\langle e\rangle}$ be an extreme point of $\Sigma(\Omega)$. Let $\partial\Omega = BD$, $\partial(\Omega\setminus(BD)) = C$ and $D' = \partial(\Omega\setminus(BCD))$. (A possible choice of regions is illustrated in Fig. 35.) Then*

$$\Delta(B,\Omega\setminus\partial\Omega,D)_{\rho^{\langle e\rangle}} = \Delta(B,C,DD')_{\rho^{\langle e\rangle}}. \tag{D.4}$$

$$\Delta(B,\Omega\setminus\partial\Omega,D)_{\rho^{\langle e\rangle}} = \Delta(BD',C,D)_{\rho^{\langle e\rangle}}. \tag{D.5}$$

The remarkable fact is that on the right-hand side of Eq.(D.4), all regions are near the boundary of region $\Omega$. The result is independent of the detailed partition of $\partial\Omega$ into $BD$, and it is flexible enough to cover the cases where $\Omega$ has multiple boundary components and various higher dimensional settings.

*Proof.* The idea of the proof of Eq. (D.4) is illustrated by the color setting of Fig. 35. First, we show

$$\Delta(B,\Omega\setminus\partial\Omega,D)_{\rho^{\langle e\rangle}} = \Delta(B,C,DD'')_{\rho^{\langle e\rangle}}. \tag{D.6}$$

This is true because for the extreme point $\rho_\Omega^{\langle e\rangle}$,

$$\begin{aligned} S_{DD''} &= S_D + S_{D''}, \\ S_{BC} - S_{D''} &= S_{B(\Omega\setminus\partial\Omega)}. \end{aligned} \tag{D.7}$$

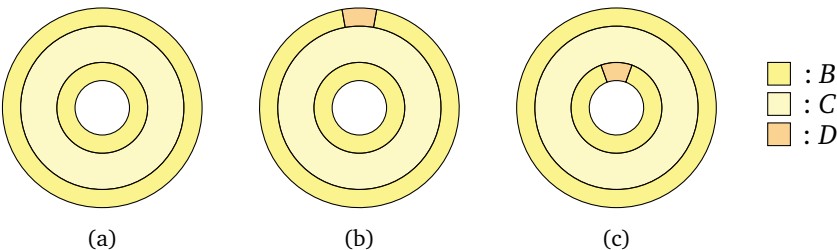

Figure 36: An annulus embedded in a disk and its three basic partitions. For all three partitions, $BD$ ($D = \emptyset$ for (a)) is the thickened boundary of the annulus, whereas $C$ is the interior.

Both conditions follow from the factorization property and SSA. Second, we find that

$$\Delta(B,C,DD'')_{\rho^{\langle e \rangle}} = \Delta(B,C,DD')_{\rho^{\langle e \rangle}}. \tag{D.8}$$

This is because for the extreme point $\rho_{\Omega}^{\langle e \rangle}$,

$$
\begin{aligned}
S_{DD'} - S_{DD''} &= S_{D''\setminus D'}, \\
S_{CDD'} - S_{CDD''} &= S_{D''\setminus D'}.
\end{aligned}
\tag{D.9}
$$

Both conditions follow from the factorization property and SSA. This completes the proof of Eq. (D.4).

The proof of Eq. (D.5) is similar since $\Delta(B,C,D)$ is invariant under the exchange of $B$ and $D$. $\qquad\square$

### D.1.1 Two-dimensional regions

**Proposition D.2.** *Let $\sigma$ be a 2d reference state on a disk (which satisfies axioms* **A0** *and* **A1***). The entropy combinations*

1. $\Delta(B,C)_\sigma = 0$ *for the partition in Fig. 36(a).*

2. $\Delta(B,C,D))\sigma = 0$ *for the partition in Fig. 36(b) and (c).*

*Proof.* To prove the first statement, $\Delta(B,C)_\sigma = 0$ for Fig. 36(a), we consider a ball for which enlarged **A0** holds. We then use the decoupling lemma to go the sphere shell. Now the left-hand side of Eq. (D.5) becomes zero (due to enlarged **A0**, and we have set $D = \emptyset$); the right-hand side becomes the desired answer. An alternative method of proving the first statement is to use the vacuum lemma (Lemma 3.2) and the extreme point criterion (Lemma 2.13).

$\Delta(B,C,D)_\sigma = 0$ for Fig. 36(b) follows from enlarged **A1** and the decoupling lemma. The logic is analogous to the first method described in the previous paragraph.

$\Delta(B,C,D)_\sigma = 0$ for Fig. 36(c) follows from that for Fig. 36(b) and the sphere completion lemma. The useful observation is that one can flip the inside and the outside of an annulus by smoothly deforming it on a sphere. $\qquad\square$

With these relatively simple conditions, one can derive more conditions by applying the decoupling lemma. Below are some examples.

**Example D.3.** $\Delta(B,C,D)_\sigma = 0$ for any $BCD$ configuration in Fig. 37.

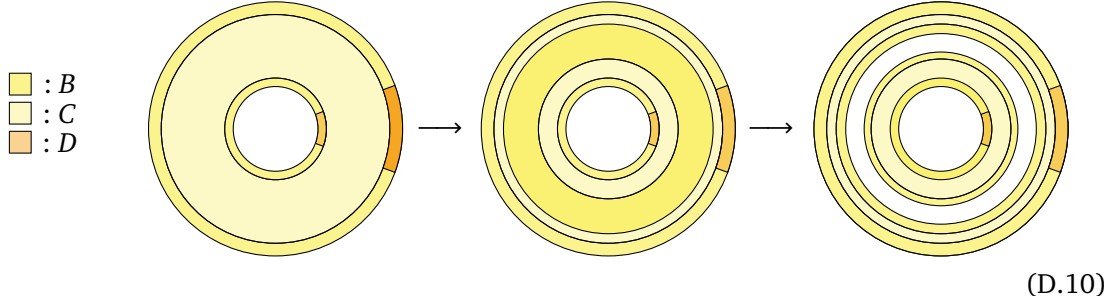

Figure 37: All regions are contained in a disk. For all these partitions, $BD$ is the thickened boundary and $C$ is the interior.

To illustrate the idea, here are the steps that justifies case (a) of Fig. 37:

$$\tag{D.10}$$

The three figures here correspond to the three in Fig. 35 translated to this particular case.

**Quantum dimension in 2d.** Below, we obtain an expression for the quantum dimension by replacing the reference state with an appropriate extreme point.

**Proposition D.4.** *For the extreme points with labels specified in the table:*

| $\Delta(B,C,D)$ | when | 2d regions |
|:---:|:---:|:---:|
| $2\ln d_a$ | $\forall a \in \mathcal{C}$ | |
| $4\ln d_a$ | $\forall a \in \mathcal{C}$ | |
| $2\ln d_a$ | $\forall a,b,c \in \mathcal{C}$ | |
| $2(\ln d_a + \ln d_b)$ | $\forall a,b,c \in \mathcal{C}$ | |
| $2(\ln d_a + \ln d_b + \ln d_c)$ | $\forall a,b,c \in \mathcal{C}$ | |

☐ : $B$
☐ : $C$
☐ : $D$

*Proof.* This is proved by three steps: (1) Use the decoupling lemma to reduce the problem to the thickened boundary of the region in question, (2) take entropy differences between the extreme point in question and the reference state, and (3) use the definition of the quantum dimension (2.10). □

**Remark.** One remarkable thing lies behind Proposition D.4. By calculating the entropy of a single wave function, one can identify the quantum dimension of an individual superselection sector. This proposition is a useful alternative definition of the quantum dimension compared to Eq. (2.10). This definition can work in contexts not covered by Eq. (2.10), e.g., for immersed regions and in the presence of defects, where the existence of a vacuum sector is not guaranteed.

Furthermore, as a corollary, one can easily infer that the quantum dimension

$$d_a \geq 1, \quad \forall a \in \mathcal{C}, \tag{D.11}$$

which is a consequence of SSA, namely $\Delta(B,C,D) \geq 0$. While this may also be derived from the highly constraining fusion rules [1], this new logic is succinct and elementary. We shall see the usefulness of this logic in 3d later.

### D.1.2 Three-dimensional regions

**Proposition D.5.** *Let $\sigma$ be a 3d entanglement bootstrap reference state (which satisfies the 3d version of axioms* **A0** *and* **A1***). The entropy combinations*

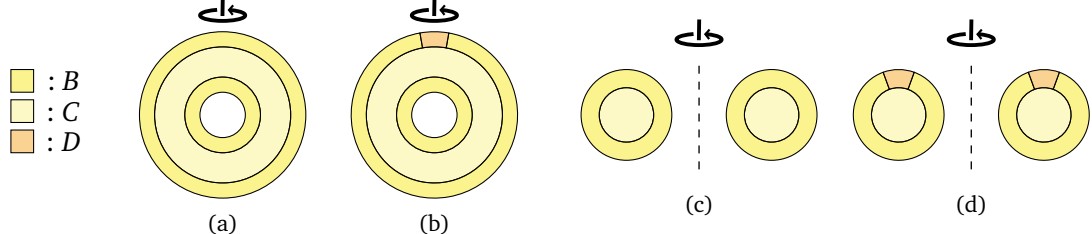

Figure 38: A few basic partitions for 3d. The subsystems are partitions of either a sphere shell or a solid torus. These regions are embedded in a ball.

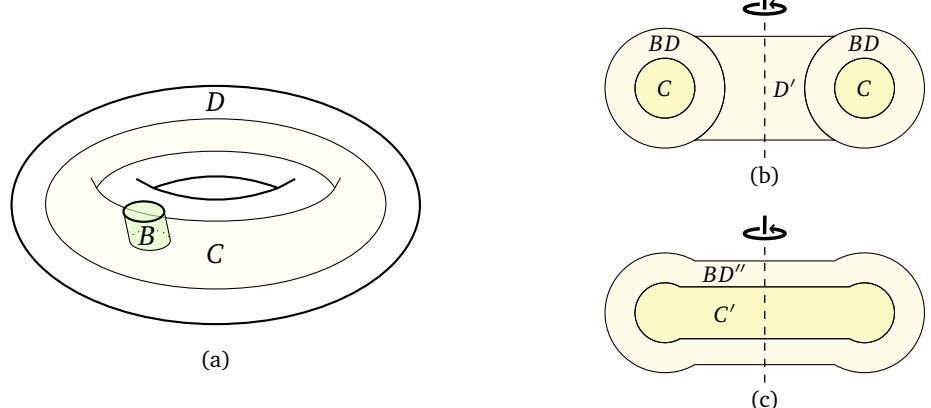

Figure 39: (a) A solid torus $T$ embedded in a ball and its partition $T = BCD$. (This is a reprint of Fig. 20.) (b) Add disk $D'$. (c) An illustration of $C' \supset C$ and $D'' \subset DD'$, where $BC'D''$ is a partition of a ball for which an enlarged **A1** ($\Delta(B, C', D'')_\sigma = 0$) holds. Note that $C' \cap B = \emptyset$.

1. $\Delta(B, C)_\sigma = 0$ *for Fig. 38(a) and (c).*

2. $\Delta(B, C, D)_\sigma = 0$ *for Fig. 38(b) and (d).*

*Proof.* The statement $\Delta(B, C)_\sigma = 0$ for sphere shell Fig. 38(a) can follow from enlarged **A0** and the decoupling lemma. Here, **A0** is on a ball that contains the sphere shell, and the decoupling lemma allows us to remove the interior of the ball, getting the condition we look for. By an alternative method, the vacuum lemma (Lemma 3.2) and the extreme point criterion (Lemma 2.13), one can see $\Delta(B, C)_\sigma = 0$ is true for both Fig. 38(a) and (c).

The statement that $\Delta(B, C, D)_\sigma = 0$ for Fig. 38(b) follows from enlarged **A1** and the decoupling lemma. Again, the decoupling lemma allows us to remove the interior of the ball. $\Delta(B, C, D)_\sigma = 0$ for Fig. 38(d) is a special case of Lemma 5.7, which is proved in the main text. $\qquad\square$

**Proposition D.6.** *Let $\sigma$ be a reference state on a ball. The entropy combination $\Delta(B, C, D)_\sigma = 0$ for the partition of solid torus $T = BCD$ shown in Fig. 39(a).*

*Proof.* Since $\sigma$ is the reference state, we can reversibly fill in the "hole" of the solid torus $T$ with a region $D'$ (which does not touch $B$); see Fig. 39(b). This means

$$\Delta(B, C, D)_\sigma = \Delta(B, C, DD')_\sigma. \tag{D.12}$$

Note that $TD'$ is a ball. Now we extend $C$ into $C' \supset C$ ($C' \cap B = \emptyset$) shown in Fig. 39(c) and let $D'' \equiv CDD' \setminus C'$. Now $BD''$ is a sphere shell surrounding $C'$. The right-hand side of Eq. (D.12) is then constrained to vanish:

$$
\begin{aligned}
\Delta(B, C, DD')_\sigma &\le \Delta(B, C', D'')_\sigma \\
&= 0.
\end{aligned}
\tag{D.13}
$$

The first line is a consequence of SSA (the last line of Eq. (D.3)). The second line follows from enlarged **A1**. This completes the proof. □

Below is a statement for a generic coprime $(p, q)$ partition of a solid torus, which generalizes Lemma 5.7 of the main text.

**Proposition D.7.** *Let $\sigma$ be a reference state on a ball. The entropy combination $\Delta(B, C, D)_\sigma = 0$ for the partition of solid torus $T = BCD$ shown in Fig. 40, for any pair of relatively-prime integers $(p, q)$.*

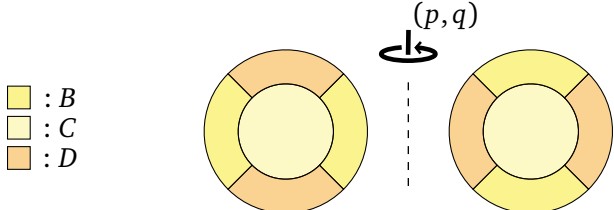

Figure 40: The torus $T = BCD$ is embedded in a ball. The partition involves a $(p, q)$ rotation, where $p$ and $q$ are coprime integers.

*Proof.* The special case of $p = 1$ (or $q = 1$) is proved in Lemma 5.7. The idea is to use the sphere completion lemma, and consider the decomposition of the 3-sphere as $S^3 = T \cup \widetilde{T}$, where $\partial T = \partial \widetilde{T} = BD$. To solve the generic coprime $(p, q)$, we shall convert the problem to the simpler cases $((1, m)$ or $(n, 1))$ using deformations of solid tori through a sequence of nontrivial immersion.

This key technique is a well-known (and popular) way to show *"in the spherical braid group the coil has order two"* (see figure 4 on Page 133 of Ref. [55]). This trick is also known as "the Dirac belt trick" [56]. In our setup, this trick can be phrased as: it is possible to add a $4\pi$ twist to a closed ribbon through immersion in 3d space;[27] see Fig. 41 for an illustration.

If we think of the thickening of the closed ribbon in Fig. 41 as the solid torus $T = BCD$ that we are interested in, then $T$ can map back to $T$. Furthermore, with the technique in Fig. 19 we see that the reference state $\sigma_T$ is mapped back to itself. Nonetheless, the process induces a change on the topological class of the partition $BCD$, and $(p, q)$ changes:

$$
(p, q) \to (p, q \pm 2p).
\tag{D.14}
$$

The reason is that there are $p$ braids in the spiral region $D$ and each is twisted by $4\pi$. Furthermore, by the sphere completion lemma, there is a "dual" solid torus $\widetilde{T} = BAD$, where $A = S^3 \setminus BCD$. Applying the same trick to $\widetilde{T}$, we have

$$
(p, q) \to (p \pm 2q, q).
\tag{D.15}
$$

Now it is clear that if we start from an arbitrary coprime $(p, q)$, we can always end up with $(1, 0)$, $(0, 1)$, $(1, 1)$ or $(1, -1)$. This is because one of the operations in Eq. (D.14) and (D.15) can decrease the absolute value of the entry that has a larger absolute value, and that $(p, q)$ is equivalent to $(-p, -q)$. This completes the proof. □

---

[27] One important point is that $4\pi$ is possible while $2\pi$ is not.

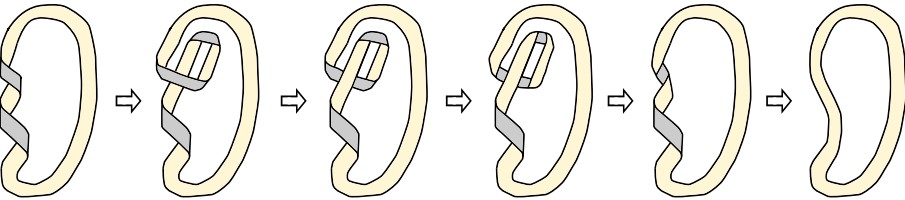

Figure 41: An illustration of the belt trick, which can add $\pm 4\pi$ twist to a closed ribbon. Note that the second step needs immersion.

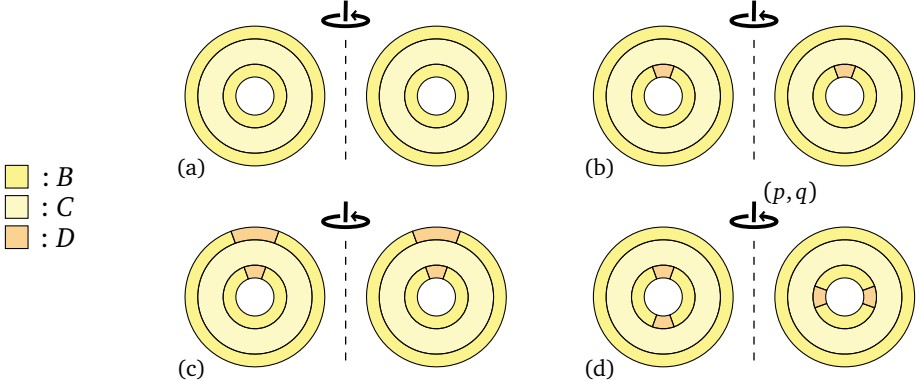

Figure 42: All configurations are contained in a ball for which the reference state is defined. For (d), any coprime pair $(p,q)$ is allowed while the depiction is accurate for $p = 2$.

Here are a few more examples. They follow directly from the decoupling lemma and the basic configurations discussed above. The sphere completion lemma is needed in some cases.

**Example D.8.** The following statements hold:

1. $\Delta(B,C)_\sigma = 0$ for Fig. 42(a).

2. $\Delta(B,C,D)_\sigma = 0$ for Fig. 42(b) and (c).

3. $\Delta(B,C,D)_\sigma = 0$ for Fig. 42(d), for any coprime $(p,q)$.

**Quantum dimension in 3d:** Similar to 2d cases, the quantum dimensions of various superselection sectors show up when considering an extreme point.

**Proposition D.9.** *For the extreme points with labels specified in the table:*

| $\Delta(B,C,D)$ | when | 3d regions |
|---|---|---|
| $2\ln d_a$ | $\forall a \in \mathcal{C}_{\text{point}}$ | |
| $0$ | $\forall \mu \in \mathcal{C}_{\text{flux}}$ | |
| $2\ln d_\mu$ | $\forall \mu \in \mathcal{C}_{\text{flux}}$ | |
| $4\left(\ln d_\mu - \ln d_{t_p(\mu)}\right)$ | $\forall \mu \in \mathcal{C}_{\text{flux}}$, $\forall(p,q)$ coprime | |
| $2\left(\ln d_\eta - \ln d_\mu^2\right)$ | $\forall \eta \in \mathcal{C}_{\text{Hopf}}$, $\mu = t_{(1,0)}(\eta)$ | |
| $4\left(\ln d_\eta - \ln d_\mu^2\right)$ | $\forall \eta \in \mathcal{C}_{\text{Hopf}}$, $\mu = t_{(1,0)}(\eta)$ | |
| $2\left(\ln d_\eta - \ln d_{t_{(p,q)}(\eta)}^2\right)$ | $\forall \eta \in \mathcal{C}_{\text{Hopf}}$ | |

*Here $t_p : \mathcal{C}_{\text{flux}} \to \mathcal{C}_{\text{flux}}$ is the spiral map on fluxes, as is defined in Eq. (5.6); $t_{(p,q)} : \mathcal{C}_{\text{Hopf}} \to \mathcal{C}_{\text{flux}}$ is the map defined in Eq. (5.19).*

Among these statements, the one for Fig. 39(a), (i.e. condition 3) requires some explanation. Below is the proof of it. (We omit the proof of other conditions because they follow directly from Eq. (2.10), the definition of the quantum dimension.)

*Proof.* Because $\Delta(B,C,D)_\sigma = 0$ for the partition in Fig. 39(a) (Proposition D.6), all we need is to calculate the entropy differences between the extreme point labeled by $\mu$ and the vacuum 1. Below we denote the reduced density matrix of extreme point $\rho_T^\mu$ on its subsystem $X$ as $\rho_X^\mu$. Since $B$ is a ball,

$$\Delta S_B \equiv S(\rho_B^\mu) - S(\sigma_B) = 0. \tag{D.16}$$

Both $BC$ and $CD$ are solid tori, so

$$\Delta S_{BC} \equiv S(\rho_{BC}^\mu) - S(\sigma_{BC}) = 2\ln d_\mu. \tag{D.17}$$

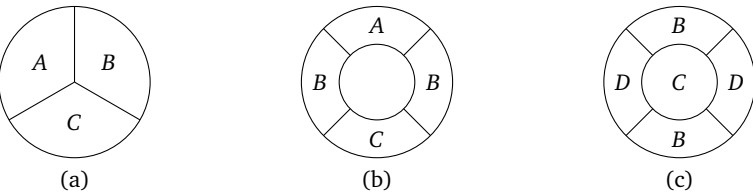

Figure 43: Partitions for 2d TEE. All regions are subsystems of a disk.

$$\Delta S_{CD} \equiv S(\rho_{CD}^{\mu}) - S(\sigma_{CD}) = 2 \ln d_{\mu}. \tag{D.18}$$

Finally, $D$ is a genus-two handlebody. But because there are no excitations in $C$, there is a reversible quantum channel taking the state of $D$ to a solid torus in the sector $\mu$. This channel preserves entropy differences, so

$$\Delta S_D \equiv S(\rho_D^{\mu}) - S(\sigma_D) = 2 \ln d_{\mu}. \tag{D.19}$$

Putting these together, we have $\Delta(B, C, D)_{\rho^{\mu}} = \Delta S_{BC} + \Delta S_{CD} - \Delta S_B - \Delta S_D = 2 \ln d_{\mu}$, which completes the proof of condition 3. $\qquad\square$

**Corollary D.9.1** (Bounds for quantum dimensions)**.** *The following statements hold:*

1. *$d_a \geq 1$, for any $a \in \mathcal{C}_{\text{point}}$.*

2. *$d_{\mu} \geq 1$ for any $\mu \in \mathcal{C}_{\text{flux}}$.*

3. *$d_{\mu} \geq d_{t_p(\mu)}$, for any $\mu \in \mathcal{C}_{\text{flux}}$ and any integer $p$.*

4. *$d_{\eta} \geq d_{t_{(p,q)}}^2(\eta)$, for any $\eta \in \mathcal{C}_{\text{Hopf}}$.*

## D.2 Equivalent definitions of topological entanglement entropy

In this appendix, we discuss a few equivalent definitions of the topological entanglement entropy (TEE) in 2d and 3d. Below $\sigma$ is a reference state on a large enough disk (ball). Our statement can be inferred from previous literature [6, 7, 38] if the familiar form of strict area law $S(A) = \alpha \ell - \gamma$ is assumed. The purpose of this appendix is to derive the statements merely from axioms **A0** and **A1** on bounded radius disks (balls).

### D.2.1 2d topological entanglement entropy

**Proposition D.10.** *The following definitions of 2d TEE are equivalent:*

1. *In terms of Fig. 43(a), $\gamma = (S_{AB} + S_{BC} + S_{CA} - S_A - S_B - S_C - S_{ABC})_{\sigma}$.*

2. *In terms of the subsystems in Fig. 43(b), $\gamma = \frac{1}{2} I(A : C | B)_{\sigma}$.*

3. *In terms of the subsystems in Fig. 43(c), $\gamma = \frac{1}{2} \Delta(B, C, D)_{\sigma}$.*

*Proof.* First of all, each linear combination is a topological invariant. This is because axiom **A1** implies that smoothly deforming the boundary of the regions preserves the entropy combination. This argument can be found in Fig. 4 and 5 of [1]; the proof of $1 \Leftrightarrow 2$ is done in Proposition 5.2 of the same reference. Below we show $2 \Leftrightarrow 3$.

Because of the sphere completion lemma, one can do the analysis on a sphere. Let the complement of $BCD$ of Fig. 43(c) to be $A = S^2 \setminus (BCD)$. Let the reference state on the sphere be $|\psi_{S^2}\rangle$. We immediately see

$$I(A : C | B)_{|\psi_{S^2}\rangle} = \Delta(B, C, D)_{\sigma}. \tag{D.20}$$

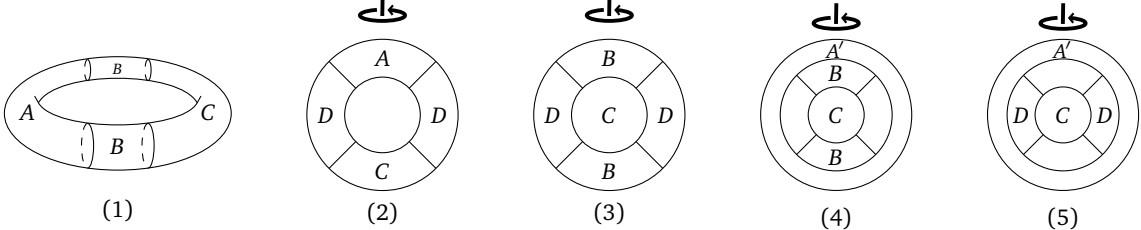

Figure 44: Partitions for 3d TEE. All regions are subsystems of a ball.

Now the *ABC* partition is the same as that in Fig. 43(b). Thus, $2 \Leftrightarrow 3$. This completes the proof. □

### D.2.2   3d topological entanglement entropy

**Proposition D.11.** *The following definitions of 3d TEE are equivalent:*

1. *In terms of Fig. 44(1)* $\gamma = \frac{1}{2}I(A:C|B)_\sigma$. *Here ABC is a solid torus.*

2. *In terms of Fig. 44(2)* $\gamma = \frac{1}{2}I(A:C|D)_\sigma$. *Here ADC is a sphere shell.*

3. *In terms of Fig. 44(3)* $\gamma = \frac{1}{2}\Delta(B,C,D)_\sigma$.

4. *In terms of Fig. 44(4)* $\gamma = \frac{1}{2}I(A':C|B)_\sigma$.

5. *In terms of Fig. 44(5)* $\gamma = \frac{1}{2}I(A':C|D)_\sigma$.

*Proof.* Same as the 2d cases, each linear combination is invariant under smooth deformation of the subsystems. We now apply the sphere completion lemma to obtain a pure reference state $|\psi_{S^3}\rangle$ on a 3-sphere, $S^3 = ABCD$. Here $|\psi_{S^3}\rangle$ is the completion of reference state $\sigma$ to $S^3$. It follows that,

$$I(A:C|B)_{|\psi_{S^3}\rangle} = I(A:C|D)_{|\psi_{S^3}\rangle} = \Delta(B,C,D)_{|\psi_{S^3}\rangle}. \tag{D.21}$$

The geometry of *ABC*, *ADC* and *BCD* are the same as that in item 1, 2 and 3, respectively. This proves that $1 \Leftrightarrow 2 \Leftrightarrow 3$.

Next, we prove $1 \Leftrightarrow 4$ and $2 \Leftrightarrow 5$. Let us do the analysis on the 3-sphere $S^3 = ABCD$. Let $A = A'A''$, where $A''$ is the ball that is surrounded by sphere shell $A'$. The strong subadditivity implies that

$$I(A:C|B)_{|\psi_{S^3}\rangle} \geq I(A':C|B)_{|\psi_{S^3}\rangle}. \tag{D.22}$$

This "$\geq$" can actually be replaced by "$=$". This is because the enlarged **A0** implies $\Delta(A',A'')_{|\psi_{S^3}\rangle} = 0$, which bounds the difference between the two sides of (D.22) to zero. This proves that $1 \Leftrightarrow 4$. The proof for $2 \Leftrightarrow 5$ is analogous (done by replacing *B* with *D*). This completes the proof. □

## E   Dimensional reduction and beyond

In this appendix, we develop a dimensional reduction point of view, which provides additional insight into some of the basic fusion data identified in §3.2 and §3.3. This dimensional reduction picture (Appendix E.1) is rigorous in the sense that we are able to obtain a 2d entanglement bootstrap reference state, and from there, we can "bootstrap" the related 2d

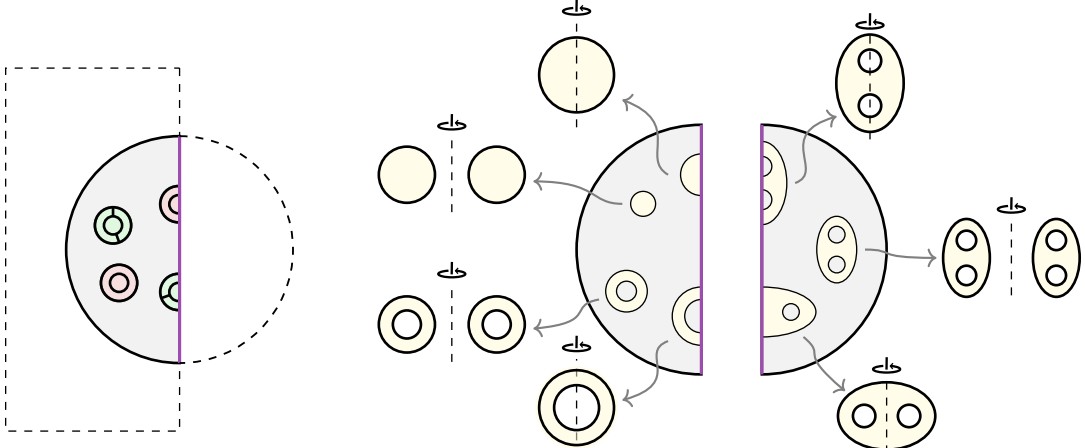

Figure 45: (Vacuum reduction) The reference state on a ball can be viewed as a 2d entanglement bootstrap reference state on a disk adjacent to a gapped boundary. The gapped boundary is shown in purple. Various superselection sectors and fusion multiplicities can be understood from this dimensional reduction picture.

superselection sectors and fusion rules. We then map these objects back to 3d. We summarize the facts about antiparticles, quantum dimensions, and the fusion rules associated with these basic data (Appendix E.2).

On the other hand, there are constraints on 3d fusion data that are beyond the description of the dimensional reduction picture that we develop. One important class of such constraints comes from topologically nontrivial paths of deforming a region back to itself. We discuss some of these constraints in Appendix E.3.

## E.1 Dimensional reduction

We initiate a study of dimensional reduction in the entanglement bootstrap program. We shall describe two rigorous dimensional reduction procedures. In §E.1.1 we describe a dimensional reduction of the vacuum sector (vacuum reduction); the end result is a 2d system with a gapped boundary. In §E.1.2 we describe the dimensional reduction for an arbitrary flux sector (flux reduction). Open questions about yet-to-be-understood (possibly inequivalent) dimensional reduction procedures are discussed.

### E.1.1 Dimensional reduction of the vacuum

Below, we show that the reference state on a 3d ball can be viewed as a reference state on a 2d disk adjacent to a gapped boundary. This is done by making connections to the 2d region that undergoes the revolution; see Fig. 45 for an illustration. We shall see, on this 2d reference state, both the bulk axioms and the boundary axioms are satisfied.[28] The 3d versions of **A0** and **A1** are enough for the dimensional reduction to work. No rotational symmetry is assumed.

This procedure associates a region in 2d with its revolution in 3d. The Hilbert space and the quantum state of the 3d region are carried over to the associated 2d region. The dimensional reduction, thus, provides a quantum state in 2d. In fact, this quantum state is a valid 2d entanglement bootstrap reference state. (For this case, both the bulk axioms and the boundary axioms are satisfied.) To see this, we observe that:

---

[28]A gapped boundary is a gapped domain wall [2, 57] separating the vacuum and a 2d topological order.

- A disk in the bulk of the 2d system corresponds to a solid torus in 3d. The 2d bulk version of **A0** and **A1** follow from Proposition D.5.

- A disk adjacent to the boundary of the 2d system corresponds to a ball in 3d. The 2d boundary version of **A0** (**A1**) follows directly from the 3d version of **A0** (**A1**) for its preimage.

**Remark.** Suppose $\Omega_\downarrow$ is a 2d region, and $\Omega$ is its revolution. Then the information convex set $\Sigma(\Omega_\downarrow)$, defined for the 2d reference state, is, in general, not isomorphic to the information convex set $\Sigma(\Omega)$ of the 3d region defined for the 3d reference state. In general, $\Sigma(\Omega_\downarrow)$ is a smaller set. This is because a state in $\Sigma(\Omega_\downarrow)$ is indistinguishable from the 2d reference state on a set of small balls, and when viewed in 3d, this state is indistinguishable from the 3d reference state on a set of thin solid tori.

With this knowledge, we can apply 2d entanglement bootstrap [1,2] to understand various superselection sectors in 3d. Superselection sectors and fusion processes understood with this picture are:

1. $\mathcal{C}_{\text{point}}$ corresponds to the boundary excitations.

2. $\mathcal{C}_{\text{loop}}$ corresponds to the anyons. This is because an extreme point in the information convex set of a 2d annulus corresponds to a state in the information convex set of a 3d torus shell (the revolution of the 2d annulus), labeled by an element in $\mathcal{C}_{\text{Hopf}}^{[1]}$.

3. The shrinking rule Eq. (3.17) corresponds to fusing an anyon onto the boundary.

4. The $\mathcal{C}_{\text{flux}}$ corresponds to the subset of anyons that can condense onto the boundary. (The quantum dimensions of the fluxes, however, have an extra square root, by our convention.) This is the dimensional reduction view of the embedding $\mathcal{C}_{\text{flux}} \overset{\varphi}{\hookrightarrow} \mathcal{C}_{\text{loop}}$, considered in Eq. (3.10).

The existence of anti-sectors and consistency relations of the fusion rules can be seen with this line of reasoning. See Appendix E.2 for the details.

**Remark.** A few remarks are in order:

1. This vacuum reduction can be implemented on $S^3$ as well. The result is a pure reference state on a disk $D^2$. It has an entire boundary. The bulk/boundary versions of the axioms are satisfied for the entire disk.

2. Because the 2d reference state obtained in this way allows a gapped boundary, the system cannot be chiral. From the recent perspective [12], this manifests in the fact that the modular commutator is zero.

3. The existence of a gapped boundary of the dimensional reduction does not imply the existence of a gapped boundary of the 3d system. It remains an open question whether the following conjecture in Ref. [33], "*all 3+1D bosonic topological orders have gappable boundary.*" can be verified in the framework of entanglement bootstrap.

### E.1.2 Dimensional reduction of flux sectors

Below, we introduce a more flexible dimensional reduction. It generates a reference state $\sigma^{[\mu]}$ on a 2d disk (within the bulk), for each flux $\mu \in \mathcal{C}_{\text{flux}}$. The idea is to take an extreme point of a solid torus and make use of the revolution; see Fig. 46 for an illustration. The 2d bulk version

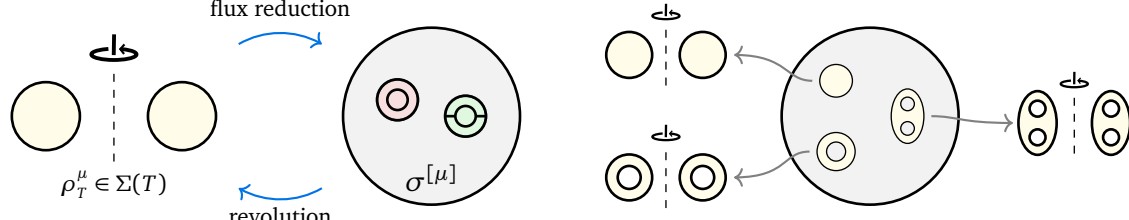

Figure 46: (Flux reduction) An extreme point of the solid torus ($\rho_T^\mu \in \Sigma(T)$) can be viewed as a valid entanglement bootstrap reference state in 2d ($\sigma^{[\mu]}$), defined on a disk; $\sigma^{[\mu]}$ satisfies axioms **A0** and **A1** in the 2d bulk. Various superselection sectors and fusion spaces of 3d can be understood by this picture.

of axioms **A0** and **A1** are checked explicitly. This is because the revolution of a disk in the 2d bulk is a solid torus, and these conditions are verified by Lemma 2.13 and Proposition D.9.

The set $\mathcal{C}_{\text{Hopf}}^{[\mu]}$ is mapped to the anyons in the 2d entanglement bootstrap problem defined by the reference state $\sigma^{[\mu]}$. Thus, it makes sense to talk about the fusion and braiding of these loop excitations from this dimensional reduction viewpoint. For $\mu = 1$, this corresponds to the fusion and braiding of shrinkable loops. This has been covered in the vacuum reduction above. When $\mu \neq 1$, this corresponds to the fusion and braiding of two loops that are linked with a third loop. The configuration of the loop excitations is the same as that in the 3-loop braiding statistics [29, 30].

**Remark.** Here are a few open problems:

1. It remains an open problem if the 2d reference state $\sigma^{[\mu]}$ allows a gapped boundary for $\mu \neq 1$.

2. Another set of dimensional reduction, for each flux, can be obtained by taking the rotation in Fig. 46 be the $(1, 1)$ rotation, instead of the ordinary $(1, 0)$ rotation.[29] We do not know if this set of dimensional reductions is different or not.

3. The dimensional reduction with $(1, 1)$ is already interesting for the vacuum sector. It provides another vacuum reduction. The $S^3$ has a global Hopf fibration, and therefore this dimensional reduction maps a reference state on $S^3$ to a reference state on $S^2$. This dimensional reduction does not generate a boundary explicitly. It is an open question (1) if the 2d reference state so obtained allows a gapped boundary, and (2) if it is within the same phase as the vacuum reduction shown in Fig. 45.

4. Even more generally, a solid torus can be obtained by a revolution labeled by a pair $(p, q)$, where $p$ and $q$ are coprime integers. Every such revolution provides a quantum state on a 2d disk. This is more general since it covers new cases, e.g., $(2, 1)$ and $(2, 3)$. For the most general choice of flux, however, **A1** can break for a special disk in 2d.[30] These dimensional reductions might provide references with a non-Abelian anyon, or possibly a topological defect [22, 58], on the disk. We leave this for future investigation.

5. Another open question is whether it is possible to come up with a dimensional reduction such that the 2d system is on a non-orientable surface (possibly with boundaries).

---

[29]We consider $(1, 1)$ instead of the more general $(1, p)$ because, these can be reduced to either $(1, 1)$ or $(1, 0)$ by the belt trick (Fig. 41).

[30]This is closed related to "standard fibered torus", a notion used in the study of Seifert fiber space. In this language, **A1** may be violated at a disk, which becomes a solid torus containing an exceptional fiber under the $(p, q)$ revolution.

### E.2 Fusion rules, quantum dimensions and consistency relations

We summarize facts about anti-sectors and the constraints of the fusion rules. These facts can be derived straightforwardly by applying a standard method of entanglement bootstrap (see, for instance, Section 4.3 of Ref. [1]). We comment on which of the statements can be understood by the dimensional reduction methods developed in Appendix E.1.

**Point particles:** The fusion multiplicities for point particles $\{N_{ab}^c\}$ for $a, b, c \in \mathcal{C}_{\text{point}}$ satisfy:

1. Associativity

$$\sum_{i \in \mathcal{C}_{\text{point}}} N_{ai}^d N_{bc}^i = \sum_{j \in \mathcal{C}_{\text{point}}} N_{ab}^j N_{jc}^d. \tag{E.1}$$

2. Conditions related to the vacuum and the existence of antiparticles:

$$\begin{aligned} N_{1a}^c &= \delta_{a,c}\,, \\ N_{ab}^1 &= \delta_{b,\bar{a}}, \\ N_{ab}^c &= N_{\bar{b}\bar{a}}^{\bar{c}}\,. \end{aligned} \tag{E.2}$$

3. The antiparticle of $a \in \mathcal{C}_{\text{point}}$, denoted as $\bar{a} \in \mathcal{C}_{\text{point}}$, satisfies $\bar{1} = 1$ and $\bar{\bar{a}} = a$.

4. The set of quantum dimensions $\{d_a\}_{a \in \mathcal{C}_{\text{point}}}$, defined according to Eq. (2.10), is the unique positive solution of

$$d_a d_b = \sum_c N_{ab}^c d_c. \tag{E.3}$$

   This further implies, $d_1 = 1$ and $d_{\bar{a}} = d_a \geq 1$.

5. Symmetry under the exchange of the two lower labels:

$$N_{ab}^c = N_{ba}^c. \tag{E.4}$$

**Remark.** All the statements in the list above, except Eq. (E.4), can be understood from the dimensional reduction of the vacuum. The symmetry $N_{ab}^c = N_{ba}^c$ is, however, not true for a generic 2d gapped boundary, where $a, b, c$ are the boundary excitations. This implies that the gapped boundary obtained by the dimensional reduction of a 3d theory is special.

**Shrinkable loops and fluxes:** By the dimensional reduction of the vacuum, the set of shrinkable loops in $\mathcal{C}_{\text{loop}}$ corresponds to the set of anyons of a 2d entanglement bootstrap problem. More precisely, we have a map,

$$R : \mathcal{C}_{\text{loop}} \to \mathcal{C}_{\text{anyon}}^{[1]}. \tag{E.5}$$

Here, $\mathcal{C}_{\text{anyon}}^{[1]}$ represents the set of anyons obtained by the dimensional reduction of the vacuum. This map preserves the multiplicities and the quantum dimensions:

$$N_{l_1 l_2}^{l_3} = N_{R(l_1)R(l_2)}^{R(l_3)}, \quad d_l = d_{R(l)}, \quad \forall l_1, l_2, l_3, l \in \mathcal{C}_{\text{loop}}. \tag{E.6}$$

We define the anti-sector for shrinkable loops, such that $R(\bar{l})$ is the anti-anyon of $R(l)$. An interesting consequence is:

**Proposition E.1.** *In the 3d theory, we have*

$$N_{\bar{l}}^{\bar{a}} = N_l^a \,, \tag{E.7}$$

$$d_l = \sum_{a \in \mathcal{C}_{\text{point}}} N_l^a d_a \,, \quad l \in \mathcal{C}_{\text{loop}} \,, \tag{E.8}$$

$$d_a = \frac{1}{\mathcal{D}^2} \sum_{l \in \mathcal{C}_{\text{loop}}} N_l^a d_l \,, \quad a \in \mathcal{C}_{\text{point}} \,, \tag{E.9}$$

$$\mathcal{D}^2 = \sqrt{\sum_{l \in \mathcal{C}_{\text{loop}}} d_l^2} \,. \tag{E.10}$$

*Here, $\mathcal{D}$ is the total quantum dimension of the 3d system, defined in Eq. (3.4).*

(Note that we gave 3d proofs of (E.8) and (E.9) in §6.2.1.)

*Proof.* First, we apply the dimensional reduction of the vacuum to convert the problem to that of a 2d system with a gapped boundary. After that, we see that each statement is converted to a consistency relation for anyons and boundary excitations. We use the known 2d results for gapped boundaries, previously derived in the framework of entanglement bootstrap; see Ref. [2] and also Section VI E of [16]. □

The subset $\{N_l^1\} \subset \{N_l^a\}$ is the set of condensation multiplicities. For a generic gapped boundary, some of these numbers can be greater than 1. However, because the boundary obtained by the dimensional reduction is special, we have:

$$N_l^1 = \sum_{\mu \in \mathcal{C}_{\text{flux}}} \delta_{l,\varphi(\mu)} \,. \tag{E.11}$$

Here the map $\varphi : \mathcal{C}_{\text{flux}} \to \mathcal{C}_{\text{loop}}$ is that defined in Eq. (3.10). We see that $N_l^1$ must be either 0 or 1, because $\varphi$ is an embedding. From the dimensional reduction point of view, $\bar{\mu}$ is the flux such that $\varphi(\bar{\mu})$ is the anti-sector of $\varphi(\mu)$.

**Remark.** In fact, the number of point particles is always equal to be number of fluxes: $|\mathcal{C}_{\text{point}}| = |\mathcal{C}_{\text{flux}}|$. This result has been conjectured in [27]. We shall provide an entanglement bootstrap derivation of this formula in [20].

**The set $\mathcal{C}_{\text{Hopf}}^{[\mu]}$:** Properties of $\mathcal{C}_{\text{Hopf}}^{[\mu]}$ can be studied by the flux reduction. We denote the map to the 2d anyon theory as:

$$R : \mathcal{C}_{\text{Hopf}}^{[\mu]} \to \mathcal{C}_{\text{anyon}}^{[\mu]} \,. \tag{E.12}$$

Here, $\mathcal{C}_{\text{anyon}}^{[\mu]}$ represents the set of anyons obtained by the dimensional reduction of flux $\mu$. This map preserves the multiplicities:

$$N_{\eta_1 \eta_2}^{\eta_3} = N_{R(\eta_1)R(\eta_2)}^{R(\eta_3)} \,, \quad \forall \eta_1, \eta_2, \eta_3 \in \mathcal{C}_{\text{Hopf}}^{[\mu]} \,. \tag{E.13}$$

The quantum dimensions are related by

$$d_\eta = d_\mu^2 \, d_{R(\eta)} \,, \quad \forall \eta \in \mathcal{C}_{\text{Hopf}}^{[\mu]} \,. \tag{E.14}$$

**Proposition E.2.** *Let $\mathcal{D}^{[\mu]}_{\text{anyon}}$ be the total quantum dimension of the anyons in $\mathcal{C}^{[\mu]}_{\text{anyon}}$. $\mathcal{D}$ is the total quantum dimension of the 3d system. Then*

$$\mathcal{D}^{[\mu]}_{\text{anyon}} = \frac{\mathcal{D}^2}{d_\mu^2}, \quad \forall \mu \in \mathcal{C}_{\text{flux}}. \tag{E.15}$$

*Furthermore,*

$$\sqrt{\sum_{\eta \in \mathcal{C}^{[\mu]}_{\text{Hopf}}} d_\eta^2} = \mathcal{D}^2, \quad \forall \mu \in \mathcal{C}_{\text{flux}}. \tag{E.16}$$

*Proof.* The $\mu = 1$ case follows from Proposition E.1. The total quantum dimension $\mathcal{D}^{[1]}_{\text{anyon}} = \mathcal{D}^{[\mu]}_{\text{anyon}} \cdot d_\mu^2$, and this follows from the calculation of the difference of the topological entanglement entropies for the dimensional reduction of $\mu$ and 1. Finally, Eq. (E.16) is derived by applying Eq. (E.14). □

### E.3 Constraints from topologically-nontrivial paths

In the previous section, we already observed a few facts that are not obvious from dimensional reduction but are, nonetheless, simple to see from the 3d point of view. The most obvious one is $N^c_{ab} = N^c_{ba}$ for point particles. This is a consequence of the (generalized) isomorphism theorem. The crucial observation is the existence of a nontrivial path that permutes the two holes of a ball minus two balls. In the 3d view, deformations not obvious in 2d can be seen.

Below, we discuss several constraints that arise from similar considerations. The goal is to illustrate the general idea. Some are already mentioned in previous sections or references.

First, let us consider paths formed by regions embedded in a ball.

- The absence of topologically nontrivial path is of importance as well. In 2d, it is impossible to flip an annulus inside out by deforming it on a large disk. This gives rise to a well-defined "reference frame" to compare anyon types (Lemma 4.3 of Ref. [1]).

- In 3d, the same observation generalizes: a sphere shell cannot be flipped inside out by deformations within a ball. Here we require that the sphere shell to remain embedded.

- In 3d, it is easy to flip a solid torus. This path is topologically nontrivial and it gives rise to the map $\mu \to \bar{\mu}$; we have illustrated this map in Fig. 9(b).

- A similar flip on the torus shell $\mathbb{T}$ generalizes a relabeling of $\mathcal{C}_{\text{Hopf}}$. Let us denote this map as $f : \mathcal{C}_{\text{Hopf}} \to \mathcal{C}_{\text{Hopf}}$. Then it is obvious that if $\eta \in \mathcal{C}_{\text{Hopf}}$ then $f(\eta) \in \mathcal{C}^{[\bar{\mu}]}_{\text{Hopf}}$. Therefore, $f$ maps $\mathcal{C}_{\text{loop}}$ to itself. As a consequence

$$N^a_{f(l)} = N^a_l, \quad f \circ \phi(a) = \phi(a), \quad f \circ \varphi(\mu) = \varphi(\bar{\mu}). \tag{E.17}$$

  The first two conditions can be seen by examining the deformation of a ball minus an unknot. The third condition is implied by the discussion above.

Secondly, let us consider paths formed by regions embedded in a sphere. (Spheres are available by the sphere completion lemma 3.1.)

- In 2d, on a sphere, it is possible to flip the annulus inside out. This nontrivial path induces an automorphism of the information convex set of the annulus. This maps $a \to \bar{a}$, and it provides an intuitive understanding of antiparticles.

- In 3d, on a 3-sphere, it is possible to flip a sphere shell inside out. Similarly, this generates an automorphism of the information convex set of the sphere shell, such that particles are mapped to antiparticles: $a \to \bar{a}$, for $a \in \mathcal{C}_{\text{point}}$.

- The complement of torus shell $\mathbb{T}$ on 3-sphere is the Hopf link. A path which permutes the pair of linked loops generates the map $\eta \to \eta^\vee$ (§3.2).

- Knot complement $S^3 \setminus K$ can be deformed back to itself by nontrivial paths as well. (This applies to trefoil knot, for example.) It remains to be seen what can be learned from this.

Thirdly, we discuss the extra freedom of deformation provided by paths with immersed regions.

- Paths with immersed regions can turn a sphere shell inside out; this is essentially sphere eversion [46–48]. We conjecture that all sphere eversion processes generate the same permutation of labels in $\mathcal{C}_{\text{point}}$, namely $a \to \bar{a}$. (Note that this process does not have any 2d analog. An annulus cannot be turned inside out with a sequence of immersed regions as intermediate steps within a large disk because the winding number of the boundary is an invariant.)
- Paths with immersed regions can turn a torus shell inside out as well. In fact, there are many inequivalent ways to do so; see [59, 60]. These maps should provide inequivalent ways of mapping $\mathcal{C}_{\text{Hopf}}$ to itself. It remains to be seen what can be learned from these maps.

Finally, let us remark that these do not exhaust all the techniques to generate constraints on the fusion data beyond dimensional reduction. Other known examples include restricting a sectorizable region to a subsystem (Proposition 2.23), and the spiral maps (§5), which combine both partial trace and homotopy through immersions.

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
