# Peer review of "Knots and entanglement"

_SciPost Physics, doi:SciPost Phys. 14, 141 (2023)_

## Round 1 · Referee Report · Anonymous (Referee 1) · 2022-6-30

Strengths

1- This manuscript extends the authors' previous work on entanglement bootstrap from two to three spatial dimensions. It is an independent and distinguished framework, comparing to parallel approaches.

2- This work provides a meaningful connection between quantum information, in particular, quantum entanglement, and liquid topological ordered phases.

3- Most of the physical results obtained are reasonable.

Weaknesses

1- Unclear, inaccurate writing, especially in Section 2. I list a few examples below. (1) A few lines above Example 2.11, $I$ was for an interval, but in and after Example 2.11, $I$ is for sector label, without any clarification. Moreover, $I={a,b,c}$ ought to be an ordered triple instead of merely a set. This abuse of notation also happens multiple times elsewhere, and can be more confusing, for example in Proposition 2.18. (2) Proposition 2.22 "Let $\Omega$ be a region embedded in $S$", some conditions seem missing for such embedding, since based on the proof, $\Omega$ can not share boundary regions with $S$. (3) $\eta^\vee$ in Figure 8 is confusing. According to Point 3 on Page 31, $^\vee$ is applied to the label of the entire Hopf-link excitation. (4) Point 4 on Page 32, below equation (3.10) "Deform BC to AB by a path" seems to flip the interior to the exterior of a solid torus. Can the author be more explicit about the path, which is quite counterintuitive?

2- Not written in a self-contained way. Poor organization. The authors frequently omit proofs similar to those in earlier works, or postpone proofs and details to the appendix. This style is acceptable, however, given the current length of the manuscript, not necessary. In most cases it does not make the storyline more smooth and adds pain to reading.

3- This part is related to point 1- and 2-, but let me put it as a standalone point for emphasis: The notion of immersion does not receive serious treatment (or maybe it does, but not clear exposition): (1) Definition 2.1. What is the precise meaning of "lift" and "layered regions"? Why is it always possible to lift? (2) In the followings, since the authors kept omitting proofs while referring to their previous work, and keeping only one datum $\Omega$ while dropping $\Omega^+$ and $\mathfrak p$, I cannot find the very detail about, and thus get convinced that "We generalize the information convex set to a new class of regions called immersed regions, promoting various theorems to this new context" as claimed in the abstract. (3) In most later applications after Section 2, the authors simply write "regions". It is unclear where the immersion really plays a role.

Report

The article suffers from an awkward balance between mathematical rigor and physical ideas. The mathematical statements should be, at least, precise enough for experts to follow and repeat. The manuscript failed to reach such a point. However, the authors seem to have good enough intuitions and pictures underneath, by which they obtain quite reasonable physical results. A recommendation for publication should be promising if the authors could make the exposition more clear and precise.

Requested changes

1- Duplicated references [32] and [50].

---

## Round 1 · Referee Report · Anonymous (Referee 2) · 2022-8-4

Strengths

  1. Big piece of work with in-depth discussion of many aspects

  2. Original idea and approach

Weaknesses

  1. The paper is too long and technical

  2. I do not think it fits the readership of SciPost Physics

Report

Our understanding of topological entanglement entropy in two spatial dimension [6] is the culmination of decades of research on the Chern-Simons theory and its relation with conformal field theory, knots and the geometry of three-manifolds. Many deep mathematical aspects entering these theories have eventually become understandable by a large theoretical physics audience. Therefore, there are many papers on this subject in physics journal, also discussing experimental and numerical verifications. The situation in one dimension higher is very different.

This paper originates from an abstract reformulation of known properties of 2d entanglement by some some of the authors and extends this approach to 3d. It is rather clear that this brave step is very technical and should be first proposed to a mathematical physics audience, writing a paper in the language appropriate for it and submitting it to e.g. Commun. Math. Phys., not to SciPost Physics.

The paper is not only theorems and lemmas. There is an example in Section 4, the Kitaev quantum double model, where this approach is put in practice.

Requested changes

I do not think this paper is appropriate for SciPost Physics.

If I were the authors, I would write another paper, that contains Section 4 only, the idea being that of introducing concepts from the example. I would refer to the longer math-oriented paper for proofs. This other paper would be definitely short, could be appropriate for SciPost and would let many colleague appreciate the novel approach to entanglement.

---

## Round 2 · Referee Report · Anonymous (Referee 1) · 2023-2-23

Report

The authors have put many efforts into the revision and the presentation is much more clear.

---

## Round 2 · Referee Report · Anonymous (Referee 2) · 2023-2-24

Report

I think the revised version of the paper can be accepted.

The length and math setting (based on theorems, corollaries, remarks, etc) do not help to digest the content, but we cannot ask for a complete rewriting. Maybe in future works, the Authors might consider using a different style that could be more effective in conveying their results. It is also true that papers in this field of topological phases of matter are often very technical.

Nonetheless, the paper presents interesting material that is worth publishing. The present study of entanglement entropy in combination with topological properties of braiding and fusing is very interesting and promising.

---

## Round 2 · Author Response

The first referee report is generally positive but expressed reservations about the organization and presentation of our results. These comments were quite helpful to us; we have revised the draft accordingly and replied in detail below. The second referee report expresses concerns about the readership of SciPost Physics; we argue below that our work is suitable for SciPost Physics by examining the journal's criteria for appropriate subjects and demonstrating previous publication records. We made corresponding revisions and would like to resubmit the paper for publication in SciPost Physics.

Response to the 1st referee report:

Here we address in turn specific points raised by the first referee:

Strengths

1- This manuscript extends the authors' previous work on entanglement bootstrap from two to three spatial dimensions. It is an independent and distinguished framework comparing to parallel approaches. 2- This work provides a meaningful connection between quantum information, in particular, quantum entanglement, and liquid topological ordered phases. 3- Most of the physical results obtained are reasonable.

We are grateful for the positive feedback about its originality and interdisciplinary nature.

Weaknesses

1- Unclear, inaccurate writing, especially in Section 2. I list a few examples below.

(1) A few lines above Example 2.11, $I$ was for an interval, but in and after Example 2.11, $I$ is for sector label, without any clarification. Moreover, $I = {a,b,c}$ ought to be an ordered triple instead of merely a set. This abuse of notation also happens multiple times elsewhere and can be more confusing, for example in Proposition 2.18.

We kept $I$ for the ordered tuple of superselection sector labels and have clarified in the text our use of this symbol. A new symbol $\mathbb{I}$ was introduced to represent an interval.

(2) Proposition 2.22 ``Let $\Omega$ be a region embedded in $S$", some conditions seem missing for such embedding since based on the proof, $\Omega$ can not share boundary regions with $S$.

Thanks for pointing this out! This is related to a step we should not have omitted in our proof! The statement of the proposition holds without change (even if $\Omega$ and $S$ share boundary regions) and we have strengthened our proof accordingly.

(3) $\eta^{\vee}$ in Figure 8 is confusing. According to Point 3 on Page 31, $^\vee$ is applied to the label of the entire Hopf-link excitation.

Indeed the referee is correct that $\eta^\vee$ refers to an operation on the whole Hopf excitation. We have tried to clarify our figure by explaining in the caption that the label on a Hopf excitation is associated with a neighborhood of one of the loops in a manner analogous to the labels on topological particle and anti-particle excitations (shown in the same figure).

(4) Point 4 on Page 32, below equation (3.10), ``Deform BC to AB by a path" seems to flip the interior to the exterior of a solid torus. Can the author be more explicit about the path, which is quite counterintuitive?

We have added some explanation about this deformation to the text. The upshot is that a 3-sphere is the union of a pair of solid tori (i.e., the interior and the exterior that the referee mentioned). Exchanging the two can be done by rotating the 3-sphere.

2- Not written in a self-contained way. Poor organization. The authors frequently omit proofs similar to those in earlier works or postpone proofs and details to the appendix. This style is acceptable. However, given the current length of the manuscript, not necessary. In most cases, it does not make the storyline more smooth and adds pain to reading.

Indeed our paper frequently uses previous work, especially from the paper Fusion Rules from Entanglement. The proofs of various statements in that paper also work in three dimensions. While we don't think it would improve the readability to copy those proofs into our paper, we have tried in the revision to make the reading experience more pleasant by adding very specific references to parts of this paper and others where they arise. We feel strongly that sequestering some of the proofs and details to the Appendices (really just the Associativity Theorem) is also a way to make the reading experience more pleasant.

3- This part is related to points 1- and 2-, but let me put it as a standalone point for emphasis: The notion of immersion does not receive serious treatment (or maybe it does, but not clear exposition):

(1) Definition 2.1. What is the precise meaning of "lift" and "layered regions"? Why is it always possible to lift?

We are grateful to the referee for pointing out that we have not defined these terms precisely and that they are necessary to understand our (important) definition of the concept of immersed region. In the revision, we solved this problem. More below on this.

(2) In the following, since the authors kept omitting proofs while referring to their previous work and keeping only one datum $\Omega$ while dropping $\Omega_+$ and $\mathfrak{p}$, I cannot find the very detail about, and thus get convinced that "We generalize the information convex set to a new class of regions called immersed regions, promoting various theorems to this new context" as claimed in the abstract.

With our newly refined definition of immersed regions, we are optimistic that the logic of this section will be much improved.

(3) In most later applications after Section 2, the authors simply write "regions." It is unclear where immersion really plays a role.

Immersed regions are much broader than usually considered embedded regions in a ball. For example, most punctured 2-manifolds and 3-manifolds cannot be embedded in a ball. Immersion plays a crucial role whenever the region is not embedded.

The essence of immersed regions (Definition~2.1) is the fact that they are locally embedded (defined by local homeomorphisms). With this, we can clearly define the Hilbert space associated with them as the tensor product of the Hilbert spaces associated with embedded pieces (Definition 2.2). Then, we can discuss their information convex set (Definition~2.3).

We thank the referee for pointing out our sloppy use of language related to immersion. In the revision, we have clarified our use of all of these terms, improved the definition of the immersed region, and added a figure to make the definition more digestible.
(Also, we avoided using $\Omega_+$ (and $\widehat{\Omega}_+$) in the definition of immersed regions to be consistent with previous references. That is, in previous papers on entanglement bootstrap, only regions that can be thickened are used in the definition of information convex set, but the thickening is left implicit.)

Further, we have tried to clarify which statements are applicable to general immersed regions and which require an embedded region. To be more specific, the following concepts and statements apply to general immersed regions:

1. Information convex set in Definition 2.3
2. Generalized isomorphism theorem in Theorem 2.5
3. Merging theorem in Theorem 2.7
4. Simplex theorem in Theorem 2.10
5. Extreme point criterion in Lemma 2.13
6. Hilbert space theorem in Theorem 2.14
7. Associativity theorem in Theorem 2.22
8. Proposition 5.4 (It's about density matrices on two knots, which are embedded in a general immersed region.)
9. Decoupling lemma in Lemma D.1

While following items are somewhat restricted to embedded regions:

1. Quantum dimension in Definition 2.11 (Some immersed regions may not have a vacuum sector) and all the statements involved with quantum dimension.
2. Sectorizable region and restriction in Proposition 2.23. ($\Omega$ is embedded in $S$ while $S$ could be a general sectorizable region that is immersed.).
3. Vacuum lemma in Lemma 3.2 ($\Omega$ is embedded in a ball.).
4. 2d and 3d topological entanglement entropy.

Thanks again for the helpful feedback.

Report

The article suffers from an awkward balance between mathematical rigor and physical ideas. The mathematical statements should be, at least, precise enough for experts to follow and repeat. The manuscript failed to reach such a point. However, the authors seem to have good enough intuitions and pictures underneath, by which they obtain quite reasonable physical results. A recommendation for publication should be promising if the authors could make the exposition more clear and more precise.

We have revised the manuscript to be more precise and clear so that it's easier for experts to repeat the results. We agree that some of the results in this manuscript heavily rely on our previous work, which could prevent readers from repeating those results without referencing previous papers. From this perspective, it is indeed less self-contained. However, we made the decision that it's better not to repeat previous proofs and techniques here again because it's an unnecessary redundancy and will make this manuscript much longer than it needs to be. We accommodate this issue by citing propositions and theorems in specific locations of previous works so that readers could easily check previous proof techniques if they are interested. Currently, we think it's the best way to organize this paper. We're open and grateful to suggestions if the referee has better ideas.

Requested changes

1- Duplicated references [32] and [50].

We thank the referee for the careful check of reference duplication. Now we avoided duplication by citing [32] consistently.

Response to the 2nd referee report:

Strengths

  1. Big piece of work with an in-depth discussion of many aspects
  2. Original idea and approach

We thank the referee for the positive evaluation of the originality of the manuscript.

Weaknesses

  1. The paper is too long and technical
  2. I do not think it fits the readership of SciPost Physics

Requested changes

I do not think this paper is appropriate for SciPost Physics. If I were the authors, I would write another paper that contains Section 4 only, the idea being that of introducing concepts from the example. I would refer to the longer math-oriented paper for proofs. This other paper would be definitely short, could be appropriate for SciPost and would let many colleague appreciate the novel approach to entanglement.

The referee raised concerns about the readership of the paper and whether SciPost Physics is appropriate for this manuscript. One particular suggestion was to split the paper into two: theorems and examples, and publish only the examples in SciPost.

We argue that our paper is suitable for readers of SciPost Physics and provide reasons that it is more appropriate not to split the paper.

  1. The examples are a crucial part of the paper. They not only confirm the derived formulas but also show that the range of application is nontrivial. For physical readers, examples are also useful for people to absorb the content of the more abstract propositions and theorems.

  2. Another evidence that our paper is suitable for SciPost Physics is shown in this link which says:

    ``SciPost Physics is home to exemplary and visionary scholarship; as far as content is concerned, scientific quality is what matters.

    The format is flexible and should reflect the nature of the content. Articles can be short letter-style communications of breakthroughs or in-depth reports of groundbreaking research. There is no minimum or maximum length: the article should however constitute a complete, self-contained unit.

    The target audience is similarly flexible: it can be researchers in many different subject areas or specialists in one. Broad interest is appreciated but not required."

    The lengthy technical details in this paper are necessary to make it a complete, self-contained unit.

  3. There are already many long and technical papers published in SciPost Physics. Some recent examples can be easily found. Here we list three of them:

  4. We appreciate the fact that the referee suggests that the content can be of interest to mathematical physics journals, e.g., ``Commun. Math. Phys.". While we agree with this judgment to some extent, we believe that the interdisciplinary nature of this work is pronounced enough that it is better suited for SciPost Physics. What we mean is that the main tools are strong subadditivity and quantum Markov states (quantum physics, quantum information) and topology of 3-manifolds, immersion, and knot theory (math); the physical systems we study are gapped liquid topological orders (condensed matter physics), which is also related to topological quantum field theories (high energy theory). This is why we think SciPost Physics, a journal of interdisciplinary nature, is a favorable place for our work.

    We reserve the referee's suggestion for possible future works in the entanglement bootstrap program. For instance, future collaborated work with mathematicians could be an instance where the referee's suggested journal is very well suited.

---

## Round 2 · List of Changes

List of changes

Section 2.1

  1. We improved the definition of immersed regions (Definition 2.1). We avoided using thickened regions $\Omega_+$ (and $\widehat{\Omega}_+$) in the definition of immersed regions to be consistent with previous references. These thickened regions will only appear later when we want to define information convex sets.

  2. We improved figure 1 to make the definition more digestible.

  3. We separated the definition of Hilbert space of immersed region from the previous Definition 2.1. Now it is Definition 2.2.

  4. Various remarks about Definition 2.1 are improved accordingly.

  5. Mentioned the usage of thickening $\Omega_+$ before introducing information convex sets.

  6. Added Eq.(2.6)

  7. A minor rewriting of the definition of information convex set in a way closer to that in the previous references. (Definition 2.3.)

  8. Improved Example 2.4 and Figure 2 so that the branch cuts labeling is explained.

Sections 2.2 and 2.3

  1. Page 19 and later: We kept $I$ for the ordered tuple of superselection sector labels and have clarified in the text our use of this symbol. A new symbol $\mathbb{I}$ was introduced to represent an interval.

  2. We have strengthened our proof of Proposition 2.23 (i.e., Proposition 2.22 of the previous version.)

The rest of the paper

  1. The caption of Fig. 8 is improved. We have tried to clarify our figure by explaining in the caption that the label on a Hopf excitation is associated with a neighborhood of one of the loops in a manner analogous to the labels on topological particle and anti-particle excitations (shown in the same figure).
  2. The paragraph below Eq.(3.7) is improved to explain the physics of Hopf sectors better.

  3. An improvement is made below Eq.(3.10). We have added some explanation about this deformation to the text. The upshot is that a 3-sphere is the union of a pair of solid tori (i.e., the interior and the exterior that the referee mentioned). Exchanging the two can be done by rotating the 3-sphere.

  4. (multiple places) We have tried to clarify which statements apply to general immersed regions and which require an embedded region. Explicitly, we updated the term "immersed region" in the first line of Proposition 5.4 (Page 63) and the first line of Lemma D.1 (Page 101).

  5. We avoided duplication by citing [32] consistently.

---

## Editorial Decision

published